# MeFBO: A Moreau Envelope Based First-Order Stochastic Gradient Method for Nonconvex Federated Bilevel Optimization

## Abstract

Federated Bilevel Optimization (FBO) enables training machine learning models with nested structures across distributed devices while preserving data privacy. However, current FBO methods often impose restrictive assumptions, particularly the requirement of strong convexity in the lower-level objective. To overcome this limitation, we propose a first-order stochastic gradient method for general FBO problems, leveraging a Moreau envelope-based min-max optimization reformulation to handle potentially non-convex lower-level objectives. Unlike implicit gradient methods, our approach eliminates the need for second-order derivative information. We also establish rigorous theoretical guarantees for convergence rate and communication complexity, demonstrating linear speedup as the number of devices increases. Numerical experiments validate the effectiveness and efficiency of our method, showing comparable or superior performances in challenging scenarios, including federated loss function tuning on imbalanced datasets and federated hyper-representation.

## 1 Introduction

Bilevel optimization has gained prominence in machine learning due to its effectiveness in solving nested structural problems, with applications in areas such as hyperparameter tuning (Franceschi et al., 2018; Bao et al., 2021; Sinha et al., 2024), meta-learning (Franceschi et al., 2018; Jia & Zhang, 2024), and reinforcement learning (Hu et al., 2024b; Yang et al., 2024). These approaches rely on access to the entire dataset, raising concerns about privacy leakage. With the growing importance of data privacy, federated bilevel optimization (FBO) has emerged as a crucial paradigm. FBO enables collaborative learning on distributed datasets while preserving individual privacy, addressing complex nested optimization problems such as federated reinforcement learning (Ruan et al., 2024; Yin et al., 2024).

FBO combines the challenges of bilevel optimization and federated learning, addressing both nested optimization difficulties and the complexities of distributed learning. Existing algorithms, including AID-based methods such as Huang et al. (2023); Huang (2022), and ITD-based methods such as Xiao & Ji (2023), often require strong convexity in the lower-level objective functions to compute federated hypergradients via the implicit function theorem (Kearns, 1989). The recently proposed single-loop SimFBO algorithm (Yang et al., 2023), based on SOBA (Dagréou et al., 2022), also depends on strong convexity at the lower level. These requirements significantly limit the applicability of current algorithms, as many problems naturally involve non-convex lower-level objectives, such as in Federated Transfer Learning (Liang et al., 2022; Zhang & Li, 2021).

There have been numerous recent studies addressing the issue of non-strong-convexity in the lower-level of bilevel optimization (BLO) under the single-machine (non-federated) setting (Liu et al., 2022; Kwon et al., 2023; Huang, 2023b; Yao et al., 2023; Liu et al., 2024; Kwon et al., 2024). Given these significant advancements in addressing lower-level non-strong-convexity in BLO, a natural question arises:

> Can we develop a stochastic gradient method for FBO that does not require strong convexity in the lower-level problem, while reducing computation and communication costs?

To achieve this goal, several challenging issues must be addressed.

**Inapplicability of the implicit function theorem.** Current approaches to FBO rely heavily on the implicit function theorem, particularly for computing hypergradients—a key component of the optimization process. While powerful, the implicit function theorem imposes strict conditions on the problem structure, most notably the requirement for strong convexity in the lower-level objective.

**High computational cost of Hessian-Jacobian computations.** Existing FBO algorithms often require matrix-vector products involving the Hessian or Jacobian matrices of the lower-level objective, which are computationally expensive. Developing new algorithms that eliminate the need for Hessian or Jacobian computations is essential for scaling to large-scale applications.

**Linear speedup challenges under lower-level non-strong convexity.** In federated settings, achieving linear speedup with respect to the number of devices is critical for fully leveraging distributed resources. However, it remains unclear whether convergence rates can attain linear speedup in the absence of strong convexity at the lower level in FBO.

### 1.1 MAIN CONTRIBUTIONS

To address the aforementioned challenges, inspired by the recent advances of Liu et al. (2024) in bilevel optimization, we develop a first-order stochastic gradient method, named MeFBO, for federated bilevel optimization based on a Moreau envelope-based minimax optimization reformulation, which allows for nonconvexity in the lower-level problem.

- To the best of our knowledge, MeFBO is the first federated bilevel optimization approach that uses only first-order derivatives while addressing nonconvex lower-level problems.

- We establish rigorous theoretical guarantees for convergence rate and sample complexity, demonstrating linear speedup with respect to the number of participating clients (sampling without replacement).

- We validate MeFBO empirically on three federated bilevel learning tasks: federated hyper-representation, federated loss function tuning on imbalanced datasets, and federated data hyper-cleaning. The results demonstrate that MeFBO achieves comparable or superior, and more robust, performance compared to existing FBO approaches.

Table 1: Comparison of MeFBO with closely related FBO approaches. Below, LL-convexity refers to the convexity condition of the lower-level objective function. LL-Lipschitz continuity denotes the Lipschitz continuity requirement of the lower-level problem, which involves multiple orders of information. H-J free indicates that the method does not require Hessian or Jacobian computations. For simplicity, we exclude methods with momentum-based acceleration or those designed to handle system-level heterogeneity.

| Method | LL-convexity | LL-Lipschitz continuity | H-J free | Partial participation | Linear speedup |
|---|---|---|---|---|---|
| FedNest (Tarzanagh et al., 2022) | strongly convex | second-order | ✗ | ✗ | ✗ |
| FBO-AggITD (Xiao & Ji, 2023) | strongly convex | second-order | ✗ | ✗ | ✗ |
| FedBiO (Li et al., 2023) | strongly convex | second-order | ✗ | ✗ | ✓ |
| FedMBO (Huang et al., 2023) | strongly convex | second-order | ✗ | ✓ | ✓ |
| SimFBO (Yang et al., 2023) | strongly convex | third-order | ✗ | ✓ | ✓ |
| MeFBO (ours) | non-convex | first-order | ✓ | ✓ | ✓ |

As this work leverages insights from both Liu et al. (2024) and (Yang et al., 2023), we explain the differences from the most relevant literature.

Compared to our work, MEHA by Liu et al. (2024) is limited to deterministic settings for non-federated bilevel optimization. In contrast, even when reduced to the non-federated setting, our MeFBO is a stochastic algorithm with convergence rate and sample complexity guarantees.

More importantly, efficiently solving federated bilevel optimization problems is even more challenging due to the need to preserve privacy and reduce communication costs, which requires a variable number of local updates on the client side. This adds complexity to both algorithm design and theoretical analysis compared to non-federated bilevel optimization. For example, local client updates introduce challenges in setting algorithm hyperparameters (such as client and server learning rates) and in analyzing communication and sample complexity.

As a result, compared to the analysis in Liu et al. (2024), in the federated learning setting, we must control the bounds of client drifts and harmonize them with the variance from local stochastic gradient estimation and errors from client sampling to achieve convergence. Specifically, it is crucial that these bounds (to be established) explicitly depend on the effects of local update rounds and the number of participating clients per communication round, as discussed in Section 3.2. This significantly increases the complexity of the theoretical analysis.

The comparison between SimFBO (Yang et al., 2023) and our work is highlighted in Table 1. The key difference lies in the starting points of algorithm design. SimFBO is designed and analyzed from the perspective of hypergradient estimation, relying on the strong convexity assumption and involving second-order derivatives. In contrast, we leverage insights from Liu et al. (2024) and use local stochastic gradient estimators.

## 1.2 RELATED WORK

We provide a concise review of recent works directly related to ours, with a more comprehensive review presented in Section G.

FedNest (Tarzanagh et al., 2022), one of the earliest FBO methods, is a federated alternating stochastic gradient method (FedNest) that uses AID-based hypergradient estimation to address general federated nested problems. Xiao & Ji (2023) introduced an FBO algorithm employing ITD-based hypergradient estimation, though these approaches do not achieve linear speedup. The study by Huang et al. (2023), which uses AID-based hypergradient estimation, achieves linear speedup without requiring full client participation in every communication round. Recent FBO methods, such as SimFBO (Yang et al., 2023) and FedBiOAcc (Li et al., 2023), draw inspiration from SOBA (Dagréou et al., 2022). These approaches transform a linear system problem into a quadratic one, improving computational efficiency within a single-loop algorithmic framework. Notably, FedBiOAcc incorporates a momentum-based technique. While these algorithms have made significant progress, they continue to rely on Hessian matrix computations and are constrained by the requirement for lower-level strong convexity (LLSC).

In addition to the aforementioned works, there is a growing body of research on bilevel optimization in asynchronous settings, such as those by Jiao et al. (2022) and Li et al. (2024). Furthermore, FBO has demonstrated promising practical applications, particularly in the fine-tuning of large language models (LLMs) within federated settings. For instance, Wu et al. (2024) investigates the use of FBO for local fine-tuning of LLMs. As noted in Table 2 of Wu et al. (2024), these models can be optimized using either single-level or bilevel approaches. Notably, the bilevel optimization method, FedBiOT, proposed by Wu et al. (2024), exhibits significant advantages over single-level optimization, especially in scenarios involving hierarchical problem structures.

## 2 PROPOSED APPROACH

In this work, we study the federated bilevel optimization defined by:

$$\min_{x \in X, y \in Y} \mathbf{F}(x,y) := \sum_{i=1}^{n} w_i f_i(x,y) \quad \text{s.t.} \quad y \in S(x), \tag{1}$$

where $S(x)$ is the set of optimal solutions for the lower-level program

$$(\mathrm{P}_x) \qquad \min_{y \in Y} \mathbf{G}(x,y) := \sum_{i=1}^{n} w_i g_i(x,y).$$

Here $X$ and $Y$ are closed convex sets in $\mathbb{R}^{d_x}$ and $\mathbb{R}^{d_y}$, respectively, and $n$ denotes the total number of clients. For each client $i \in [n] := 1, 2, \ldots, n$, the constant $w_i$ represents the weight of client

$i$, satisfying $\sum_{i=1}^{n} w_i = 1$ and $\frac{\beta_{\min}}{n} \leq w_i \leq \frac{\beta_{\max}}{n}$ for all $i \in [n]$. The upper- and lower-level functions, $f_i(x, y) := \mathbb{E}_{\xi_i}[f_i(x, y; \xi_i)]$ and $g_i(x, y) := \mathbb{E}_{\zeta_i}[g_i(x, y; \zeta_i)]$, respectively, are expressed as expectations with respect to the random variables $\xi_i$ and $\zeta_i$.

## 2.1 MOREAU ENVELOPE-BASED MINIMAX OPTIMIZATION REFORMULATION

The goal of this work is to study first-order stochastic gradient methods for solving problem (1) in the context of federated learning. To this end, inspired by the recent advance in bilevel optimization (Gao et al. (2023); Liu et al. (2024)), we first observe that problem (1) as a bilevel optimization can be reformulated as a constrained problem

$$\min_{(x,y) \in X \times Y} \mathbf{F}(x, y) \quad \text{s.t.} \quad \mathbf{G}(x, y) - \mathbf{v}_\gamma(x, y) \leq 0, \tag{2}$$

where $\mathbf{v}_\gamma(x, y)$ is the Moreau envelope of $\mathbf{G}$ defined as

$$\mathbf{v}_\gamma(x, y) := \min_{\theta \in Y} \left\{ \mathbf{G}(x, \theta) + \frac{1}{2\gamma} \|\theta - y\|^2 \right\} \text{ with } \gamma > 0. \tag{3}$$

Note that $\mathbf{G}(x, y) - \mathbf{v}_\gamma(x, y) \geq 0$ for all $(x, y) \in X \times Y$. A direct and effective way to solve the constrained problem (2) is by addressing its corresponding penalty problem:

$$\min_{(x,y) \in X \times Y} \Upsilon_{c_t}(x, y) := \frac{1}{c_t} \mathbf{F}(x, y) + \mathbf{G}(x, y) - \mathbf{v}_\gamma(x, y), \tag{4}$$

where $c_t > 0$ is a penalty parameter. Recalling the definition (3) of $\mathbf{v}_\gamma$, we further observe that problem (4) is equivalent to the following minimax problem:

$$\min_{(x,y) \in X \times Y} \max_{\theta \in Y} \Upsilon_{c_t}(x, y, \theta) := \frac{1}{c_t} \mathbf{F}(x, y) + \mathbf{G}(x, y) - \mathbf{G}(x, \theta) - \frac{1}{2\gamma} \|\theta - y\|^2. \tag{5}$$

We refer to problem (5) as the **Moreau envelope-based minimax optimization reformulation** of problem (1). This reformulation enjoys two favorable properties that facilitate simpler and more efficient implementation in federated bilevel optimization.

(I) When $\mathbf{G}(x, \cdot)$ is weakly convex, namely, $\mathbf{G}(x, \cdot) + \rho \| \cdot \|^2/2$ is convex for some positive constant $\rho$, then for $\gamma \in (0, 1/\rho)$ the problem (5) is a non-convex-strongly-concave minimax optimization problem. Consequently, the inner maximizer problem has a unique solution $\theta_\gamma^*(x, y)$ for any $(x, y)$. Moreover, at this time, $\mathbf{v}_\gamma(x, y)$ is differentiable and

$$\nabla \mathbf{v}_\gamma(x, y) = \left( \nabla_1 \mathbf{G}(x, \theta_\gamma^*(x, y)), \left( y - \theta_\gamma^*(x, y) \right)/\gamma \right), \tag{6}$$

where $\nabla_1$ represent the gradient of a function with respect to its first variable.

(II) The objective function $\Upsilon_{c_t}(x, y, \theta)$ of problem (5) exhibits a favorable linear structure with respect to the upper and lower level objectives $\mathbf{F}$ and $\mathbf{G}$. Consequently, within the context of federated learning, this gives rise to a local model on the client side:

$$\min_{x,y} \max_{\theta} \Upsilon_{c_t}^i(x, y, \theta) := \frac{1}{c_t} f_i(x, y) + g_i(x, y) - g_i(x, \theta) - \frac{1}{2\gamma} \|\theta - y\|^2. \tag{7}$$

Note that we intentionally disregard the constraints on $x$, $y$ and $\theta$ on the client side, as the corresponding projection operators can be applied on the server side (see Algorithm 1).

## 2.2 PROPOSED ALGORITHM

In this section, we introduce our MeFBO algorithm for federated stochastic bilevel optimization, which supports partial client participation, detailed in Algorithm 1.

**On the client side.** For each communication round $t$, a subset $C^{(t)}$ of participating clients is selected without replacement. Each active client $i \in C^{(t)}$ then performs $\tau$-step stochastic gradient descent ascent (SGDA) on the local model (7) to update the local variables:

$$\left( \theta_i^{(t,k+1)}, x_i^{(t,k+1)}, y_i^{(t,k+1)} \right) \leftarrow \left( \theta_i^{(t,k)} - \eta_\theta^{(t)} h_{i,\theta}^{(t,k)}, x_i^{(t,k)} - \eta_x^{(t)} h_{i,x}^{(t,k)}, y_i^{(t,k)} - \eta_y^{(t)} h_{i,y}^{(t,k)} \right), \tag{8}$$

where $\eta_\theta^{(t)}, \eta_x^{(t)}, \eta_y^{(t)}$ are local learning rates (step sizes), and

$$h_{\theta,i}^{(t,k)} := \nabla_2 g_i(x_i^{(t,k)}, \theta_i^{(t,k)}; \zeta_i^{(t,k)}) + \frac{1}{\gamma}(\theta_i^{(t,k)} - y_i^{(t,k)}),$$

$$h_{y,i}^{(t,k)} := \frac{1}{c_t}\nabla_2 f_i(x_i^{(t,k)}, y_i^{(t,k)}; \xi_i^{(t,k)}) + \nabla_2 g_i(x_i^{(t,k)}, y_i^{(t,k)}; \zeta_i^{(t,k)}) - \frac{1}{\gamma}(y_i^{(t,k)} - \theta_i^{(t,k)}), \qquad (9)$$

$$h_{x,i}^{(t,k)} := \frac{1}{c_t}\nabla_1 f_i(x_i^{(t,k)}, y_i^{(t,k)}; \xi_i^{(t,k)}) + \nabla_1 g_i(x_i^{(t,k)}, y_i^{(t,k)}; \zeta_i^{(t,k)}) - \nabla_1 g_i(x_i^{(t,k)}, \theta_i^{(t,k)}; \zeta_i^{(t,k)}),$$

are unbiased estimates of $-\nabla_\theta \Upsilon_{c_t}^i(x, y, \theta)$, $\nabla_y \Upsilon_{c_t}^i(x, y, \theta)$, and $\nabla_x \Upsilon_{c_t}^i(x, y, \theta)$ at $(x_i^{(t,k)}, y_i^{(t,k)}, \theta_i^{(t,k)})$, respectively, where $\nabla_2$ represent the gradient of a function with respect to its second variable. The output of each client $i \in C^{(t)}$ in this stage is the locally averaged stochastic gradient estimators:

$$h_{\theta,i}^{(t)} := \frac{1}{\tau}\sum_{k=0}^{\tau-1} h_{\theta,i}^{(t,k)}, \quad h_{x,i}^{(t)} := \frac{1}{\tau}\sum_{k=0}^{\tau-1} h_{x,i}^{(t,k)}, \quad h_{y,i}^{(t)} := \frac{1}{\tau}\sum_{k=0}^{\tau-1} h_{y,i}^{(t,k)}. \qquad (10)$$

**Communication and aggregating gradients.** In this stage (communication round $t$), each client $i \in C^{(t)}$ sends the averaged stochastic gradient estimators in (10) to the server, which aggregates these local estimators to compute

$$h_\theta^{(t)} := \sum_{i \in C^{(t)}} \widetilde{w}_i h_{\theta,i}^{(t)}, \quad h_x^{(t)} := \sum_{i \in C^{(t)}} \widetilde{w}_i h_{x,i}^{(t)}, \quad h_y^{(t)} := \sum_{i \in C^{(t)}} \widetilde{w}_i h_{y,i}^{(t)}, \qquad (11)$$

where $\widetilde{w}_i := w_i n / |C^{(t)}|$ is the effective weight of participating client $i$.

**On the server side.** Leveraging the aggregated directions $h_\theta^{(t)}, h_x^{(t)},$ and $h_y^{(t)}$ from (11), the server performs one-step projected gradient descent ascent to update the global variables:

$$\left(\theta^{(t+1)}, x^{(t+1)}, y^{(t+1)}\right) = \text{Proj}_{Y \times X \times Y}\left(\theta^{(t)} - \lambda_\theta^{(t)} h_\theta^{(t)}, x^{(t)} - \lambda_x^{(t)} h_x^{(t)}, y^{(t)} - \lambda_y^{(t)} h_y^{(t)}\right), \qquad (12)$$

where $\lambda_\theta^{(t)}, \lambda_x^{(t)},$ and $\lambda_y^{(t)}$ are the server-side learning rates.

---

**Algorithm 1** MeFBO

---

**Input:** initialization $x^{(0)}, y^{(0)},$ and $\theta^{(0)}$, communication rounds $T$, local update rounds $\tau$, client learning rates $\{\eta_\theta^{(t)}\eta_x^{(t)}, \eta_y^{(t)}\}$, server learning rates $\{\lambda_\theta^{(t)}, \lambda_x^{(t)}, \lambda_y^{(t)}\}$, penalty parameter $c_t$, and proximal parameter $\gamma$.
**for** $t = 0, 1, 2, ..., T-1$ **do**
    Sample a subset $C^{(t)}$ of participating clients;
    For client $i \in C^{(t)}$, initialize $\theta_i^{(t,0)} = \theta^{(t)}, x_i^{(t,0)} = x^{(t)}, y_i^{(t,0)} = y^{(t)}$;
    **for** $k = 0, 1, 2, ..., \tau-1$ **do**
        Locally update $\theta_i^{(t,k)}, x_i^{(t,k)},$ and $y_i^{(t,k)}$ simultaneously using Eq.(8);
    **end for**
    Client $i$ computes $\{h_{\theta,i}^{(t)}, h_{x,i}^{(t)}, h_{y,i}^{(t)}\}$ using Eq. (10) and sends the results to the server;
    Server aggregates the local estimators to compute $\{h_\theta^{(t)}, h_x^{(t)}, h_y^{(t)}\}$ using Eq. (11) ;
    Server updates the global variables using Eq. (12) .
**end for**

---

**Remark and other possible algorithmic designs.** (1) Since MeFBO uses only stochastic gradient estimators, it is a first-order stochastic gradient method, making it significantly different from existing FBO methods. Another notable feature of MeFBO is its ability to handle constraints on both the upper- and lower-level variables. Additionally, the corresponding projection operators are implemented exclusively on the server side, enhancing MeFBO's practicality and efficiency. (2) MeFBO also offers strong extensibility. For instance, instead of using SGDA as in (9), one could employ other stochastic gradient estimation techniques (e.g., SAGA, STORM) to develop more advanced federated stochastic bilevel optimization algorithms. Refer to recent advances in bilevel optimization Dagréou et al. (2022); Chen et al. (2023); Huang (2023b); Chu et al. (2024).

## 3 THEORETICAL INVESTIGATION

### 3.1 ASSUMPTIONS

We make the following assumptions in the theoretical investigation.

**Assumption 3.1.** For the upper-level objective, the following conditions hold:

  (i) The UL objective $\mathbf{F}$ is bounded below, i.e., $\underline{F} := \inf_{(x,y)\in X\times Y} \mathbf{F}(x,y) > -\infty$.

 (ii) For each $i \in [n]$, $f_i(x,y)$ is twice continuously differentiable, and $L_f$-Lipschitz continuous. The gradients $\nabla f_i(x,y)$ is $L_1$-Lipschitz continuous, i.e., $f_i(x,y)$ is $L_1$-smooth.

(iii) For each $i \in [n]$, $\nabla f_i(x,y;\xi_i)$ is an unbiased estimator of $\nabla f_i(x,y)$. Furthermore, there exists a constant $\sigma_f$ such that $\mathbb{E}\left[\|\nabla f_i(x,y) - \nabla f_i(x,y;\xi_i)\|^2\right] \le \sigma_f^2$.

**Assumption 3.2.** For the lower-level objective, the following conditions hold:

  (i) For each $i \in [n]$, $g_i(x,y)$ is twice continuously differentiable, and $g_i(\cdot,y)$ is $L_g$-Lipschitz continuous for any $y$, and $g_i(x,y)$ is $L_2$-smooth.

 (ii) For each $i \in [n]$, $\nabla g_i(x,y;\zeta_i)$ is an unbiased estimator of $\nabla g_i(x,y)$, and there exists a constant $\sigma_g$ such that $\mathbb{E}\left[\|\nabla g_i(x,y) - \nabla g_i(x,y;\zeta_i)\|^2\right] \le \sigma_g^2$.

(iii) There exists a constant $\Delta$ such that $\sum_{i=1}^n w_i \|\nabla_y g_i(x,y) - \nabla_y \mathbf{G}(x,y)\|^2 \le \Delta^2$.

Assumption 3.1(ii) and Assumption 3.2(i) impose smoothness and Lipschitz continuity conditions. Notably, we only require the Lipschitz continuity of the first-order derivatives, a key distinction from other FBO literature, which typically also assumes the Lipschitz continuity of second-order derivatives. Assumption 3.1(iii) and Assumption 3.2(ii) are standard assumptions for unbiased gradient estimators and variance bounds in stochastic gradients. In federated learning, Assumption 3.2(iii) is commonly used to bound data heterogeneity. The heterogeneity parameter $\Delta$ represents the level of data heterogeneity, with $\Delta = 0$ corresponds to the homogeneous data setting.

*Remark* 3.3. (1) Assumption 3.2(iii) is used to mitigate the impact of client drifts in $y^{(t)}$ and $\theta^{(t)}$ on the final convergence. It employs a single parameter $\Delta$ to describe the degree of heterogeneity, as also used in Li et al. (2023, Assumption 3.5). This differs from Yang et al. (2023, Assumption 4), which uses two parameters. (2) The upper-level objective does not require a similar assumption to Assumption 3.2(iii) because $f_i(x,y)$ is assumed to be Lipschitz continuous with respect to both $x$ and $y$ in Assumption 3.1(ii).

### 3.2 CONVERGENCE RESULT AND COMPLEXITY ANALYSIS

In this section, we provide the convergence rate and sample complexity of MeFBO under Assumptions 3.1 and 3.2. For simplicity, we let $P := |C^{(t)}|$ for all $t$. When the lower-level problem of bilevel optimization is strongly convex and unconstrained, the hypergradient norm is typically used as a stationary measure for algorithms. Unfortunately, this measure is not easily extendable to nonconvex lower-level objectives. Therefore, we introduce local surrogates.

Inspired by Liu et al. (2024), and by leveraging the stationarity condition of problem (4), we introduce the following stationarity measure for nonconvex federated bilevel optimization:

$$R_t(x,y) := \left[\text{dist}\left(0, \nabla \Upsilon_{c_t}(x,y) + \mathcal{N}_{X\times Y}(x,y)\right)\right]^2. \tag{13}$$

where $\mathcal{N}_{X\times Y}(x,y)$ is the normal cone of $X \times Y$ at $(x,y)$. Clearly, $R_t(x,y)$ is well-defined when the hypergradient is. We refer readers to Lemma A.13 in Liu et al. (2024) for a comparison of these two criteria. Furthermore, $R_t(x,y) = 0$ if and only if $0 \in \nabla \Upsilon_{c_t}(x,y) + \mathcal{N}_{X\times Y}(x,y)$, i.e., the point $(x,y)$ is a stationary point of problem (4).

**Theorem 3.4** (Fixed step size)**.** *Fix the number of communication rounds $T$, local update rounds $\tau$, and the number $P$ of participating clients per communication round. Assume that Assumptions 3.1 and 3.2 hold. Let $c_t = \underline{c}(t+1)^p$ with $\underline{c} > 0$, $p \in (0, 1/4)$, and $\gamma \in (0, \frac{1}{2L_2})$. Consider fixed server*

*and client step sizes*

$$\lambda_\theta^{(t)} = c_\lambda \tau^{\frac{1}{4}} T^{-1/2}, \quad \lambda_x^{(t)} = \lambda_y^{(t)} = c_\theta \lambda_\theta^{(t)}, \quad \eta_x^{(t)} = \eta_y^{(t)} = \eta_\theta^{(t)} = c_\eta \frac{1}{\tau^{7/8} T^{1/4} \sqrt{P}},$$

*where $c_\lambda$, $c_\theta$, and $c_\eta$ are constants given in Lemma D.8. Then the sequence $(x^{(t)}, y^{(t)}, \theta^{(t)})$ generated by Algorithm 1 satisfies*

$$\min_{0 \le t \le T-1} \mathbb{E}\left[R_t(x^{(t+1)}, y^{(t+1)})\right] = \mathcal{O}\left(\frac{1}{P}\left(\frac{1}{T^{1/2}\tau^{1/2}} + \frac{\tau^{1/2}(n-P)}{T^{1/2}n} + \frac{1}{T\tau^{1/4}}\right)\right). \quad (14)$$

Note that the server step sizes in Theorem 3.4 are constant with respect to the number $P$ of participating clients, but the client step sizes scale with $P^{-1/2}$. The proof of Theorem 3.4 refers to that of Theorem D.11 in Appendix D.4. A proof sketch is provided in Appendix D.2. Additionally, we present theoretical results for the **decreasing step size** in Appendix C.

Several remarks about Theorem 3.4 are as follows: (1) By Theorem 3.4, MeFBO achieves a linear convergence speedup with respect to the number of participating clients (sampling without replacement); (2) When $n = 1$, MeFBO reduces to a first-order stochastic gradient algorithm for nonconvex bilevel optimization with a convergence rate of $\mathcal{O}(T^{-1/2})$ by setting $\tau = 1$ in Theorem 3.4; (3) In the case of full client participation (i.e., $P = n$), Theorem 3.4 suggests that increasing the number $\tau$ of local update steps can help improve the convergence rate. In contrast, for partial client participation, increasing $\tau$ may negatively affect the convergence rate. Theorem 3.4 highlights an important trade-off in the selection of local update rounds $\tau$. Next, we further analyze the communication and sample complexity of MeFBO.

**Corollary 3.5** (Fixed step size). *Under the setting of Theorem 3.4, we have the following results:*

(i) *In the case of full client participation, setting $\tau = \mathcal{O}(T)$, the per-client sample complexity is $\tau T = \mathcal{O}(\epsilon^{-2})$ and the communication complexity $T = \mathcal{O}(\epsilon^{-1})$.*

(ii) *For partial client participation, setting $\tau = \mathcal{O}(1)$, the per-client sample complexity is $\tau T = \mathcal{O}(P^{-2}\epsilon^{-2})$ and the communication complexity $T = \mathcal{O}(P^{-2}\epsilon^{-2})$.*

For partial client participation, we set $\tau = \mathcal{O}(1)$ because further increasing $\tau$ does not improve the final convergence rate due to the dominant estimator $\mathcal{O}(P^{-1}\tau^{1/2}T^{-1/2})$ in (14). The proof of Corollary 3.5 is provided in Appendix D.5.

## 4 NUMERICAL EXPERIMENTS

We evaluate MeFBO on three federated bilevel learning tasks: federated hyper-representation, federated loss function tuning on imbalanced datasets, and federated data hyper-cleaning. We compare its performance with other FBO baselines, including SimFBO (Yang et al. (2023)), and FedNest (Tarzanagh et al. (2022)) and LFedNest (Tarzanagh et al. (2022)). Although the motivation and theoretical analyses of these algorithms (except for MeFBO) rely on the strong convexity of the lower-level problem, all of them can be implemented in nonconvex scenarios. Following the experimental setup in Tarzanagh et al. (2022); Li et al. (2023), we use the MNIST dataset (LeCun et al. (1998)) with i.i.d., non-i.i.d., and imbalanced data partitioning methods. To ensure the reliability and stability of our results, all numerical experiments were repeated 10 times, and the reported values represent the average of these repetitions. Details of all experimental specifications are provided in Appendix B.

### 4.1 FEDERATED HYPER-REPRESENTATION LEARNING

In this learning task, the training and validation datasets are distributed among clients. The goal is to learn a representation and a header on the joint training and validation datasets while preserving privacy. Following Tarzanagh et al. (2022); Xiao et al. (2023), the problem can be formulated as a special case of (1), given by

$$\min_x \frac{1}{n} \sum_{i=1}^n f_{ce}(x, y^*(x); \mathcal{D}_{val}^i) \quad \text{s.t.} \quad y^*(x) = \arg\min_y \frac{1}{n} \sum_{i=1}^n f_{ce}(x, y; \mathcal{D}_{tr}^i) + rc\|y\|^2, \quad (15)$$

where $x$ and $y$ are the parameters of the representation layer and the classifier layer, respectively. The datasets $\mathcal{D}_{\mathrm{tr}}^i$ and $\mathcal{D}_{\mathrm{val}}^i$ are the training and validation sets of client $i \in [n]$, respectively. The function $f_{\mathrm{ce}}$ is the cross-entropy loss, and $rc$ is the regularization parameter used to ensure the (possible) strong convexity condition. In previous experiments (Yang et al. (2023); Tarzanagh et al. (2022)), $rc$ was set to 0.05. We refer to Appendix B.1 for the details on the implementation and only discuss the key model hyperparameter $rc$ in the main text.

**Result with the default setting of** $rc = 0.05$**.** Table 2 presents the comparison of test accuracy versus communication rounds in both i.i.d. and non-i.i.d. settings. From these results, we can draw the following observations: From the perspective of mean and variance, MeFBO outperforms the other methods in both i.i.d. and non-i.i.d. settings, indicating that MeFBO is more robust.

Table 2: Comparison of the results about test accuracy v.s. communication rounds for hyper-representation on a 2-layer MLP and MNIST dataset in i.i.d. setiing and non-i.i.d. setting. M-F: MeFBO(ours); S-F: SimFBO (Yang et al., 2023); L-F:L-FedNest (Tarzanagh et al., 2022); F-N: FedNest (Tarzanagh et al., 2022).

| Alg. | i.i.d. | | | non-i.i.d. | | |
|---|---|---|---|---|---|---|
| | 600 | 1000 | 1500 | 600 | 1000 | 1500 |
| F-N | $88.53 \pm 0.26$ | $89.68 \pm 0.17$ | $90.47 \pm 0.16$ | $85.98 \pm 1.64$ | $87.67 \pm 0.46$ | $88.38 \pm 0.83$ |
| L-F | $90.16 \pm 0.19$ | $90.87 \pm 0.14$ | $91.44 \pm 0.11$ | $79.81 \pm 3.90$ | $81.83 \pm 2.11$ | $83.17 \pm 1.89$ |
| S-F | $96.94 \pm 0.23$ | $97.11 \pm 0.20$ | $97.30 \pm 0.17$ | $95.58 \pm 0.64$ | $96.05 \pm 0.39$ | $96.26 \pm 0.45$ |
| M-F | $\mathbf{97.12 \pm 0.07}$ | $\mathbf{97.54 \pm 0.11}$ | $\mathbf{97.72 \pm 0.10}$ | $\mathbf{96.40 \pm 0.13}$ | $\mathbf{96.85 \pm 0.09}$ | $\mathbf{97.09 \pm 0.06}$ |

**Robustness to** $rc$**.** We test the sensitivity of MeFBO and other FBO algorithms to the model hyperparameter $rc$. From Figure 1, we observe the following:

- MeFBO is the most robust to the choice of the regularization parameter $rc$.

- Although the theoretical analyses of SimFBO, LFedNest, and FedNest rely on strong convexity, they achieve the best accuracy when $rc = 0$, not at the default setting of $rc = 0.05$. This indicates that the regularization technique used to enforce strong convexity may degrade performance, highlighting the urgent need to design and study FBO algorithms for non-convex scenarios.

- In repeated experiments with 300 communication rounds, SimFBO (Yang et al. (2023)) achieves the best accuracy when the $rc$ value is not equal to $0.07$. For a more intuitive comparison, please refer to Figure 5.

More comprehensive experimental results are presented in Figures 4 and 5 in appendix A.1.

### 4.2 FEDERATED LOSS FUNCTION TUNING ON IMBALANCED DATASET

Following the federated setting in Tarzanagh et al. (2022), the goal of this task is to tune a loss function for learning on an imbalanced MNIST dataset distributed among clients. The specific formulation and experimental details are provided in Appendix B.2.

**Results.** Figure 2 illustrates the comparative performance of FBO algorithms in terms of test accuracy versus communication rounds, employing local round $\tau = 3$ for the i.i.d. setting and local round $\tau = 1$ for the non-i.i.d. setting. This experimental design serves to showcase the performance of different algorithms with different local rounds under varying degrees of data heterogeneity.

- It clearly shows that MeFBO achieves a faster convergence rate and higher accuracy under different local round $\tau$ , both in the i.i.d and non-i.i.d. setting.

- As illustrated in Figure 2 (b), in a highly heterogeneous environment, it is evident that MeFBO demonstrates enhanced robustness as the number of communication rounds increases.

More comprehensive experimental results are presented in Figures 6 and 7 in Appendix A.2. We also test the sensitivity of MeFBO and the baselines to the regularization parameter $rc$. The results are summarized in Figures 8 and 9 in Appendix A.2, indicating that MeFBO is robust to the choice of $rc$.

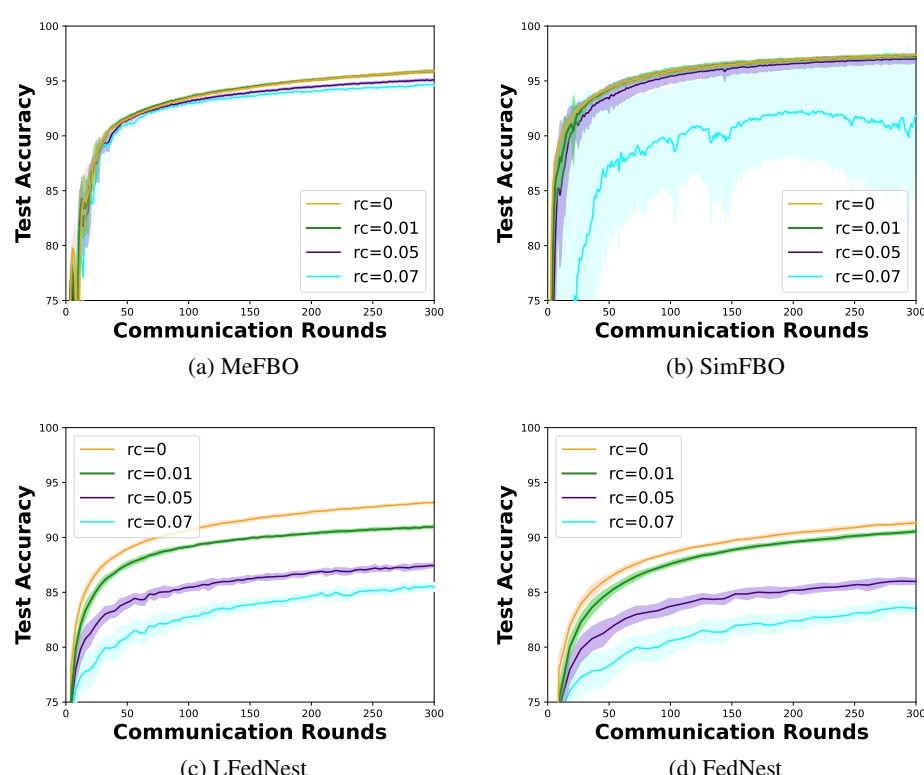

Figure 1: Comparison of the results about test accuracy v.s. communication rounds for federated hyper-representation under varying regularization coefficients $rc$ of lower-level objectives.

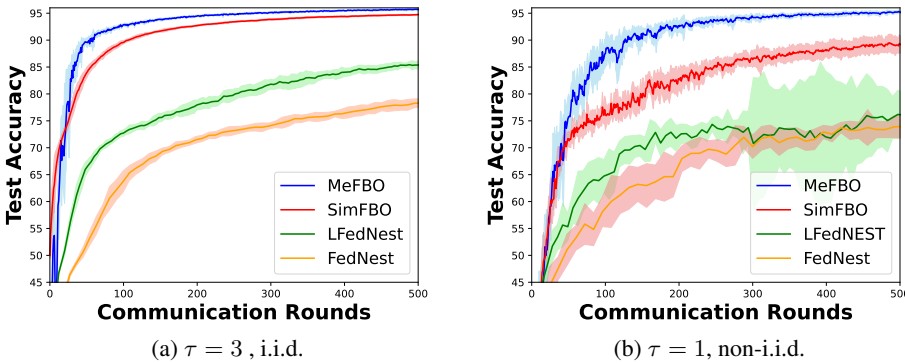

Figure 2: Comparison of the results about test accuracy v.s. communication rounds for federated loss function tuning on imbalanced dataset.

## 4.3 FEDERATED DATA HYPER-CLEANING.

In this task, each distributed client is given a noisy training dataset, where the labels are corrupted by noise at a corruption rate $cr$, along with a clean validation set. The goal is to learn weights for the training samples such that a model trained on the weighted training set performs well on the validation set. The specific formulation and experimental details, which differ slightly from the model in Li et al. (2023), are provided in Appendix B.3.

**Results.** In Figure 3, we compare the performance of different methods with $cr = 0.7$ in i.i.d. setting and $cr = 0.9$ in non-i.i.d. settings. This experimental design serves to showcase the performance of

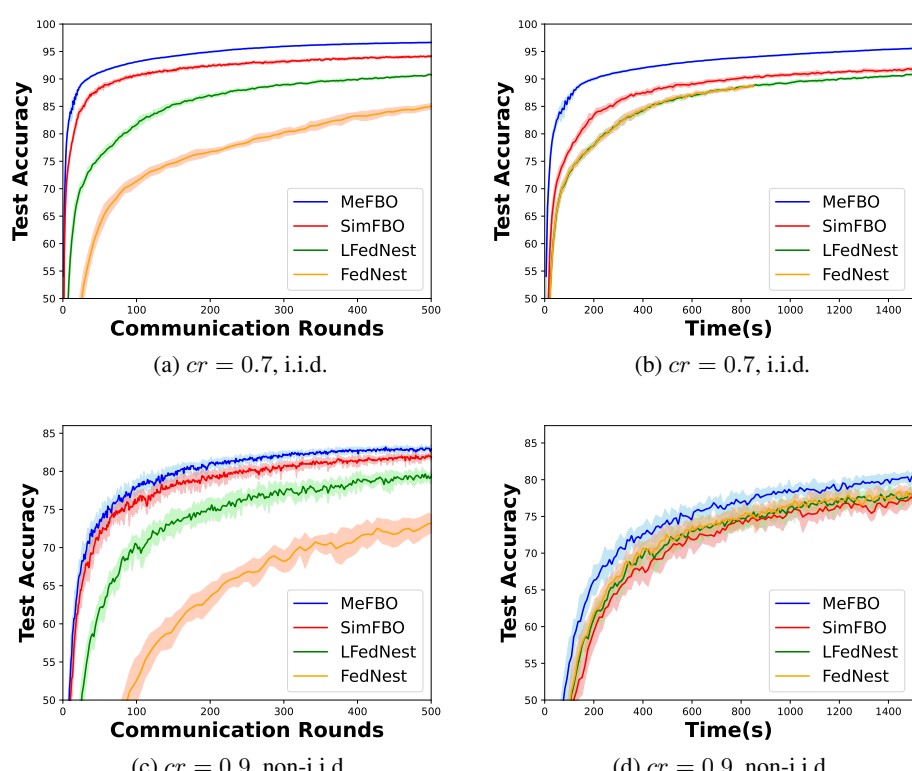

Figure 3: Comparison of the results about test accuracy v.s. communication rounds for federated data hyper-cleaning.

different algorithms under varying degrees of data heterogeneity and noise levels. The results show that MeFBO achieves better performance in terms of test accuracy, more comprehensive experimental results are presented in Figures 10 and 11 in Appendix A.3 .

Since an $rc$ regularization technique is commonly used to enforce strong convexity of the lower-level problem, we also test the sensitivity of MeFBO and the baselines to the regularization parameter $rc$. The results are summarized in Figures 12 and 13 in Appendix A.3, indicating that MeFBO is robust to the choice of $rc$.

## 5 CONCLUSION

This paper investigates federated bilevel optimization problems with non-convex lower-level objectives and introduces MeFBO, a novel, flexible, fully gradient-based algorithm. We provide a rigorous convergence analysis and complexity analysis for our method with both fixed and decreasing step sizes. Our results demonstrate that MeFBO achieves linear speedup with respect to the number of clients in federated bilevel optimization, even in the absence of convexity in the lower-level objectives. Experiments highlight the advantages of our proposed algorithms, particularly in scenarios involving non-convex lower-level objectives.

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

# Appendix

## A SUPPLEMENTARY EXPERIMENTS

### A.1 FEDERATED HYPER-REPRESENTATION

**Robustness to $rc$.** We illustrate the performance of federated hyper-representation under varying regularization coefficients $rc$ of lower-level objectives in Figure 4, and compare the convergence behaviors of our MeFBO, SimFBO, FedNest and LFedNest in hyper-representation under different regularization settings in Figure 5. From Figure 4, we observe the following:

- SimFBO algorithm shows superior results at $rc$ values of 0, 0.001, 0.005 and 0.01 (as seen in Figure 5), but it lacks theoretical guarantees and exhibits significant instability at higher $rc$ values (0.06 and 0.07).

- Our proposed MeFBO algorithm performs comparably to SimFBO in most cases, potentially offering greater stability at higher $rc$ values.

- Notably, MeFBO demonstrates superior robustness to the choice of the regularization parameter $rc$, maintaining consistent performance across a wider range of $rc$ values compared to other algorithms.

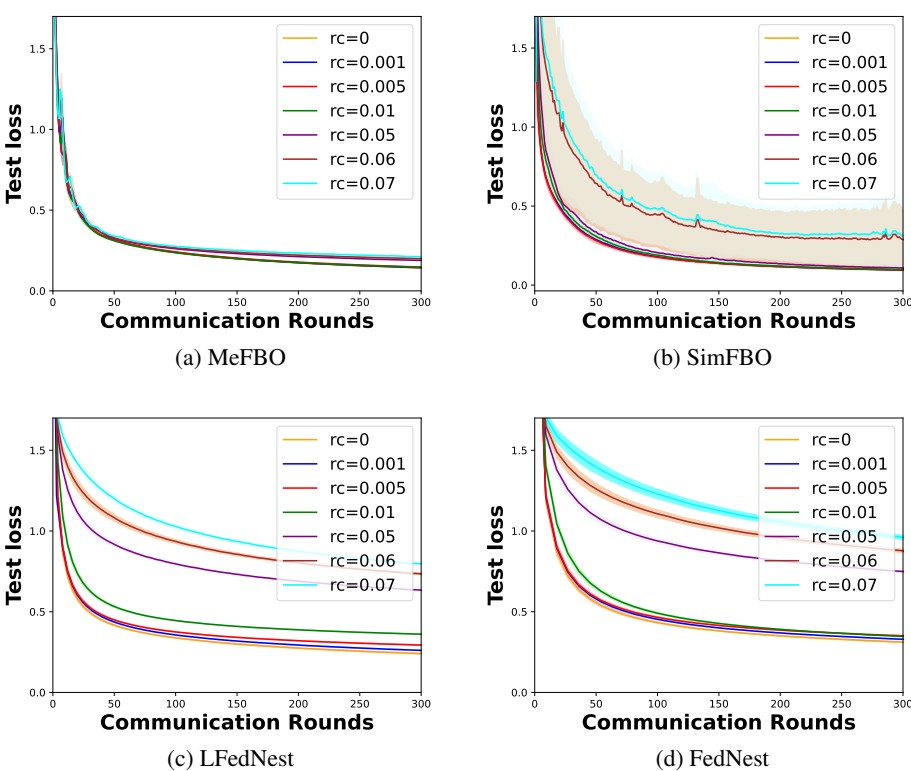

Figure 4: Federated hyper-representation under varying regularization coefficients $rc$ of lower-level objectives.

### A.2 FEDERATED LOSS FUNCTION TUNING ON IMBALANCED DATASET

**Result with the default setting of $rc = 0$.** Figures 6 and 7 illustrate the performance of various algorithms under different local round settings ($\tau = 1$ and $\tau = 3$) in i.i.d. and non-i.i.d. settings. We can draw the following key observations:

- MeFBO consistently outperforms other algorithms across all metrics (accuracy, robustness, and efficiency), particularly in heterogeneous environments.

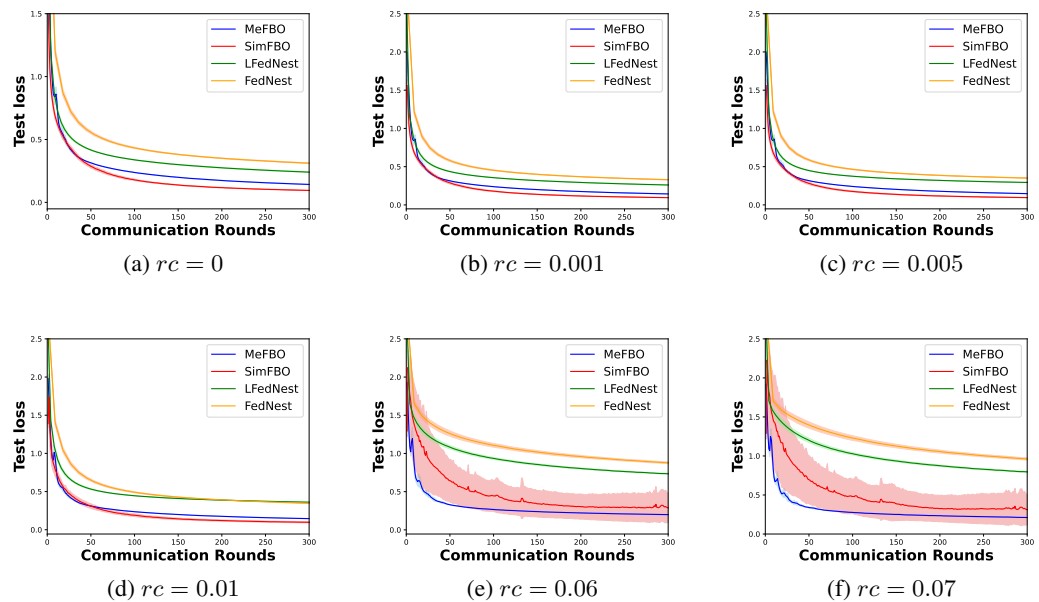

Figure 5: Comparison among our MeFBO, SimFBO, FedNest and LFedNest in federated hyper-representation under varying regularization coefficients $rc$ of lower-level objectives.

- Statistical heterogeneity from both non-i.i.d. and imbalanced datasets may have a smaller impact on fully first-order algorithms like MeFBO.

**Robustness to $rc$.** We illustrate the performance of federated loss function tuning on imbalanced datasets under varying regularization coefficients in Figure 8, and compare the convergence behaviors of MeFBO, SimFBO, FedNest and LFedNest for federated loss function tuning on imbalanced datasets across different regularization settings in Figure 9. We can draw the following key observations:

- MeFBO demonstrates superior performance in both test accuracy and loss metrics across diverse settings, exhibiting enhanced robustness particularly when $rc$ values are high.

- The relative resilience of MeFBO to excessively large $rc$ values highlights its robustness in extreme regularization conditions. This characteristic reinforces potential of MeFB0 as a preferred method for challenging data hyper-cleaning tasks in federated settings, outperforming other algorithms in both i.i.d. and non-i.i.d. environments.

## A.3 FEDERATED DATA HYPER-CLEANING

**Result with the default setting of $rc = 0$.** Figures 6 and 7 illustrate the performance of various algorithms under label corruption rate $cr$ ($cr = 0.7$ and $cr = 0.9$) in i.i.d. and non-i.i.d. settings. These figures demonstrate that our proposed algorithm, MeFBO, outperforms other methods in both i.i.d. and non-i.i.d. settings, achieving superior results within the same number of communication rounds or time frame.

**Robustness to $rc$.** We illustrate the performance of federated data hyper-cleaning under varying regularization coefficients in Figure 12, and compare the convergence behaviors of MeFBO, SimFBO, FedNest and LFedNest for federated loss function tuning on imbalanced datasets across different regularization settings in Figure 13. We can draw the following key observations:

- MeFBO demonstrates superior performance in both test accuracy and loss metrics across diverse settings, exhibiting enhanced robustness particularly when $rc$ values are high. This consistent superiority underscores the efficacy and stability of MeFBO in various federated learning scenarios.

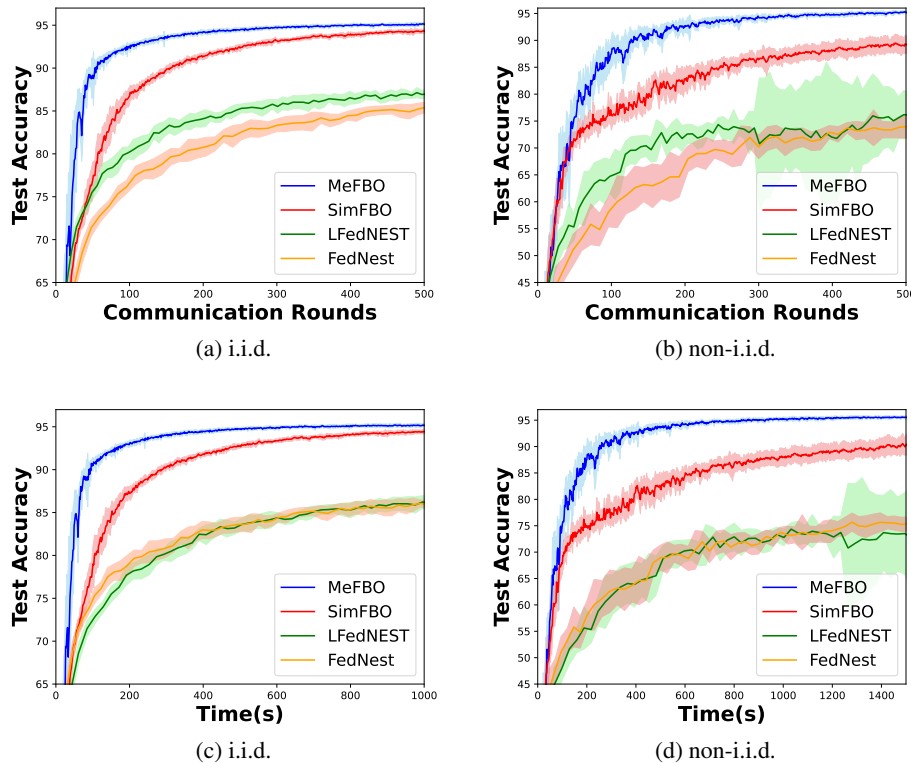

Figure 6: Federated loss function tuning on imbalanced dataset with local update round $\tau = 1$.

- The relative resilience of MeFBO to excessively large $rc$ values highlights its robustness in extreme regularization conditions. This characteristic reinforces MeFBO's potential as a preferred method for challenging data hyper-cleaning tasks in federated settings, outperforming other algorithms in both i.i.d. and non-i.i.d. environments.

## B  DETAILS OF EXPERIMENTS

In this section, we present the specific configurations used in the experiments outlined in Section 4. For the federated bilevel learning experiments, we designate the number of workers as $n = 100$, and each local network is structured as a 2 or 3-layer multilayer perceptron with a hidden dimension of 200 on a MNIST dataset. The hyperparameters are determined through a grid search, taking into account both the convergence speed and algorithm stability, and we provide a detailed report of these settings. For the baseline methods FedNest and LFedNest, we use their published codes in **https://github.com/ucr-optml/FedNest**. For SimFBO in federated hyper-representation, we use the source codes sent from the authors. The experiments were performed utilizing Python 3.7 on a computer equipped with an Intel(R) Xeon(R) Gold 5218R CPU @ 2.10GHz and an NVIDIA A100 GPU boasting 40GB of memory.

### B.1  FEDERATED HYPER-REPRESENTATION

In this section, we apply the MeFBO algorithm in Algorithm 1 to the task of federated hyper-representation learning with a 2-layer MLP on MNIST Dataset with i.i.d. distribution and non-i.i.d. distribution. The classic machine learning approach jointly learns a data representation and downstream header on the training dataset. In contrast, bilevel representation learning Tarzanagh et al. (2022) seeks to learn the data representation on the validation set while learning the header on the training set. This bilevel representation learning procedure can be formulated as a bilevel optimization problem. In a federated representation learning scenario involving $n = 100$ clients, the

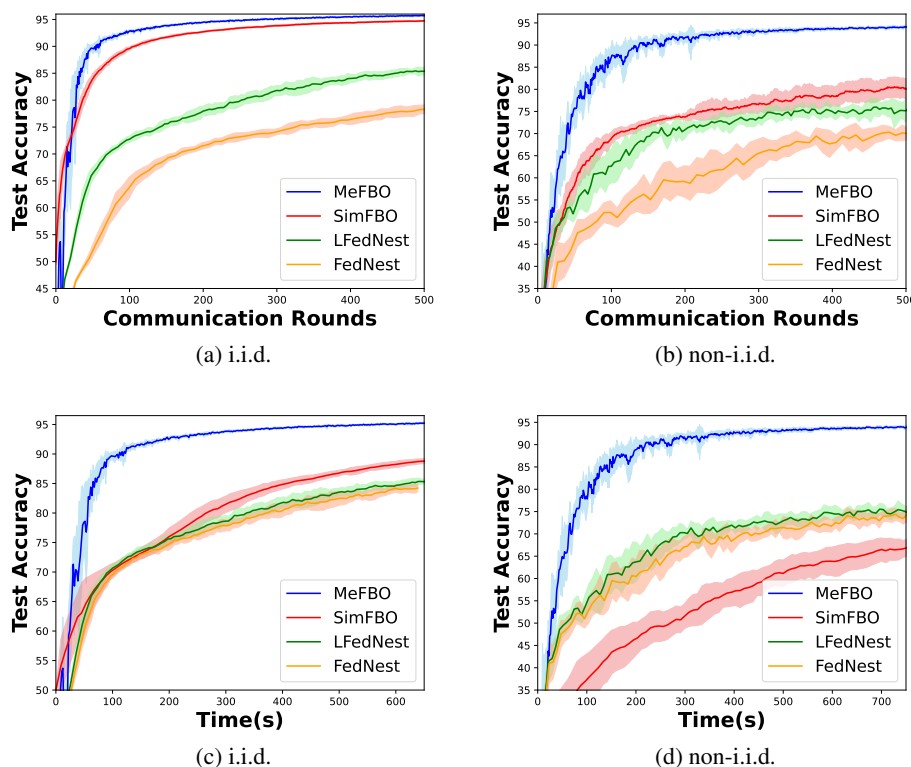

Figure 7: Federated loss function tuning on imbalanced dataset with local update round $\tau = 3$, (a), (b), (c), (d) : test accuracy vs. communication rounds.

validation and training datasets are distributed across these clients. The objective is to concurrently learn a representation and header on the combined validation and training dataset, all while ensuring the privacy of the data. Refer to Xiao et al. (2023) ,the problem can be formulated as :

$$\min_{x \in \mathcal{X}} \frac{1}{n} \sum_{i=1}^{n} f_{\mathrm{ce}}(x, y^*(x); \mathcal{D}_{\mathrm{val}}^i)$$

$$\text{s.t. } y^*(x) = \arg\min_{y \in \mathcal{Y}} \frac{1}{n} \sum_{i=1}^{n} f_{\mathrm{ce}}(x, y; \mathcal{D}_{\mathrm{tr}}^i) + rc\|y\|^2, \tag{16}$$

where $x$ is the parameters of the representation layer; $y$ is the parameter of the classifier layer; $\mathcal{D}_{\mathrm{tr}}^i$ and $\mathcal{D}_{\mathrm{val}}^i$ are, respectively, the training and validation set of client $i$. The cross-entropy loss $f_{ce}$ is defined as

$$f_{\mathrm{ce}}(x, y; \mathcal{D}) := -\frac{1}{|\mathcal{D}|} \sum_{d_n \in \mathcal{D}} \log \frac{\exp\left(h_{l_m}(x, y; d_m)\right)}{\sum_{c=1}^{C} \exp\left(h_c(x, y; d_m)\right)},$$

where $C$ is the number of classes, $d_m$ is the $m$-th data from class in dataset $\mathcal{D}$ and $h(x, y; d_m) = [h_1(x, y; d_m), ..., h_C(x, y; d_m)]^\top \in \mathbb{R}^C$ is the output of the model with parameter $(x, y)$ and input $d_m$. In Table 2, we employ a regularization coefficient (rc) value of 0.05. For the analysis presented in Figures 4 and 5, we utilize a range of $rc$ values: 0, 0.001, 0.005, 0.01, 0.05, 0.06, and 0.07.

**Hyperparameters.** For all methods, 10 clients from 100 clients are chosen randomly and participate in each communication, all algorithms are implemented with a batch size of 64. For our method MeFBO and SimFBO, we take the number of local updates, $\tau_i$, for each client $i$ to be 1, $a_i^{(t,k)}$ to be 1, and $\tilde{p}_i$ to be 0.1.For our method , MeFBO, the $c_k = 2.7(k+1)^{0.001}$ and $\gamma = 0.015$, local step sizes $[\eta_x^{(t)}, \eta_y^{(t)}, \eta_\theta^{(t)}]$ and $[\lambda_x^{(t)}, \lambda_y^{(t)}, \lambda_\theta^{(t)}]$ are both $[0.1, 0.1, 0.07]$. For SimFBO: local step sizes $[\eta_x, \eta_y, \eta_v]$ and $[\gamma_x, \gamma_y, \gamma_v]$ are both $[0.2, 0.1, 0.05]$. FedNest and LFedNest: we take the inner step size and outer

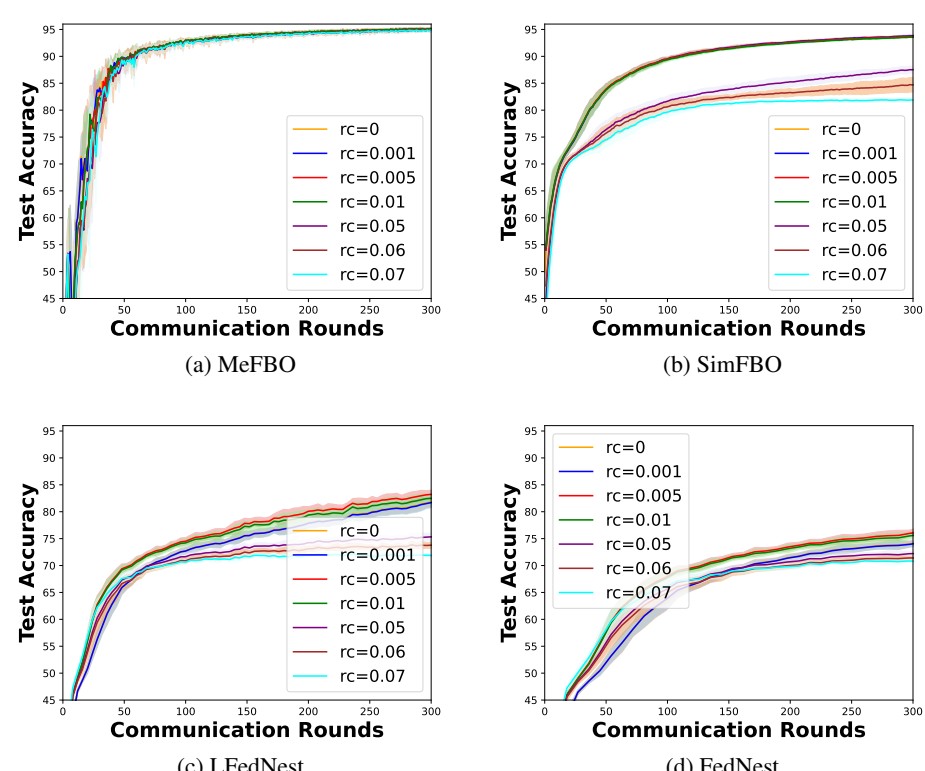

Figure 8: Federated loss function tuning on imbalanced dataset under varying regularization coefficients $rc$ of lower-level objectives in i.i.d. setting.

step size, $\alpha = 0.01, \beta = 0.02$. For the regularization coefficient case, we set $c_k = 2.7(k+1)^{0.001}$ and $\gamma = 0.015$ as fixed values for our MeFBO algorithm across various $rc$ values, as we observed their impact to be negligible. The step sizes are provided in Table 3.

|  | MeFBO | SimFBO | LFedNest | FedNest |
|---|---|---|---|---|
| rc = 0 | [0.1, 0.09, 0.05] | [0.2, 0.09, 0.04] | [0.015, 0.025] | [0.015, 0.025] |
| rc = 0.001 | [0.1, 0.09, 0.05] | [0.2, 0.09, 0.04] | [0.015, 0.025] | [0.015, 0.025] |
| rc = 0.005 | [0.1, 0.09, 0.05] | [0.2, 0.1, 0.04] | [0.015, 0.025] | [0.015, 0.025] |
| rc = 0.01 | [0.12, 0.09, 0.05] | [0.2, 0.09, 0.05] | [0.01, 0.02] | [0.01, 0.02] |
| rc = 0.05 | [0.1, 0.1, 0.07] | [0.2, 0.1, 0.05] | [0.01, 0.02] | [0.01, 0.02] |
| rc = 0.06 | [0.12, 0.12, 0.07] | [0.2, 0.12, 0.06] | [0.01, 0.015] | [0.01, 0.015] |
| rc = 0.07 | [0.13, 0.12, 0.07] | [0.2, 0.12, 0.06] | [0.01, 0.015] | [0.01, 0.015] |

Table 3: Values for the step sizes of federated hyper-representation under various $rc$. For MeFBO, the values in the table represent $[\eta_x^{(t)}/\lambda_x^{(t)}, \eta_y^{(t)}/\lambda_y^{(t)}, \eta_\theta^{(t)}/\lambda_\theta^{(t)}]$; for SimFBO, the values indicate $[\eta_x/\gamma_x, \eta_y/\gamma_y, \eta_v/\gamma_v]$. In the cases of LFedNest and FedNest, the table provides the inner and outer step sizes, denoted as $[\alpha, \beta]$.

## B.2 FEDERATED LOSS FUNCTION TUNING ON IMBALANCED DATASET

In this section, we apply the MeFBO algorithm in Algorithm 1 to the task of federated loss function tuning on imbalanced dataset with a 3-layer MLP on MNIST Dataset with i.i.d. distribution and non-i.i.d. distribution. The goal is to learn a model that ensures both fairness and generalization on datasets with under-represented classesLi et al. (2021a). In the upper-level (UL), the loss-tuning parameters are optimized to improve generalization and fairness. In the lower-level (LL), the model

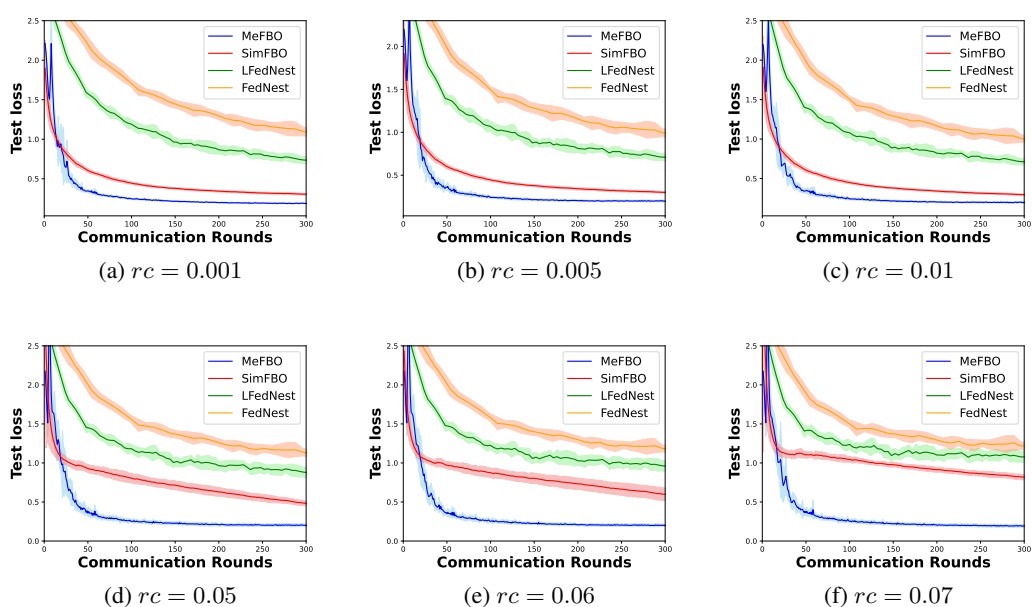

Figure 9: Comparison among our MeFBO, SimFBO, FedNest and LFedNest in federated loss function tuning on imbalanced dataset under varying regularization coefficients $rc$ of lower-level objectives.

parameters are trained on a potentially imbalanced dataset. The problem can be formulated as :

$$\min_{x \in \mathcal{X}} \frac{1}{n} \sum_{i=1}^{n} f_{\text{vs}}^{\text{up}}(y_i^*(x); \mathcal{D}_{\text{val}}^i) \qquad (17)$$

$$\text{s.t. } y^*(x) = \arg\min_{y \in \mathcal{Y}} \frac{1}{n} \sum_{i=1}^{n} f_{\text{vs}}^{\text{low}}(x, y_i; \mathcal{D}_{\text{tr}}^i) + rc\|y\|^2. \qquad (18)$$

where the number of clients is $n = 100$, $x$ is the loss-tuning parameters and $y$ is the parameter of the neural network. Here $\mathcal{D}_{\text{val}}^i$ and $\mathcal{D}_{\text{tr}}^i$ are respectively the training and validation set of client $i$. The numbers of data of different classes are imbalanced in the training data-set $\{\mathcal{D}_{\text{tr}}^i\}_{i=1}^n$. The vector-scaling loss $f_{\text{vs}}^{\text{low}}$ is defined as

$$f_{\text{vs}}^{\text{low}}(x, y; \mathcal{D}) := -\frac{1}{|\mathcal{D}|} \sum_{d_m \in \mathcal{D}} \omega_{l_m} \log \frac{\exp\left(\delta_{l_m} h_{l_m}(y; d_m) + \tau_{l_m}\right)}{\sum_{c=1}^C \exp\left(\delta_c h_c(y; d_m) + \tau_c\right)}, \qquad (19)$$

where $M$ is the data set size, $C$ is the number of classes, $d_m$ is the $m$-th data with label class $l_n$ in dataset $\mathcal{D}$ and $h(y; d_n) = [h_1(y; d_n), ..., h_C(y; d_n)]^\top \in \mathbb{R}^C$ is the logit output of the neural network with parameter $y$ and input $d_m$. Define $x = (\omega, \delta, \tau)$ where $\omega := [\omega_1, ..., \omega_C]^\top \in \mathbb{R}^C$ and $\delta, \tau$ can be defined similarly. The upper-level loss $f_{\text{vs}}^{\text{up}}$ is a special case of $f_{\text{vs}}^{\text{low}}$ with $\delta = 1, \tau = 0$ and $\omega$ is a fixed class weight vector for the validation dataset. In Figures 6 and 7, we employ a regularization coefficient (rc) value of 0. For the analysis presented in Figures 8 and 9, we utilize a range of $rc$ values: 0, 0.001, 0.005, 0.01, 0.05, 0.06, and 0.07.

**Hyperparameters.** For all methods, 10 clients from 100 clients are chosen randomly and participate in each communication. For our method MeFBO and SimFBO, we take the number of local updates $\tau = 1$ and 3, and $w_i$ to be 0.1. In the case of $\tau = 1$, for our method, MeFBO, the $c_k = 4(t + 1)^{0.1}$ and $\gamma = 0.015$, local step sizes $[\eta_x^{(t)}, \eta_y^{(t)}, \eta_\theta^{(t)}]$ and $[\lambda_x^{(t)}, \lambda_y^{(t)}, \lambda_\theta^{(t)}]$ are both $[0.1, 0.06, 0.01]$. For SimFBO: local step sizes $[\eta_x, \eta_y, \eta_v]$ and $[\gamma_x, \gamma_y, \gamma_v]$ are both $[0.08, 0.05, 0.01]$. FedNest and LFedNest: we take the best inner step size and outer step size, $\alpha = 0.02$, $\beta = 0.03$. In the case of $\tau = 3$, for our method, MeFBO, the $c_k = 4(t + 1)^{0.1}$ and $\gamma = 0.015$, local step sizes $[\eta_x^{(t)}, \eta_y^{(t)}, \eta_\theta^{(t)}]$ and $[\lambda_x^{(t)}, \lambda_y^{(t)}, \lambda_\theta^{(t)}]$ are both $[0.1, 0.06, 0.01]$. For SimFBO: local step sizes $[\eta_y, \eta_v, \eta_x]$ and $[\gamma_y, \gamma_v, \eta_x]$ are both $[0.25, 0.15, 0.03]$ in i.i.d. setting and local step sizes $[\eta_x, \eta_y, \eta_v]$

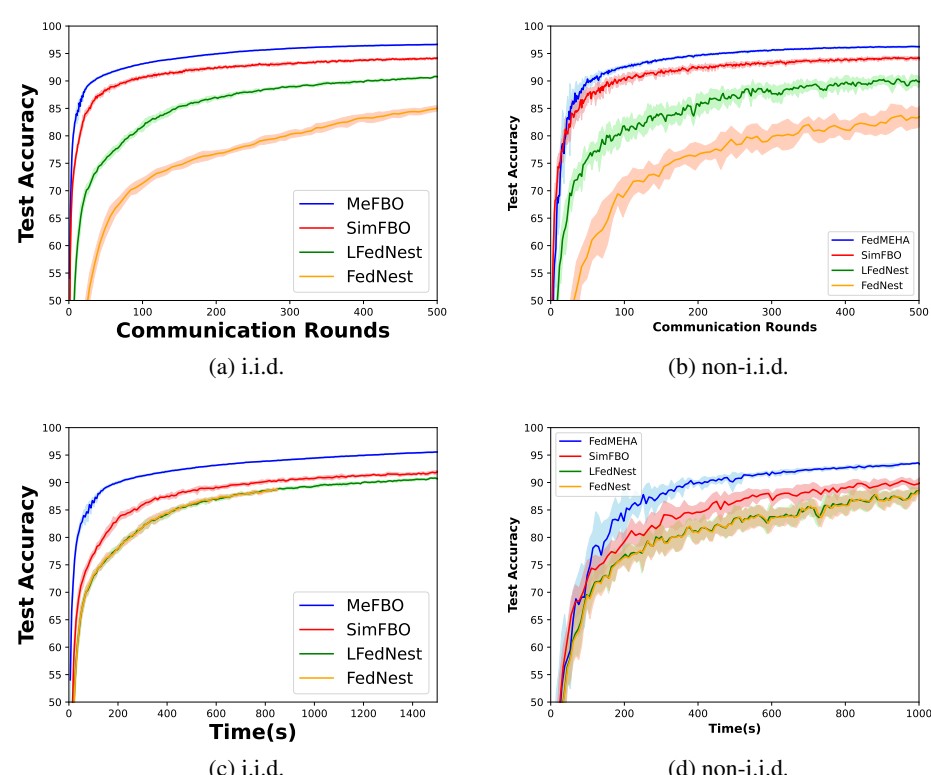

Figure 10: Federated data hyper-cleaning with a label corruption rate of $cr = 0.7$.

and $[\gamma_x, \gamma_y, \gamma_v]$ are both $[0.1, 0.08, 0.015]$ in non-i.i.d. setting . FedNest and LFedNest: we take the inner step size and outer step size $\alpha = 0.01$, $\beta = 0.02$. For the regularization coefficient case, we set $c_k = 4(t+1)^{0.1}$ and $\gamma = 0.015$ as fixed values for our MeFBO algorithm across various $rc$ values, as we observed their impact to be negligible. The step sizes are provided in Table 4:

|  | MeFBO | SimFBO | LFedNest | FedNest |
|---|---|---|---|---|
| rc = 0 | [0.1, 0.06, 0.01] | [0.25, 0.15, 0.03] | [0.01, 0.025] | [0.01, 0.02] |
| rc = 0.001 | [0.1, 0.05, 0.01] | [0.25, 0.15, 0.03] | [0.01, 0.025] | [0.01, 0.02] |
| rc = 0.005 | [0.12, 0.05, 0.01] | [0.26, 0.15, 0.03] | [0.015, 0.02] | [0.015, 0.02] |
| rc = 0.01 | [0.12, 0.05, 0.01] | [0.27, 0.16, 0.03] | [0.015, 0.02] | [0.015, 0.02] |
| rc = 0.05 | [0.15, 0.06, 0.015] | [0.26, 0.15, 0.04] | [0.015, 0.025] | [0.015, 0.025] |
| rc = 0.06 | [0.15, 0.07, 0.015] | [0.27, 0.16, 0.04] | [0.015, 0.025] | [0.015, 0.025] |
| rc = 0.07 | [0.15, 0.07, 0.015] | [0.27, 0.15, 0.05] | [0.015, 0.025] | [0.015, 0.025] |

Table 4: Values for the step sizes of federated loss function tuning on imbalanced dataset under various $rc$ with $\tau = 3$. For MeFBO, the values in the table represent $[\eta_x^{(t)}/\lambda_x^{(t)}, \eta_y^{(t)}/\lambda_y^{(t)}, \eta_\theta^{(t)}/\lambda_\theta^{(t)}]$; for SimFBO, the values indicate $[\eta_x/\gamma_x, \eta_y/\gamma_y, \eta_v/\gamma_v]$. In the case of LFedNest and FedNest, the table provides the inner and outer step sizes, denoted as $[\alpha, \beta]$.

### B.3 FEDERATED DATA HYPER-CLEANING

In this section, we apply the MeFBO algorithm (Algorithm 1) to the data hyper-cleaning task using a 2-layer MLP on the MNIST dataset with both i.i.d. and non-i.i.d. distributions. Following the approach in Tarzanagh et al. (2022), we partition the MNIST dataset into training, validation, and test sets using both i.i.d. and non-i.i.d. methods. Inspired by the work of Li et al. (2023), to mitigate issues of data quality and heterogeneity in federated learning settings, a promising approach is federated data hyper-cleaning. This technique can be formulated as a federated bilevel optimization (FBO)

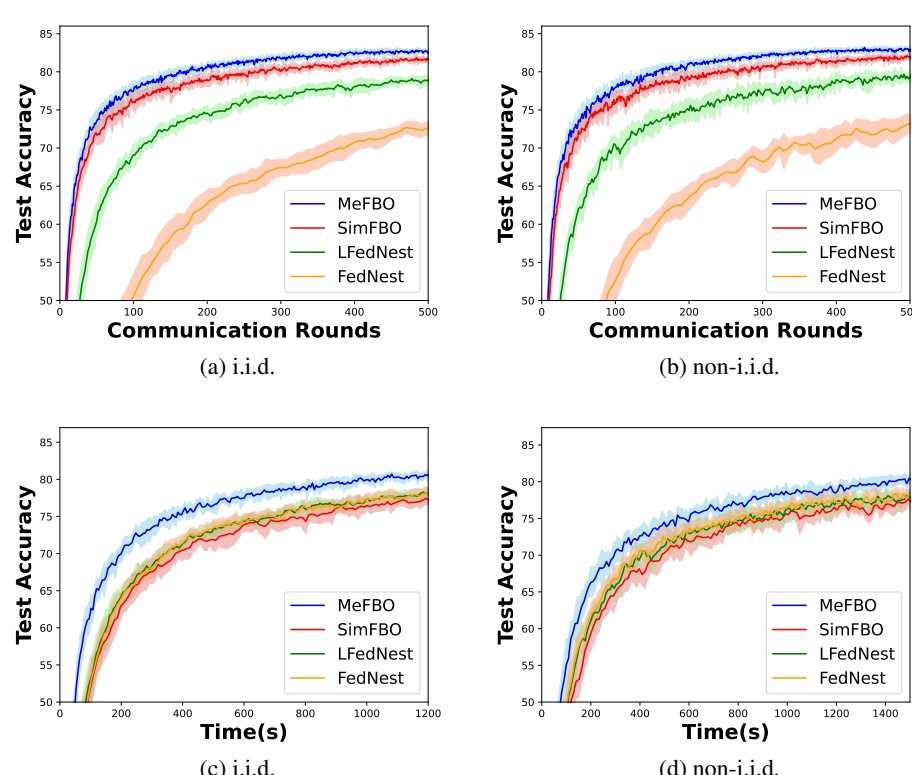

Figure 11: Federated data hyper-cleaning with a label corruption rate of $cr = 0.9$.

problem, where the upper-level objective aims to learn a globally optimal data cleaning policy, while the lower-level objectives correspond to the individual client objectives after applying the learned cleaning policy to their local datasets. Notably, the lower-level functions in this formulation exhibit non-convexity, rendering the overall problem setting more challenging than the strongly convex case. In this experiment, we are presented with a noisy training dataset whose labels are corrupted by noise with a corruption rate $cr$, along with a clean validation set. Our objective is to determine optimal weights for the training samples such that a model learned over the weighted training set exhibits superior performance on the validation set. The problem can be formulated as :

$$\min_{\psi, \boldsymbol{w}} F(\psi, \boldsymbol{w}) = \sum_{i=1}^{N} \frac{1}{|\mathcal{D}_{\text{val}}^i|} \sum_{(\mathbf{x}_j, y_j) \in \mathcal{D}_{\text{val}}^i} \mathcal{L}\big(h(\mathbf{x}_j^\top; \boldsymbol{w}), y_j\big)$$

$$\text{s.t. } \boldsymbol{w} = \arg\min_{\boldsymbol{w}'} f(\boldsymbol{\psi}, \boldsymbol{w}') = \sum_{i=1}^{N} \frac{1}{|\mathcal{D}_{\text{tr}}^i|} \sum_{(\mathbf{x}_j, y_j) \in \mathcal{D}_{\text{tr}}^i} \sigma(\psi_j) \mathcal{L}\big(h(\mathbf{x}_j^\top; \boldsymbol{w}'), y_j\big) + rc\|\boldsymbol{w}'\|^2, \tag{20}$$

where $D_{\text{tr}}^i$ and $D_{\text{val}}^i$ denote the training and validation datasets on $i^{\text{th}}$ client, respectively. $(\mathbf{x}_j, y_j)$ denote the $j^{\text{th}}$ data and label. $\sigma(\cdot)$ is the Sigmoid function, $\mathcal{L}$ is the cross-entropy loss, $N$ is the number of workers in the federated system. In Figures 10 and 11, we employ a regularization coefficient (rc) value of 0. For the analysis presented in Figures 12 and 13, we utilize a range of $rc$ values: 0, 0.001, 0.005, 0.01, 0.05, 0.06, and 0.07.

**Hyperparameters.** For all methods, 10 clients from 100 clients are chosen randomly and participate in each communication, all algorithms are implemented with a batch size of 10. For our method MeFBO and SimFBO, we take the number of local updates $\tau = 1$, and $w_i$ to be 0.1. For our method, MeFBO, the $c_k = 2(t+1)^{0.01}$ and $\gamma = 0.05$, local step sizes $[\eta_x^{(t)}, \eta_y^{(t)}, \eta_\theta^{(t)}]$ and $[\lambda_x^{(t)}, \lambda_y^{(t)}, \lambda_\theta^{(t)}]$ are both $[0.1, 0.05, 0.03]$. For SimFBO: local step sizes $[\eta_x, \eta_y, \eta_v]$ and $[\gamma_x, \gamma_y, \eta_v]$ are both $[0.08, 0.06, 0.03]$. FedNest and LFedNest: we take the inner step size and outer step size,

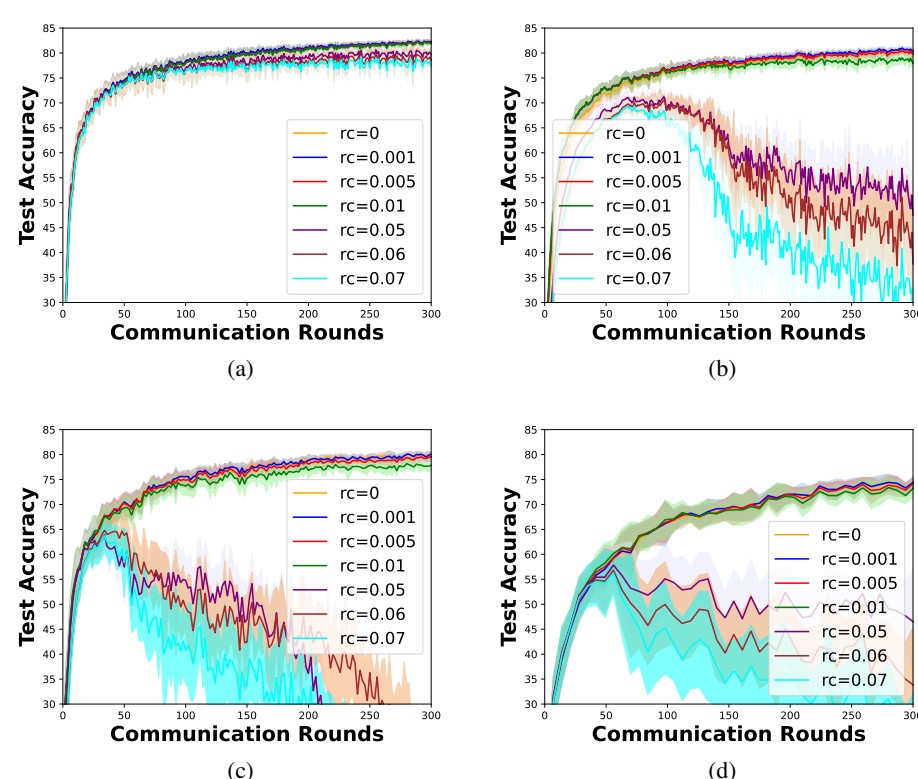

Figure 12: Federated data hyper-cleaning with a label corruption rate of $cr = 0.9$ under varying regularization coefficients $rc$ of lower-level objectives.

$\alpha = 0.1$, $\beta = 0.2$. For the regularization coefficient case, we set $c_k = 2(t+1)^{0.01}$ and $\gamma = 0.05$ as fixed values for our MeFBO algorithm across various $rc$ values, as we observed their impact to be negligible. The step sizes are provided in table 5.

|  | MeFBO | SimFBO | LFedNest | FedNest |
|---|---|---|---|---|
| rc = 0 | [0.12, 0.07, 0.03] | [0.1, 0.06, 0.03] | [0.1, 0.2] | [0.1, 0.2] |
| rc = 0.001 | [0.12, 0.07, 0.03] | [0.1, 0.07, 0.02] | [0.1, 0.2] | [0.1, 0.2] |
| rc = 0.005 | [0.1, 0.07, 0.02] | [0.1, 0.07, 0.03] | [0.2, 0.1] | [0.2, 0.1] |
| rc = 0.01 | [0.1, 0.06, 0.02] | [0.1, 0.06, 0.02] | [0.1, 0.1] | [0.1, 0.1] |
| rc = 0.05 | [0.1, 0.06, 0.02] | [0.05, 0.03, 0.01] | [0.1, 0.05] | [0.1, 0.05] |
| rc = 0.06 | [0.1, 0.05, 0.02] | [0.04, 0.03, 0.01] | [0.1, 0.05] | [0.1, 0.05] |
| rc = 0.07 | [0.1, 0.05, 0.02] | [0.04, 0.03, 0.01] | [0.1, 0.05] | [0.1, 0.05] |

Table 5: Values for the step sizes of federated data hyper-cleaning under various $rc$. For MeFBO, the values in the table represent $[\eta_x^{(t)}/\lambda_x^{(t)}, \eta_y^{(t)}/\lambda_y^{(t)}, \eta_\theta^{(t)}/\lambda_\theta^{(t)}]$; for SimFBO, the values indicate $[\eta_x/\gamma_x, \eta_y/\gamma_y, \eta_v/\gamma_v]$. In the case of LFedNest and FedNest, the table provides the inner and outer step sizes, denoted as $[\alpha, \beta]$.

## C SUPPLEMENTARY THEORETICAL RESULTS

In Theorem 3.4, we have presented the convergence results for an algorithm with a fixed step size. Below, we provide the convergence results for an algorithm with decreasing step sizes.

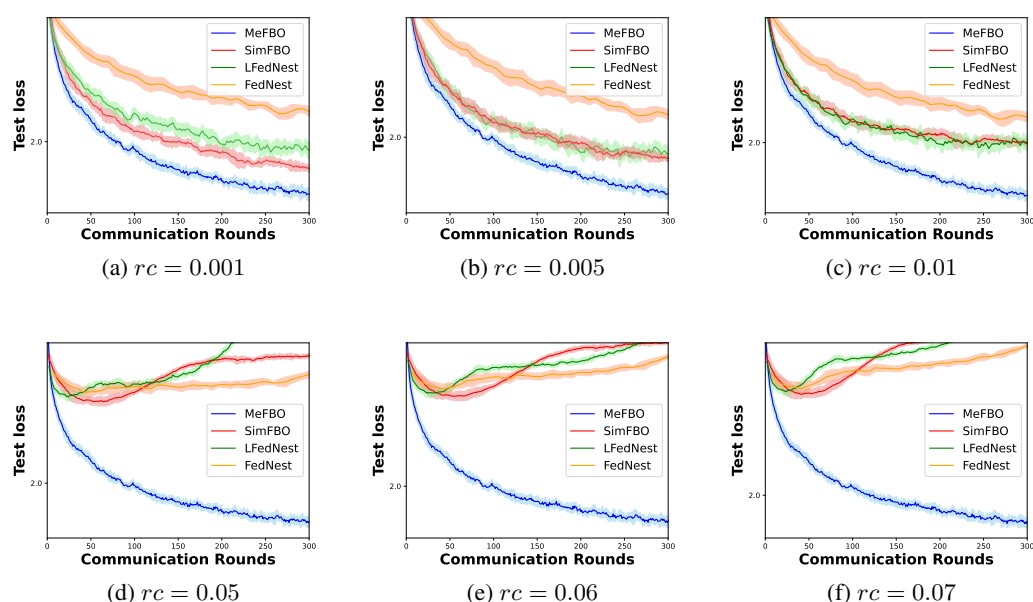

Figure 13: Comparison among our MeFBO, SimFBO, FedNest and LFedNest in federated data hyper-cleaning with a label corruption rate of $cr = 0.9$ under varying regularization coefficients $rc$ of lower-level objectives.

**Theorem C.1** (Decreasing step size). *Under Assumptions 3.1 and 3.2, we take $c_t = \underline{c}(t+1)^p$ with $\underline{c} > 0$ and $\gamma \in (0, \frac{1}{2L_2})$. Let the decreasing step sizes*

$$\lambda_\theta^{(t)} = c_\lambda \tau^{\frac{1}{4}}(t+1)^{-q}, \quad \frac{1}{2} < q < 1, \quad 0 < p < \frac{1-q}{2}, \quad \lambda_x^{(t)} = \lambda_y^{(t)} = c_\theta \lambda_\theta^{(t)},$$

$$\eta_\theta^{(t)} = \eta_x^{(t)} = \eta_y^{(t)} = c_\eta \frac{1}{\tau^{7/8}\sqrt{P}}(t+1)^{-1/4},$$

*satisfying the conditions in Lemma D.8, then the sequence of $(x^{(t)}, y^{(t)}, \theta^{(t)})$ generated by Algorithm 1 satisfies*

$$\min_{0 \le t \le T-1} \mathbb{E}\left[R_t(x^{(t+1)}, y^{(t+1)})\right] = \mathcal{O}\left(\frac{1}{P}\left(\frac{1}{T^{1-q}\tau^{1/2}} + \frac{\tau^{1/2}(n-P)}{T^{1-q}n} + \frac{1}{T^{3/2-q}\tau^{1/4}}\right)\right), \quad (21)$$

*where $c_\theta$, $c_\lambda$, and $c_\eta$ are positive constants.*

The proof of Theorem C.1 is presented in Theorem D.11 in Appendix D.4. And there are several remarks about Theorem C.1.

**Decreasing step sizes.** In contrast to fixed step sizes, the selection of decreasing step sizes can be independent of T. Moreover, the decay rate $q$ of these step sizes influences the convergence rate: a larger $q$ results in slower convergence.

**Complexity analysis.** The introduction of $q$ increases the difficulty of analyzing sample complexities or communication complexities. Yet, an appropriate trade-off in its selection can optimize both convergence rate and these complexities.

# D PROOFS

## D.1 NOTATIONS

For notational convenience, we define

$$\mathbf{F}(x, y) := \sum_{i=1}^n w_i f_i(x, y), \quad \mathbf{G}(x, y) := \sum_{i=1}^n w_i g_i(x, y), \quad (22)$$

where $X$ and $Y$ are closed convex sets in $\mathbb{R}^{d_x}$ and $\mathbb{R}^{d_y}$ , $n$ is the total number of clients, the upper- and lower-functions $f_i(x, y) = \mathbb{E}_\xi[f_i(x, y; \xi_i)]$ and $g_i(x, y) = \mathbb{E}_\zeta[g_i(x, y; \zeta_i)]$ for each client $i$ take the expectation forms w.r.t. the random variables $\xi_i$ and $\zeta_i$, and are jointly continuously differentiable.

$$\min_{x \in X, y \in Y} \mathbf{F}(x, y) := \sum_{i=1}^{n} w_i f_i(x, y)$$

$$\text{s.t.} \quad y \in S(x),$$

where $S(x)$ is the set of optimal solutions for the lower-level program:

$$\min_{y \in Y} \mathbf{G}(x, y) = \sum_{i=1}^{n} w_i g_i(x, y).$$

Similarly, we define

$$\theta_\gamma^*(x, y) = \operatorname{argmin}_{\theta \in Y} \mathbf{v}_\gamma(x, y),$$

where

$$\mathbf{v}_\gamma(x, y) := \inf_{\theta \in Y} \left\{ \mathbf{G}(x, \theta) + \frac{1}{2\gamma} \|\theta - y\|^2 \right\}. \tag{23}$$

Specifically, in each communication round $t$, each active client $i$ updates the three variables $\theta, x, y$ simultaneously during the $k$-th local iteration as

$$\begin{pmatrix} \theta_i^{(t,k+1)} \\ x_i^{(t,k+1)} \\ y_i^{(t,k+1)} \end{pmatrix} \leftarrow \begin{pmatrix} \theta_i^{(t,k)} \\ x_i^{(t,k)} \\ y_i^{(t,k)} \end{pmatrix} - \begin{pmatrix} \eta_\theta^{(t)} h_{\theta,i}^{(t,k)} \\ \eta_y^{(t)} h_{y,i}^{(t,k)} \\ \eta_x^{(t)} h_{x,i}^{(t,k)} \end{pmatrix} \tag{24}$$

where

$$h_{\theta,i}^{(t,k)} = \nabla_y g_i(x_i^{(t,k)}, \theta_i^{(t,k)}; \zeta_i^{(t,k)}) + \frac{1}{\gamma}(\theta_i^{(t,k)} - y_i^{(t,k)}),$$

$$h_{x,i}^{(t,k)} = \frac{1}{c_t} \nabla_x f_i(x_i^{(t,k)}, y_i^{(t,k)}; \xi_i^{(t,k)}) + \nabla_x g_i(x_i^{(t,k)}, y_i^{(t,k)}; \zeta_i^{(t,k)}) - \nabla_x g_i(x_i^{(t,k)}, \theta_i^{(t,k)}; \zeta_i^{(t,k)}),$$

$$h_{y,i}^{(t,k)} = \frac{1}{c_t} \nabla_y f_i(x_i^{(t,k)}, y_i^{(t,k)}) + \nabla_y g_i(x_i^{(t,k)}, y_i^{(t,k)}) - \frac{1}{\gamma}(y_i^{(t,k)} - \theta_i^{(t,k)}),$$

$$\tag{25}$$

where $\eta_\theta^{(t)}, \eta_x^{(t)}, \eta_y^{(t)}$ correspond to the local step sizes. Subsequently, we aggregate the "local gradient" of all nodes participating in the updates during round $t$:

$$h_\theta^{(t)} = \frac{n}{|C^{(t)}|} \sum_{i \in C^{(t)}} w_i h_{\theta,i}^{(t)} = \frac{n}{|C^{(t)}|} \sum_{i \in C^{(t)}} w_i \frac{1}{\tau} \sum_{k=0}^{\tau-1} h_{\theta,i}^{(t,k)},$$

$$h_x^{(t)} = \frac{n}{|C^{(t)}|} \sum_{i \in C^{(t)}} w_i h_{x,i}^{(t)} = \frac{n}{|C^{(t)}|} \sum_{i \in C^{(t)}} w_i \frac{1}{\tau} \sum_{k=0}^{\tau-1} h_{x,i}^{(t,k)}, \tag{26}$$

$$\underbrace{h_y^{(t)} = \frac{n}{|C^{(t)}|} \sum_{i \in C^{(t)}} w_i h_{y,i}^{(t)}}_{\text{Server aggregation}} = \frac{n}{|C^{(t)}|} \sum_{i \in C^{(t)}} w_i \underbrace{\frac{1}{\tau} \sum_{k=0}^{\tau-1} h_{y,i}^{(t,k)}}_{\text{Local gradients}},$$

$C^{(t)}$ means the set of participating clients in communication round $t$. We set $\frac{\beta_{\min}}{n} \leq w_i \leq \frac{\beta_{\max}}{n}$ for all $i = 1, 2, ...n$. For notational convenience, we set $\tilde{w}_i := \frac{n}{|C^{(t)}|} w_i$ And server updates $\theta^{(t+1)}, x^{(t+1)}, y^{(t+1)}$ as

$$\theta^{(t+1)} = \operatorname{Proj}_\theta \left( \theta^{(t)} - \lambda_\theta^{(t)} h_\theta^{(t)} \right),$$

$$x^{(t+1)} = \operatorname{Proj}_X \left( x^{(t)} - \lambda_x^{(t)} h_x^{(t)} \right), \tag{27}$$

$$y^{(t+1)} = \operatorname{Proj}_Y \left( y^{(t)} - \lambda_y^{(t)} h_y^{(t)} \right).$$

Then we assume $\widetilde{h}_\theta^t = \mathbb{E}[h_\theta^{(t)}], \widetilde{h}_x^t = \mathbb{E}[h_x^{(t)}], \widetilde{h}_y^t = \mathbb{E}[h_y^{(t)}]$, where $\widetilde{h}_\theta^t, \widetilde{h}_x^t, \widetilde{h}_y^t$ are defined in

$$
\begin{aligned}
\widetilde{h}_\theta^t &:= \sum_{i=0}^n w_i \left[ \widetilde{h}_{\theta,i}^t := \frac{1}{\tau} \sum_{k=0}^{\tau-1} \left[ \widetilde{h}_{\theta,i}^{(t,k)} := \mathbb{E}[h_{\theta,i}^{(t,k)}] \right] \right], \\
\widetilde{h}_x^t &:= \sum_{i=0}^n w_i \left[ \widetilde{h}_{x,i}^t := \frac{1}{\tau} \sum_{k=0}^{\tau-1} \left[ \widetilde{h}_{x,i}^{(t,k)} := \mathbb{E}[h_{x,i}^{(t,k)}] \right] \right], \\
\widetilde{h}_y^t &:= \sum_{i=0}^n w_i \left[ \widetilde{h}_{y,i}^t := \frac{1}{\tau} \sum_{k=0}^{\tau-1} \left[ \widetilde{h}_{y,i}^{(t,k)} := \mathbb{E}[h_{y,i}^{(t,k)}] \right] \right].
\end{aligned}
\tag{28}
$$

Next, we define the drifts for variables $x$, $y$, and $\theta$ across clients:

$$
\begin{aligned}
\Delta_x^{(t)} &:= \sum_{i=1}^n w_i \frac{1}{\tau} \sum_{k=0}^{\tau-1} \mathbb{E}|x_i^{(t,k)} - x^{(t)}|^2 \\
\Delta_y^{(t)} &:= \sum_{i=1}^n w_i \frac{1}{\tau} \sum_{k=0}^{\tau-1} \mathbb{E}|y_i^{(t,k)} - y^{(t)}|^2 \\
\Delta_\theta^{(t)} &:= \sum_{i=1}^n w_i \frac{1}{\tau} \sum_{k=0}^{\tau-1} \mathbb{E}|\theta_i^{(t,k)} - \theta^{(t)}|^2
\end{aligned}
\tag{29}
$$

## D.2 A UNIFIED PROOF SKETCH OF THEOREMS 3.4 AND C.1

In Section 3, Theorems 3.4 and C.1 confirm that the MeFBO algorithm is not only straightforward to implement but also guaranteed to converge theoretically. This section provides a detailed convergence analysis of the proposed federated bilevel optimization algorithm. The analysis involves two main steps: first, deriving an upper bound for the residual function, and second, obtaining precise bounds for the client drift terms. By employing Lyapunov function analysis and selecting step sizes carefully, we establish rigorous theoretical guarantees for the convergence behavior of Algorithm 1.

**Step 1. Upper bound of residual function $R_t(x^{(t)}, y^{(t)})$ with step size dependencies.**
By leveraging Assumptions 3.1 (ii), (iii), and 3.2, along with the $L$-smoothness properties of $\Phi_{c_t}$ (established in Lemma D.4), and setting the step sizes as $\eta_x^{(t)} = \eta_y^{(t)} = \eta_\theta^{(t)}$ and $\lambda_x^{(t)} = \lambda_y^{(t)} = c_\lambda \lambda_\theta^{(t)}$, where $c_\lambda$ is a positive constant, we derive the following inequality for $R_t(x^{(t)}, y^{(t)})$:

$$
\begin{aligned}
&R_t(x^{(t+1)}, y^{(t+1)}) \\
&\leq \frac{1}{\lambda_\theta^{(t)}} \Big( \underbrace{\mathcal{O}\big(\frac{1}{\lambda_\theta^{(t)}}\big)\|x^{(t+1)} - x^{(t)}\|^2 + \mathcal{O}\big(\frac{1}{\lambda_\theta^{(t)}}\big)\|y^{(t+1)} - y^{(t)}\|^2 + \mathcal{O}\big(\lambda_\theta^{(t)}\big)\|\theta^{(t)} - \theta_\gamma^*(x^{(t)}, y^{(t)})\|^2}_{S_p^{(t)}} \Big) \\
&\quad + \mathcal{O}\left( \lambda_\theta^{(t)} \tau \eta_\theta^{(t)^2} \right),
\end{aligned}
\tag{30}
$$

where $\tau$ represents the number of local update rounds. Step 1 demonstrates that $R_t(x^{(t+1)}, y^{(t+1)})$ is bounded by the terms $\lambda_\theta^{(t)^2} \tau \eta_\theta^{(t)^2}$ and $S_p^{(t)}$, where $S_p^{(t)}$ consists of three distinct components: the distances between consecutive iterations of $x$ and $y$, and the gap between the global iterate $\theta^{(t)}$ and its corresponding optimal point $\theta_\gamma^*(x^{(t)}, y^{(t)})$. Moreover, by selecting either a suitable fixed step size or a well-designed decaying step size sequence, we establish the convergence of $R_t(x^{(t)}, y^{(t)})$, provided that $S_p^{(t)}$ in (30) is appropriately bounded.

**Step 2. Bounding $S_p^{(t)}$ in (30) via Lyapunov function analysis.**
To demonstrate the descent of $S_p^{(t)}$ in (30), we introduce an appropriate Lyapunov function:

$$
\Psi_{c_t}(x^{(t)}, y^{(t)}) := \Phi_{c_t}(x^{(t)}, y^{(t)}) + K\mathbb{E}\|\theta^{(t)} - \theta_\gamma^*(x^{(t)}, y^{(t)})\|^2,
\tag{31}
$$

where $K := \frac{1}{2L_\theta}\sqrt{L_2^2 + \frac{3}{\gamma^2}}$, $L_\theta$ is a positive constant provided in Lemma D.3, and $\Phi_{c_t}$ is defined as:

$$\Phi_{c_t}(x,y) := \frac{1}{c_t}\Big(\mathbf{F}(x,y) - \underline{F}\Big) + \mathbf{G}(x,y) - \mathbf{v}_\gamma(x,y), \quad (x,y) \in X \times Y, \tag{32}$$

which ensures the non-negativity of $\Psi c_t(\cdot,\cdot)$. When the step sizes satisfy the conditions in Lemma D.8, bounds for $S_p^{(t)}$ in (30) can be established, leading to:

$$S_p^{(t)} \leq \Psi_{c_t}(x^{(t)}, y^{(t)}) - \Psi_{c_{t+1}}(x^{(t+1)}, y^{(t+1)}) + \epsilon_{\text{sto}}^{(t)} + \epsilon_{\text{dh}}^{(t)} + \epsilon_{\text{cd}}^{(t)}, \tag{33}$$

where $\epsilon_{\text{sto}}^{(t)}$, $\epsilon_{\text{dh}}^{(t)}$, and $\epsilon_{\text{cd}}^{(t)}$ are defined in Equation (85), corresponding to the variance in stochastic estimation, data heterogeneity, and client drifts, respectively. The inequality above leverages the difference in the Lyapunov function evaluated at consecutive iteration points $(x^{(t+1)}, y^{(t+1)})$ and $(x^{(t)}, y^{(t)})$.

Substituting Equation (33) into Equation (30) and summing both sides after rearrangement, and considering that $\frac{1}{c_t}$ is a decreasing sequence along with the non-negativity of $\Psi_{c_t}(\cdot,\cdot)$, we obtain

$$\sum_{t=0}^{T-1}\lambda_\theta^{(t)}R_t(x^{(t+1)}, y^{(t+1)}) \leq \mathcal{O}\left(\frac{1}{P\tau}\sum_{t=0}^{T-1}\lambda_\theta^{(t)\,2}\right) + \mathcal{O}\left(\frac{1}{P}\frac{n-P}{n-1}\sum_{t=0}^{T-1}\lambda_\theta^{(t)\,2}\right)$$

$$+ \mathcal{O}\left(\sum_{t=0}^{T-1}\lambda_\theta^{(t)\,2}\tau\eta_\theta^{(t)\,2}\right) + \Psi_{c_0}(x^{(0)}, y^{(0)}), \tag{34}$$

where $P$ is the number of clients participating in each communication round. The first, second, and third terms on the right-hand side of Equation (34) correspond to $\epsilon_{\text{sto}}^{(t)}$, $\epsilon_{\text{dh}}^{(t)}$, and $\epsilon_{\text{cd}}^{(t)}$ in Equation (33). By appropriately selecting fixed or decaying step sizes as specified in Theorem D.11, we obtain the convergence results presented in Theorems 3.4 and C.1. Notably, the first, second, and third terms in Equation (34) align with those in Equation (14) of Theorem 3.4 and Equation (21) of Theorem D.11, considering that $\Psi_{c_0}(x^{(0)}, y^{(0)})$ is a positive constant.

**Step a. Descent in $\Phi_{c_t}(x,y)$.**

$$\Phi_{c_t}(x^{(t+1)}, y^{(t+1)}) - \Phi_{c_t}(x^{(t)}, y^{(t)})$$

$$\leq -\mathcal{O}\big(\frac{1}{\lambda_x^{(t)}}\big)\mathbb{E}\|x^{(t+1)} - x^{(t)}\|^2 - \mathcal{O}\big(\frac{1}{\lambda_y^{(t)}}\big)\mathbb{E}\|y^{(t+1)} - y^{(t)}\|^2$$

$$+ \mathcal{O}\big(\lambda_x^{(t)} + \lambda_y^{(t)}\big)\mathbb{E}\|\theta^{(t)} - \theta_\gamma^*(x^{(t)}, y^{(t)})\|^2 + \mathcal{O}\big(\lambda_x^{(t)} + \lambda_y^{(t)}\big)\big(\Delta_x^{(t)} + \Delta_y^{(t)} + \Delta_\theta^{(t)}\big),$$

where $\Delta_x^{(t)}$, $\Delta_y^{(t)}$, and $\Delta_\theta^{(t)}$ arise from client drifts as defined in (85) in Appendix D.1. The proof follows a similar approach to Lemma D.5. Given the projection applied on the server side, the geometric properties of projection onto a convex set ensure that the right-hand side of the inequality remains bounded by terms involving $\mathbb{E}\|x^{(t+1)} - x^{(t)}\|^2$ and $\mathbb{E}\|y^{(t+1)} - y^{(t)}\|^2$.

**Step b. Controlling error of the distance between $\theta^{(t)}$ and $\theta_\gamma^*(x^{(t)}, y^{(t)})$ .**

$$\mathbb{E}\|\theta^{(t+1)} - \theta_\gamma^*(x^{(t+1)}, y^{(t+1)})\|^2 - \mathbb{E}\|\theta^{(t)} - \theta_\gamma^*(x^{(t)}, y^{(t)})\|^2$$

$$\leq 2L_\theta^2\big(1 + \frac{1}{\delta_{t,1}}\big)\left(\mathbb{E}\|x^{(t+1)} - x^{(t)}\|^2 + \mathbb{E}\|y^{(t+1)} - y^{(t)}\|^2\right) + (1 + \delta_{t,1})\lambda_\theta^{(t)\,2}\mathbb{E}\|h_\theta^{(t)}\|^2$$

$$+ \big((1 + \delta_{t,1})(1 - \lambda_\theta^{(t)}\rho) - 1\big)\mathbb{E}\|\theta^{(t)} - \theta_\gamma^*(x^{(t)}, y^{(t)})\|^2 + \mathcal{O}\big(\lambda_\theta^{(t)}\big)\big(1 + \delta_{t,1}\big)\big(\Delta_x^{(t)} + \Delta_y^{(t)} + \Delta_\theta^{(t)}\big).$$

where $\rho := \frac{1}{\gamma} - L_2$ and $\delta_{t,1}$ is a positive constant. The proof follows from Lemma D.6. Given the projection applied on the server side, the non-expansiveness property of projection onto a convex set ensures that the right-hand side remains bounded by $\mathbb{E}\|h_\theta^{(t)}\|^2$.

Moreover, a suitable choice of the positive constant $\delta_{t,1}$ guarantees a decreasing trend in the distance between $\theta^{(t)}$ and $\theta_\gamma^*(x^{(t)}, y^{(t)})$. Specifically, it ensures that the coefficient of $\mathbb{E}\|\theta^{(t)} - \theta_\gamma^*(x^{(t)}, y^{(t)})\|^2$

on the right-hand side is strictly negative.

Subsequently, we need to establish bounds for $\Delta_x^{(t)}$, $\Delta_y^{(t)}$, $\Delta_\theta^{(t)}$ and $\mathbb{E}\|h_\theta^{(t)}\|^2$, which can then be scaled by the corresponding step sizes.

**Step c. Bounding server stochastic gradient estimation .**

As defined in (10), the **unique** structure of the stochastic gradient $h_\theta^{(t)}$ reflects the interplay of **partial client participation**, **multiple local iterations**, and **data heterogeneity**, complicating the estimation of its bound. The following inequality captures the bound on $\mathbb{E}\|h_\theta^{(t)}\|^2$:

$$
\begin{aligned}
\mathbb{E}\|h_\theta^{(t)}\|^2 \leq &\mathcal{O}\big(\frac{\beta_{\max}}{P\tau}\big)\delta_g^2 + \mathcal{O}\big(\frac{n}{P}\frac{n-P}{n-1}\big)\big(\Delta_x^{(t)} + \Delta_y^{(t)} + \Delta_\theta^{(t)} + \mathbb{E}\|\theta^{(t)} - \theta_\gamma^*(x^{(t)}, y^{(t)})\|^2\big) \\
&+ \mathcal{O}\big(\frac{\beta_{max}}{P\tau}\frac{n-P}{n-1}\big)\big(\Delta_x^{(t)} + \Delta_y^{(t)} + \Delta_\theta^{(t)} + \Delta^2 + \mathbb{E}\|\theta^{(t)} - \theta_\gamma^*(x^{(t)}, y^{(t)})\|^2\big). \quad (35)
\end{aligned}
$$

The proof is detailed in equations (58) to (63). Since the algorithm involves only a subset of clients, the analysis employs without-replacement sampling to achieve linear speedup, which introduces the term $\frac{1}{P}$ in the bound.

**Step d. Controlling the client drifts.**

In addition to the challenges posed by unique stochastic updates, partial participation, and data heterogeneity, the following complexities further complicate the analysis:

- The interdependence between $y$ and $\theta$ introduces significant challenges into the drift analysis.
- Unlike in strongly convex settings, where certain iterates are bounded (e.g., Lemmas 1 and 2 in Yang et al. (2023)), the unbounded nature of iterates in our setting complicates the control of client drifts, thereby introducing additional variability.

The client drifts $\Delta_x^{(t)}$, $\Delta_y^{(t)}$, and $\Delta_\theta^{(t)}$ are bounded as follows:

$$
\begin{aligned}
\Delta_x^{(t)} \leq &\mathcal{O}\big(\eta_x^{(t)2}\tau\big)\big(\frac{\delta_f^2}{c_t^2} + 2\delta_g^2\big) + \mathcal{O}\big(\eta_x^{(t)2}\tau\big)\big(\frac{L_f^2}{c_t^2} + 2L_g^2\big), \\
\Delta_\theta^{(t)} \leq &\mathcal{O}\big(\tau\eta_\theta^{(t)2}\big)\delta_g^2 + \mathcal{O}\big(\tau\eta_\theta^{(t)2}\big)\Delta_x^{(t)} + \mathcal{O}\big(\tau\eta_\theta^{(t)2}\big)\Delta_\theta^{(t)} + \mathcal{O}\big(\tau\eta_\theta^{(t)2}\big)\Delta_{(y)}^t + \mathcal{O}\big(\tau\eta_\theta^{(t)2}\big)\Delta^2 \\
&+ \mathcal{O}\big(\tau\eta_\theta^{(t)2}\big)\mathbb{E}\|\theta^{(t)} - \theta_\gamma^*(x^{(t)}, y^{(t)})\|^2, \\
\Delta_y^{(t)} \leq &\mathcal{O}\big(\tau\eta_y^{(t)2}\big)\big(\frac{\delta_f^2}{c_t^2} + \delta_g^2\big) + \mathcal{O}\big(\tau\eta_y^{(t)2}\big)\big(\frac{L_f^2}{c_t^2} + L_g^2\big) + \mathcal{O}\big(\tau\eta_y^{(t)2}\big)\Delta_\theta^t + \mathcal{O}\big(\tau\eta_y^{(t)2}\big)\Delta_y^t \\
&+ \mathcal{O}\big(\tau\eta_y^{(t)2}\big)\mathbb{E}\|\theta^{(t)} - \theta_\gamma^*(x^{(t)}, y^{(t)})\|^2.
\end{aligned}
$$

The proof is provided in the appendix, specifically in the proof of Lemma D.7. The client drifts are bounded by the variances of the stochastic gradient estimators, the data heterogeneity measure $\Delta^2$ as defined in Assumption 3.2(iii), the client drifts themselves, and the distances of global iterates $\theta^{(t)}$ to their optimal solutions at each iteration $t$.

All these terms can be controlled by appropriately adjusting the local step sizes $\eta_x$, $\eta_y$, and $\eta_\theta$. By carefully tuning these step sizes, the impact of client drifts on the convergence analysis can be mitigated, ensuring improved stability and convergence.

**Step e. Deriving inequality (33) through step size adjustment.**

By combining steps a, b, c, and d, we ensure that the conditions in Lemma D.8 are satisfied through appropriate adjustment of the step sizes. This guarantees that the coefficients of $\mathbb{E}\|x^{(t+1)} - x^{(t)}\|^2$, $\mathbb{E}\|\theta^{(t)} - \theta_\gamma^*(x^{(t)}, y^{(t)})\|^2$, and $\mathbb{E}\|\theta^{(t)} - \theta_\gamma^*(x^{(t)}, y^{(t)})\|^2$ are strictly negative. Consequently, the inequality (33) is achieved.

*Remark* D.1. To simplify the notation, the heterogeneity level $\Delta$ was excluded in the convergence result of (14). Below, we present the modified convergence results that explicitly incorporate the

heterogeneity level $\Delta$, replacing the formula in (14):

$$\min_{0 \le t \le T-1} \mathbb{E} R_t(x^{(t+1)}, y^{(t+1)}) = \mathcal{O}\left( \frac{1}{P} \left( \frac{1}{T^{1/2}\tau^{1/2}} + \frac{\tau^{1/2}(n-P)}{T^{1/2}n}\Delta^2 + \frac{1}{T\tau^{1/4}}M_3^{'} \right) \right), \quad (36)$$

where $M_3' := M_3'(\Delta^2)$ is a positive constant dependent on $\Delta^2$. It is important to note that $\Delta^2 = 0$ does not imply $M_3'(\Delta^2) = 0$, as the explicit form of $M_3'$ is provided in Eq. (87). The heterogeneity estimates originate from the stochastic gradient estimation in Step c and client drifts in Step d. From Eq. (36), it is evident that an increase in the heterogeneity level $\Delta$ leads to a corresponding slowdown in the convergence rate.

### D.3 PRELIMINARY LEMMAS

By Remark 3.3 and (Liu et al., 2024, Lemmas A.5 and A.6), we can easily derive the following lemma.

**Lemma D.2** (Properties of Moreau envelopeLiu et al. (2024)). *Suppose that $g_i(x,y)$ is $L_2$-smooth on $\mathbb{R}^{d_x} \times \mathbb{R}^{d_y}$. Then for $\gamma \in (0, \frac{1}{2L_2})$, $\rho_{v_1} \ge L_2$ and $\rho_{v_2} \ge \frac{1}{\gamma}$, the function $\mathbf{v}_\gamma(x,y) + \frac{\rho_{v_1}}{2}\|x\|^2 + \frac{\rho_{v_2}}{2}\|y\|^2$ is convex on $\mathbb{R}^{d_x} \times \mathbb{R}^{d_y}$. Furthermore, for $\gamma \in (0, \frac{1}{2L_2})$, $S_\gamma(x,y) = \{\theta_\gamma^*(x,y)\}$ is a singleton and $\nabla \mathbf{v}_\gamma = \left( \nabla_x \mathbf{G}(x, \theta_\gamma^*(x,y)), (y - \theta_\gamma^*(x,y))/\gamma \right)$. In addition, the following inequality holds:*

$$-\mathbf{v}_\gamma(x,y) \le -\mathbf{v}_\gamma(\bar{x}, \bar{y}) - \left\langle \nabla \mathbf{v}_\gamma(\bar{x}, \bar{y}), (x,y) - (\bar{x}, \bar{y}) \right\rangle + \frac{\rho_{v_1}}{2}\|x - \bar{x}\|^2 + \frac{\rho_{v_2}}{2}\|y - \bar{y}\|^2, \quad (37)$$

*for $(\bar{x}, \bar{y}) \in R^{d_x} \times R^{d_y}$.*

By Remark 3.3 and (Liu et al., 2024, Lemma A.9), we can easily derive the following lemma.

**Lemma D.3** (Properties of $\theta_\gamma^*(x,y)$ Liu et al. (2024)). *Let $\gamma \in (0, \frac{1}{2L_2})$. Then, there exists $L_\theta > 0$ such that for any $(x,y), (x',y') \in \mathbb{R}^{d_x} \times \mathbb{R}^{d_y}$, the following inequality holds:*

$$\|\theta_\gamma^*(x,y) - \theta_\gamma^*(x',y')\| \le L_\theta \|(x,y) - (x',y')\|. \quad (38)$$

To establish the convergence results, we introduce an auxiliary function defined as:

$$\Phi_{c_t}(x,y) := \frac{1}{c_t}\left( \mathbf{F}(x,y) - \underline{F} \right) + \mathbf{G}(x,y) - \mathbf{v}_\gamma(x,y), \quad (x,y) \in X \times Y. \quad (39)$$

Obviously, $\Phi_{c_t}$ is non-negative over $X \times Y$.

**Lemma D.4** (Properties of $\Phi_{c_t}$). *Under Assumptions 3.1 and 3.2, if $\gamma \in (0, \frac{1}{2L_2})$, then $\Phi_{c_t}$ is $L_{\Phi_t}$-smooth w.r.t. $(x,y)$, where $L_{\Phi_t} := L_1/c_t + L_2 + \max\{L_2, 1/\gamma\}$.*

*Proof.* Under Assumptions 3.1(ii) and 3.2(ii), we have

$$\mathbb{E}\Phi_{c_t}(x^{(t+1)}, y^{(t+1)}) - \mathbb{E}\Phi_{c_t}(x^{(t)}, y^{(t)})$$

$$\le \frac{1}{c_t}\left( \mathbb{E}\left\langle \nabla \mathbf{F}(x^{(t)}, y^{(t)}), (x^{(t+1)}, y^{(t+1)}) - (x^{(t)}, y^{(t)}) \right\rangle + \frac{L_f}{2}\mathbb{E}\|(x^{(t+1)}, y^{(t+1)}) - (x^{(t)}, y^{(t)})\|^2 \right)$$

$$+ \mathbb{E}\left\langle \nabla \mathbf{G}(x^{(t)}, y^{(t)}), (x^{(t+1)}, y^{(t+1)}) - (x^{(t)}, y^{(t)}) \right\rangle + \frac{L_g}{2}\mathbb{E}\|(x^{(t+1)}, y^{(t+1)}) - (x^{(t)}, y^{(t)})\|^2$$

$$- \mathbb{E}\left\langle \nabla \mathbf{v}_\gamma(x^{(t)}, y^{(t)}), (x^{(t+1)}, y^{(t+1)}) - (x^{(t)}, y^{(t)}) \right\rangle$$

$$+ \frac{\max\{L_2, 1/\gamma\}}{2}\mathbb{E}\|x^{(t+1)} - x^{(t)}\|^2 + \frac{\max\{L_2, 1/\gamma\}}{2}\mathbb{E}\|y^{(t+1)} - y^{(t)}\|^2$$

$$\le \mathbb{E}\left\langle \nabla_x \Phi_{c_t}(x^{(t)}, y^{(t)}), x^{(t+1)} - x^{(t)} \right\rangle + \mathbb{E}\left\langle \nabla_y \Phi_{c_t}(x^{(t)}, y^{(t)}), y^{(t+1)} - y^{(t)} \right\rangle$$

$$+ \frac{L_{\Phi_t}}{2}\left( \mathbb{E}\|x^{(t+1)} - x^{(t)}\|^2 + \mathbb{E}\|y^{(t+1)} - y^{(t)}\|^2 \right),$$

$$(40)$$

with $L_{\Phi_t} := L_1/c_t + L_2 + 1/\gamma$, where the first inequality comes from the Assumption (3.1) and Assumption (3.2) and Lemma D.2. $\square$

### D.4 CONVERGENCE ANALYSIS

**Lemma D.5** (Descent in $\Phi_{c_t}(x, y)$). *Under Assumptions 3.1 and 3.2, with $\gamma \in (0, \frac{1}{2L_2})$, the sequence $(x^{(t)}, y^{(t)}, \theta^{(t)})$ generated by Algorithm 1 satisfies:*

$$\Phi_{c_t}(x^{(t+1)}, y^{(t+1)})$$

$$\leq \Phi_{c_t}(x^{(t)}, y^{(t)}) - (\frac{1}{2\lambda_x^{(t)}} - \frac{L_{\Phi_t}}{2})\mathbb{E}|x^{(t+1)} - x^{(t)}|^2 - (\frac{1}{2\lambda_y^{(t)}} - \frac{L_{\Phi_t}}{2})\mathbb{E}|y^{(t+1)} - y^{(t)}|^2$$

$$+ (\lambda_x^{(t)}L_2^2 + \lambda_y^{(t)}\frac{3}{\gamma^2})\mathbb{E}\left\|\theta^{(t)} - \theta_\gamma^{(}x^{(t)}, y^{(t)})\right\|^2 + \left(3\lambda_x^{(t)}(\frac{L_1^2}{c_t^2} + 2L_2^2) + 3\lambda_y^{(t)}(\frac{L_1^2}{c_t^2} + L_2^2)\right)\Delta_x^{(t)}$$

$$+ \left(3\lambda_x^{(t)}L_2^2 + \frac{\lambda_y^{(t)}}{2}(6\frac{L_1^2}{c_t^2} + 6L_2^2 + \frac{3}{\gamma^2})\right)\Delta_y^{(t)} + \left(3\lambda_x^{(t)}L_2^2 + \frac{3}{\gamma^2}\lambda_y^{(t)}\right)\Delta_\theta^{(t)}$$

*where $\Phi_{c_t}(x, y) := \frac{1}{c_t}(\mathbf{F}(x, y) - \underline{\mathbf{F}}) + \mathbf{G}(x, y) - \mathbf{v}_\gamma(x, y)$.*

*Proof.* By the Lemma D.4, we have

$$\Phi_{c_t}(x^{(t+1)}, y^{(t+1)})$$

$$\leq \Phi_{c_t}(x^{(t)}, y^{(t)}) + \mathbb{E}\langle \nabla_x \Phi_{c_t}(x^{(t)}, y^{(t)}), x^{(t+1)} - x^{(t)}\rangle + \mathbb{E}\langle \nabla_y \Phi_{c_t}(x^{(t)}, y^{(t)}), y^{(t+1)} - y^{(t)}\rangle$$

$$+ \frac{L_{\Phi_t}}{2}(\mathbb{E}\|x^{(t+1)} - x^{(t)}\|^2 + \mathbb{E}\|y^{(t+1)} - y^{(t)}\|^2). \tag{41}$$

Considering the update rule for the variable $x$ as defined in (12) in server and leveraging the geometric property of the projection operator $\mathrm{Proj}_X$, it follows that

$$\langle x^{(t)} - \lambda_x^{(t)}h_x^{(t)} - x^{(t+1)}, x^{(t)} - x^{(t+1)}\rangle \leq 0, \tag{42}$$

which leading to

$$\langle h_x^{(t)}, x^{(t+1)} - x^{(t)}\rangle \leq -\frac{1}{\lambda_x^{(t)}}\|x^{(t+1)} - x^{(t)}\|^2. \tag{43}$$

Similarly, we have

$$\langle h_y^{(t)}, y^{(t+1)} - y^{(t)}\rangle \leq -\frac{1}{\lambda_y^{(t)}}\|y^{(t+1)} - y^{(t)}\|^2. \tag{44}$$

Combining these inequalities (41), (43) and (44), we have

$$\Phi_{c_t}(x^{(t+1)}, y^{(t+1)}) - \Phi_{c_t}(x^{(t)}, y^{(t)})$$

$$\leq \mathbb{E}\left\langle \nabla_x \Phi_{c_t}(x^{(t)}, y^{(t)}) - h_x^{(t)}, x^{(t+1)} - x^{(t)}\right\rangle + \mathbb{E}\left\langle \nabla_y \Phi_{c_t}(x^{(t)}, y^{(t)}) - h_y^{(t)}, y^{(t+1)} - y^{(t)}\right\rangle$$

$$- \frac{1}{\lambda_x^{(t)}}\mathbb{E}\|x^{(t+1)} - x^{(t)}\|^2 - \frac{1}{\lambda_y^{(t)}}\mathbb{E}\|y^{(t+1)} - y^{(t)}\|^2 \tag{45}$$

$$+ \frac{L_{\Phi_t}}{2}(\mathbb{E}\|x^{(t+1)} - x^{(t)}\|^2 + \mathbb{E}\|y^{(t+1)} - y^{(t)}\|^2),$$

$$\leq \mathbb{E}\left\langle \nabla_x \Phi_{c_t}(x^{(t)}, y^{(t)}) - \widetilde{h}_x^{(t)}, x^{(t+1)} - x^{(t)}\right\rangle + \mathbb{E}\left\langle \nabla_y \Phi_{c_t}(x^{(t)}, y^{(t)}) - \widetilde{h}_y^{(t)}, y^{(t+1)} - y^{(t)}\right\rangle$$

$$- \frac{1}{\lambda_x^{(t)}}\mathbb{E}\|x^{(t+1)} - x^{(t)}\|^2 - \frac{1}{\lambda_y^{(t)}}\mathbb{E}\|y^{(t+1)} - y^{(t)}\|^2$$

$$+ \frac{L_{\Phi_t}}{2}(\mathbb{E}\|x^{(t+1)} - x^{(t)}\|^2 + \mathbb{E}\|y^{(t+1)} - y^{(t)}\|^2). \tag{46}$$

For $\mathbb{E}\left\langle \nabla_x \Phi_{c_t}(x^{(t)}, y^{(t)}) - \widetilde{h}_x^{(t)}, x^{(t+1)} - x^{(t)}\right\rangle$, we have

$$\mathbb{E}\left\langle \nabla_x \Phi_{c_t}(x^{(t)}, y^{(t)}) - \widetilde{h}_x^{(t)}, x^{(t+1)} - x^{(t)}\right\rangle$$

$$\leq \frac{\lambda_x^{(t)}}{2} \left\| \nabla_x \Phi_{c_t}(x^{(t)}, y^{(t)}) - \widetilde{h}_x^{(t)} \right\|^2 + \frac{1}{2\lambda_x^{(t)}} \mathbb{E}\|x^{(t+1)} - x^{(t)}\|^2, \tag{47}$$

and for $\mathbb{E}\left\langle \nabla_y \Phi_{c_t}(x^{(t)}, y^{(t)}) - \widetilde{h}_y^{(t)}, y^{(t+1)} - y^{(t)} \right\rangle$,

$$\mathbb{E}\left\langle \nabla_y \Phi_{c_t}(x^{(t)}, y^{(t)}) - \widetilde{h}_y^{(t)}, y^{(t+1)} - y^{(t)} \right\rangle$$
$$\leq \frac{\lambda_y^{(t)}}{2} \left\| \nabla_y \Phi_{c_t}(x^{(t)}, y^{(t)}) - \widetilde{h}_y^{(t)} \right\|^2 + \frac{1}{2\lambda_y^{(t)}} \mathbb{E}\|y^{(t+1)} - y^{(t)}\|^2. \tag{48}$$

Combining with (45), we can obtain that

$$\Phi_{c_t}(x^{(t+1)}, y^{(t+1)})$$
$$\leq \Phi_{c_t}(x^{(t)}, y^{(t)}) + \frac{\lambda_x^{(t)}}{2} \left\| \nabla_x \Phi_{c_t}(x^{(t)}, y^{(t)}) - \widetilde{h}_x^{(t)} \right\|^2 + \frac{\lambda_y^{(t)}}{2} \left\| \nabla_y \Phi_{c_t}(x^{(t)}, y^{(t)}) - \widetilde{h}_y^{(t)} \right\|^2$$
$$- (\frac{1}{2\lambda_x^{(t)}} - \frac{L_{\Phi_t}}{2})\mathbb{E}\|x^{(t+1)} - x^{(t)}\|^2 - (\frac{1}{2\lambda_y^{(t)}} - \frac{L_{\Phi_t}}{2})\mathbb{E}\|y^{(t+1)} - y^{(t)}\|^2. \tag{49}$$

For the term $\left\| \nabla_x \Phi_{c_t}(x^{(t)}, y^{(t)}) - \widetilde{h}_x^{(t)} \right\|^2$ in (49), according to the definition, we have

$$\left\| \nabla_x \Phi_{c_t}(x^{(t)}, y^{(t)}) - \sum_{i=1}^n w_i \frac{1}{\tau} \sum_{k=0}^{\tau-1} \widetilde{h}_{x,i}^{(t,k)} \right\|^2$$
$$= \left\| \sum_{i=1}^n w_i \left[ \nabla_x \phi_{c_t}^i(x^{(t)}, y^{(t)})[\theta_\gamma^*(x^{(t)}, y^{(t)})] - \nabla_x \phi_{c_t}^i(x^{(t)}, y^{(t)})[\theta^{(t)}] \right. \right.$$
$$\left. \left. + \nabla_x \phi_{c_t}^i(x^{(t)}, y^{(t)})[\theta^{(t)}] - \frac{1}{\tau} \sum_{k=0}^{\tau-1} \widetilde{h}_{x,i}^{(t,k)} \right] \right\|^2$$
$$\leq 2L_2^2 \mathbb{E}\left\| \theta_\gamma^*(x^{(t)}, y^{(t)}) - \theta^{(t)} \right\|^2 + 2\left\| \sum_{i=1}^n w_i \left[ \nabla_x \phi_{c_t}^i(x^{(t)}, y^{(t)})[\theta^{(t)}] \right] - \frac{1}{\tau} \sum_{k=0}^{\tau-1} \widetilde{h}_{x,i}^{(t,k)} \right\|^2. \tag{50}$$

For the term $\left\| \sum_{i=1}^n w_i \left[ \nabla_x \phi_{c_t}^i(x^{(t)}, y^{(t)})[\theta^{(t)}] - \frac{1}{\tau} \sum_{k=0}^{\tau-1} \widetilde{h}_{x,i}^{(t,k)} \right] \right\|^2$, according to the definition, we have

$$\left\| \sum_{i=1}^n w_i \left[ \nabla_x \phi_{c_t}^i(x^{(t)}, y^{(t)})[\theta^{(t)}] - \frac{1}{\tau} \sum_{k=0}^{\tau-1} \widetilde{h}_{x,i}^{(t,k)} \right] \right\|^2$$
$$\overset{(a)}{\leq} \frac{1}{\tau} \sum_{i=1}^n w_i \sum_{k=0}^{\tau-1} \left\| \nabla_x \phi_{c_t}^i(x^{(t)}, y^{(t)})[\theta^{(t)}] - \widetilde{h}_{x,i}^{(t,k)} \right\|^2$$
$$\overset{(b)}{\leq} \frac{1}{\tau} \sum_{i=1}^n w_i \sum_{k=0}^{\tau-1} \left\| \frac{1}{c_t} \nabla_x f_i(x_i^{(t,k)}, y_i^{(t,k)}) + \nabla_x g_i(x_i^{(t,k)}, y_i^{(t,k)}) - \nabla_x g_i(x_i^{(t,k)}, \theta_i^{(t,k)}) \right.$$
$$\left. - \left( \frac{1}{c_t} \nabla_x f_i(x^{(t)}, y^{(t)}) + \nabla_x g_i(x^{(t)}, y^{(t)}) - \nabla_x g_i(x^{(t)}, \theta^{(t)}) \right) \right\|^2$$
$$\overset{(c)}{\leq} \frac{1}{\tau} \sum_{k=0}^{\tau-1} \sum_{i=1}^n w_i \left( 3(\frac{L_1^2}{c_t^2} + L_2^2)\mathbb{E}\left\| (x_i^{(t,k)}, y_i^{(t,k)}) - (x^{(t)}, y^{(t)}) \right\|^2 \right.$$
$$\left. + 3L_2^2 (\mathbb{E}\left\| x_i^{(t,k)} - x^{(t)} \right\|^2 + \mathbb{E}\left\| \theta_i^{(t,k)} - \theta^{(t)} \right\|^2) \right), \tag{51}$$

where $(a)$ comes from Jensen's inequality, $(b)$ comes from the definition in (9), $(c)$ comes from the Assumption 3.1 (ii) and Assumption 3.1 (i).

Combining the inequalities (50) and (51), we have

$$\left\|\nabla_x \Phi_{c_t}(x^{(t)}, y^{(t)}) - \widetilde{h}_x^{(t)}\right\|^2$$

$$\leq 2L_2^2 \mathbb{E}\left\|\theta_\gamma^*(x^{(t)}, y^{(t)}) - \theta^{(t)}\right\|^2 + 6\big(\frac{L_1^2}{c_t^2} + 2L_2^2\big)\Delta_x^{(t)} + 6\big(\frac{L_1^2}{c_t^2} + L_2^2\big)\Delta_y^{(t)} + 6L_2^2\Delta_\theta^{(t)}. \quad (52)$$

For the term $\left\|\nabla_y \Phi_{c_t}(x^{(t)}, y^{(t)}) - \widetilde{h}_y^{(t)}\right\|^2$, similar to $\left\|\nabla_x \Phi_{c_t}(x^{(t)}, y^{(t)}) - \widetilde{h}_x^{(t)}\right\|^2$, we have

$$\left\|\nabla_y \Phi_{c_t}(x^{(t)}, y^{(t)}) - \sum_{i=1}^n w_i \widetilde{h}_{y,i}^{(t)}\right\|^2$$

$$\leq \frac{6}{\gamma^2}\mathbb{E}\left\|\theta^{(t)} - \theta_\gamma^*(x^{(t)}, y^{(t)})\right\|^2 + 6\big(\frac{L_1^2}{c_t^2} + L_2^2\big)\Delta_x^{(t)} + \big(6\frac{L_1^2}{c_t^2} + 6L_2^2 + \frac{3}{\gamma^2}\big)\Delta_y^{(t)} + \frac{6}{\gamma^2}\Delta_\theta^{(t)}.$$

Based on the above,

$$\Phi_{c_t}(x^{(t+1)}, y^{(t+1)})$$

$$\leq \Phi_{c_t}(x^{(t)}, y^{(t)}) + \frac{\lambda_x^{(t)}}{2}\mathbb{E}\left\|\nabla_x \Phi_{c_t}(x^{(t)}, y^{(t)}) - h_x^{(t)}\right\|^2 + \frac{\lambda_y^{(t)}}{2}\mathbb{E}\left\|\nabla_y \Phi_{c_t}(x^{(t)}, y^{(t)}) - h_y^{(t)}\right\|^2$$

$$- \big(\frac{1}{2\lambda_x^{(t)}} - \frac{L_{\Phi_t}}{2}\big)\mathbb{E}\|x^{(t+1)} - x^{(t)}\|^2 - \big(\frac{1}{2\lambda_y^{(t)}} - \frac{L_{\Phi_t}}{2}\big)\mathbb{E}\|y^{(t+1)} - y^{(t)}\|^2$$

$$\leq \Phi_{c_t}(x^{(t)}, y^{(t)}) - \big(\frac{1}{2\lambda_x^{(t)}} - \frac{L_{\Phi_t}}{2}\big)\mathbb{E}\|x^{(t+1)} - x^{(t)}\|^2 - \big(\frac{1}{2\lambda_y^{(t)}} - \frac{L_{\Phi_t}}{2}\big)\mathbb{E}\|y^{(t+1)} - y^{(t)}\|^2$$

$$+ \big(\lambda_x^{(t)}L_2^2 + \lambda_y^{(t)}\frac{3}{\gamma^2}\big)\mathbb{E}\left\|\theta^{(t)} - \theta_\gamma^*(x^{(t)}, y^{(t)})\right\|^2 + \Big(3\lambda_x^{(t)}\big(\frac{L_1^2}{c_t^2} + 2L_2^2\big) + 3\lambda_y^{(t)}\big(\frac{L_1^2}{c_t^2} + L_2^2\big)\Big)\Delta_x^{(t)}$$

$$+ \Big(3\lambda_x^{(t)}L_2^2 + \frac{\lambda_y^{(t)}}{2}\big(6\frac{L_1^2}{c_t^2} + 6L_2^2 + \frac{3}{\gamma^2}\big)\Big)\Delta_y^{(t)} + \Big(3\lambda_x^{(t)}L_2^2 + \frac{3}{\gamma^2}\lambda_y^{(t)}\Big)\Delta_\theta^{(t)}.$$

$$\square$$

**Lemma D.6.** *Fix the number of communication rounds $T$, local update rounds $\tau$, and the number $P$ of participating clients per communication round. Under the Assumption 3.1 and 3.2, the iterates of $\theta$ generated by Algorithm 1 satisfy*

$$\mathbb{E}\left\|\theta^{(t+1)} - \theta_\gamma^*(x^{(t+1)}, y^{(t+1)})\right\|^2$$

$$\leq 2L_\theta^2\big(1 + \frac{1}{\delta_{t,1}}\big)\mathbb{E}\left\|(x^{(t+1)}, y^{(t+1)}) - (x^{(t)}, y^{(t)})\right\|^2 + 3(1 + \delta_{t,1})\frac{\beta_{max}}{P}\Big(\frac{n-P}{n-1}\Big)\Delta^2\lambda_\theta^{(t)2}$$

$$+ (1 + \delta_{t,1})\frac{\beta_{\max}}{P\tau}\delta_g^2\lambda_\theta^{(t)2} + (1 + \delta_{t,1})\Big(\Big(1 - \rho\lambda_\theta^{(t)} + 4(L_2^2 + \frac{1}{\gamma^2})\frac{n}{P}\Big(\frac{P-1}{n-1}\Big)\lambda_\theta^{(t)2}$$

$$+ 6(L_2^2 + \frac{1}{\gamma^2})\frac{\beta_{max}}{P}\Big(\frac{n-P}{n-1}\Big)\lambda_\theta^{(t)2}\Big)\mathbb{E}\left\|\theta^{(t)} - \theta_\gamma^*(x^{(t)}, y^{(t)})\right\|^2$$

$$\Big(6L_2^2\frac{n}{P}\Big(\frac{P-1}{n-1}\Big)\lambda_\theta^{(t)2} + 9L_2^2\frac{\beta_{max}}{P}\Big(\frac{n-P}{n-1}\Big)\lambda_\theta^{(t)2} + \frac{6}{\rho}L_2^2\lambda_\theta^{(t)}\Big)\Delta_x^{(t)}$$

$$\Big(6(L_2^2 + \frac{1}{\gamma^2})\frac{n}{P}\Big(\frac{P-1}{n-1}\Big)\lambda_\theta^{(t)2} + 9(L_2^2 + \frac{1}{\gamma^2})\frac{\beta_{max}}{P}\Big(\frac{n-P}{n-1}\Big)\lambda_\theta^{(t)2} + \frac{6}{\rho}(L_2^2 + \frac{1}{\gamma^2})\lambda_\theta^{(t)}\Big)\Delta_\theta^{(t)}$$

$$\Big(6\frac{1}{\gamma^2}\frac{n}{P}\Big(\frac{P-1}{n-1}\Big)\lambda_\theta^{(t)2} + 9\frac{1}{\gamma^2}\frac{\beta_{max}}{P}\Big(\frac{n-P}{n-1}\Big)\lambda_\theta^{(t)2} + \frac{6}{\rho}\frac{1}{\gamma^2}\lambda_\theta^{(t)}\Big)\Delta_y^{(t)}\Big),$$

*where $\rho := \frac{1}{\gamma} - L_2$ and $\delta_{t,1}$ is some positive constant.*

*Proof.* For the gap of $\theta$ and $\theta^*$ on server, we have

$$\mathbb{E}\left\|\theta^{(t+1)} - \theta_\gamma^*(x^{(t+1)}, y^{(t+1)})\right\|^2$$

$$=\mathbb{E}\left\|\theta^{(t+1)} - \theta_\gamma^*(x^{(t)}, y^{(t)})\right\|^2 + \mathbb{E}\left\|\theta_\gamma^*(x^{(t+1)}, y^{(t+1)}) - \theta_\gamma^*(x^{(t)}, y^{(t)})\right\|^2 \qquad (53)$$

$$+ 2\mathbb{E}\left\langle \theta^{(t+1)} - \theta_\gamma^*(x^{(t)}, y^{(t)}), \theta_\gamma^*(x^{(t+1)}, y^{(t+1)}) - \theta_\gamma^*(x^{(t)}, y^{(t)}) \right\rangle.$$

For the last term in( 53), we have

$$2\mathbb{E}\left\langle \theta^{(t+1)} - \theta_\gamma^*(x^{(t)}, y^{(t)}), \theta_\gamma^*(x^{(t+1)}, y^{(t+1)}) - \theta_\gamma^*(x^{(t)}, y^{(t)}) \right\rangle$$

$$2\mathbb{E}\|\theta^{(t+1)} - \theta_\gamma^*(x^{(t)}, y^{(t)})\|\mathbb{E}\|\theta_\gamma^*(x^{(t+1)}, y^{(t+1)}) - \theta_\gamma^*(x^{(t)}, y^{(t)})\|$$

$$\overset{(a)}{\leq}\delta_{t,1}\mathbb{E}\|\theta^{(t+1)} - \theta_\gamma^*(x^{(t)}, y^{(t)})\|^2 + \frac{2}{\delta_{t,1}}\mathbb{E}\|\theta_\gamma^*(x^{(t+1)}, y^{(t+1)}) - \theta_\gamma^*(x^{(t)}, y^{(t)})\|^2$$

$$\overset{(b)}{\leq}\delta_{t,1}\mathbb{E}\|\theta^{(t+1)} - \theta_\gamma^*(x^{(t)}, y^{(t)})\|^2 + \frac{2L_\theta^2}{\delta_{t,1}}\mathbb{E}\|(x^{(t+1)}, y^{(t+1)}) - (x^{(t)}, y^{(t)})\|^2,$$

where $(a)$ can be derived from Young's inequality, $(b)$ comes from the Lemma D.3. Then the Eq. (53) can be reformulated as

$$\mathbb{E}\left\|\theta^{(t+1)} - \theta_\gamma^*(x^{(t+1)}, y^{(t+1)})\right\|^2 \qquad (54)$$

$$\leq(1 + \delta_{t,1})\mathbb{E}\left\|\theta^{(t+1)} - \theta_\gamma^*(x^{(t)}, y^{(t)})\right\|^2 + 2L_\theta^2\left(1 + \frac{1}{\delta_{t,1}}\right)\mathbb{E}\left\|(x^{(t+1)}, y^{(t+1)}) - (x^{(t)}, y^{(t)})\right\|^2.$$

For the term $\mathbb{E}\left\|\theta^{(t+1)} - \theta_\gamma^*(x^{(t)}, y^{(t)})\right\|^2$ in Eq. (54),

$$\mathbb{E}\left\|\theta^{(t+1)} - \theta_\gamma^*(x^{(t)}, y^{(t)})\right\|^2$$

$$\overset{(a)}{\leq}\mathbb{E}\left\|\theta^{(t)} - \lambda_\theta^{(t)}h_\theta^{(t)} - \theta_\gamma^*(x^{(t)}, y^{(t)})\right\|^2 \qquad (55)$$

$$\overset{(b)}{=}\mathbb{E}\left\|\theta^{(t)} - \theta_\gamma^*(x^{(t)}, y^{(t)})\right\|^2 + \lambda_\theta^{(t)2}\mathbb{E}\left\|h_\theta^{(t)}\right\|^2 - 2\lambda_\theta^{(t)}\mathbb{E}\left\langle \theta^{(t)} - \theta_\gamma^*(x^{(t)}, y^{(t)}), h_\theta^{(t)} \right\rangle, \qquad (56)$$

where (a) comes from the non-expansive property of $\mathrm{Proj}_\theta$ with $\theta_\gamma^*(x^{(t)}, y^{(t)}) \in Y$ and (b) holds because the clients are selected without replacement.

For the term $-\mathbb{E}\left\langle \theta^{(t)} - \theta_\gamma^*(x^{(t)}, y^{(t)}), h_\theta^{(t)} \right\rangle$ in (56),

$$-\mathbb{E}\left\langle \theta^{(t)} - \theta_\gamma^*(x^{(t)}, y^{(t)}), h_\theta^{(t)} \right\rangle$$

$$\overset{(a)}{=}-\mathbb{E}\left\langle \theta^{(t)} - \theta_\gamma^*(x^{(t)}, y^{(t)}), \widetilde{h}_\theta^{(t)} \right\rangle$$

$$\overset{(b)}{=}-\mathbb{E}\left\langle \theta^{(t)} - \theta_\gamma^*(x^{(t)}, y^{(t)}), \sum_{i=1}^n w_i\frac{1}{\tau}\sum_{k=0}^{\tau-1}\widetilde{h}_{\theta,i}^{(t,k)} \right\rangle$$

$$=-\mathbb{E}\left\langle \theta^{(t)} - \theta_\gamma^*(x^{(t)}, y^{(t)}), \sum_{i=1}^n w_i\frac{1}{\tau}\sum_{k=0}^{\tau-1}h_{\theta,i}^{(t,k)} - \sum_{i=1}^n w_i[\nabla_y g_i(x^{(t)}, \theta^{(t)}) + \frac{1}{\gamma}(\theta^{(t)} - y^{(t)})] \right\rangle$$

$$- \mathbb{E} \left\langle \theta^{(t)} - \theta^*_\gamma(x^{(t)}, y^{(t)}), \sum_{i=1}^n w_i [\nabla_y g_i(x^{(t)}, \theta^{(t)}) + \frac{1}{\gamma}(\theta^{(t)} - y^{(t)})] \right.$$

$$\left. - \sum_{i=1}^n w_i [\nabla_y g_i(x^{(t)}, \theta^*_\gamma(x^{(t)}, y^{(t)})) + \frac{1}{\gamma}(\theta^*_\gamma(x^{(t)}, y^{(t)}) - y^{(t)})] \right\rangle$$

$$\overset{(c)}{\leq} \frac{1}{\rho} \sum_{i=1}^n w_i \frac{1}{\tau} \sum_{k=0}^{\tau-1} \mathbb{E} \left\| h^{(t,k)}_{\theta,i} - \nabla_y g_i(x^{(t)}, \theta^{(t)}) + \frac{1}{\gamma}(\theta^{(t)} - y^{(t)}) \right\|^2$$

$$+ \frac{\rho}{2} \sum_{i=1}^n w_i \frac{1}{\tau} \sum_{k=0}^{\tau-1} \mathbb{E} \left\| \theta^{(t)} - \theta^*_\gamma(x^{(t)}, y^{(t)}) \right\|^2 - \rho \sum_{i=1}^n w_i \frac{1}{\tau} \sum_{k=0}^{\tau-1} \mathbb{E} \left\| \theta^{(t)} - \theta^*_\gamma(x^{(t)}, y^{(t)}) \right\|^2$$

$$\overset{(d)}{\leq} \frac{3}{\rho} L_2^2 \Delta_x^{(t)} + \frac{3}{\rho} \left( L_2^2 + \frac{1}{\gamma^2} \right) \Delta_\theta^{(t)} + \frac{3}{\rho} \frac{1}{\gamma^2} \Delta_y^{(t)} - \frac{\rho}{2} \mathbb{E} \left\| \theta^{(t)} - \theta^*_\gamma(x^{(t)}, y^{(t)}) \right\|^2, \tag{57}$$

where (a), (b) come from the Eq.(28), the first two terms in (c) comes from the Young's inequality and the last term in (c) comes from the strong convexity of $\mathbf{G}(x, \theta) + \frac{1}{2\gamma}\|\theta - y\|^2$ w.r.t. $\theta$ which can be derived from the Lemma D.2, where (d) comes from the L-smoothness of $f_i(x, y)$ and $g_i(x, y)$ in Assumption 3.1 (i) and Assumption 3.2 (i).

Next, we will analyze the term $\mathbb{E} \left\| h^{(t)}_\theta \right\|^2$ in Eq.(56).

$$\mathbb{E} \left\| h^{(t)}_\theta \right\|^2 = \mathbb{E} \left\| \sum_{i \in C^{(t)}} \widetilde{w}_i h^{(t)}_{\theta,i} \right\|^2$$

$$= \mathbb{E} \left\| \sum_{i \in C^{(t)}} \widetilde{w}_i h^{(t)}_{\theta,i} - \widetilde{w}_i \widetilde{h}^{(t)}_{\theta,i} + \widetilde{w}_i \widetilde{h}^{(t)}_{\theta,i} \right\|^2$$

$$= \mathbb{E} \left\| \sum_{i \in C^{(t)}} \widetilde{w}_i \left( h^{(t)}_{\theta,i} - \widetilde{h}^{(t)}_{\theta,i} \right) \right\|^2 + \mathbb{E} \left\| \sum_{i \in C^{(t)}} \widetilde{w}_i \widetilde{h}^{(t)}_{\theta,i} \right\|^2. \tag{58}$$

For the first term in Eq.(58),

$$\mathbb{E} \left\| \sum_{i \in C^{(t)}} \widetilde{w}_i \left( h^{(t)}_{\theta,i} - \widetilde{h}^{(t)}_{\theta,i} \right) \right\|^2$$

$$\overset{(a)}{=} \mathbb{E} \left[ \sum_{i \in C^{(t)}} \widetilde{w}_i^2 \left\| h^{(t)}_{\theta,i} - \widetilde{h}^{(t)}_{\theta,i} \right\|^2 \right]$$

$$\overset{(b)}{\leq} \frac{n}{P\tau^2} \mathbb{E} \sum_{i=1}^n \sum_{k=0}^{\tau-1} w_i^2 \left\| \nabla_y g_i(x_i^{(t,k)}, \theta_i^{(t,k)}; \zeta_i^{(t,k)}) + \frac{1}{\gamma}(\theta_i^{(t,k)} - y_i^{(t,k)}) \right.$$

$$\left. - (\nabla_y g_i(x_i^{(t,k)}, \theta_i^{(t,k)}) + \frac{1}{\gamma}(\theta_i^{(t,k)} - y_i^{(t,k)})) \right\|^2$$

$$\overset{(c)}{\leq} \frac{n}{P\tau^2} \mathbb{E} \sum_{i=1}^n \sum_{k=0}^{\tau-1} w_i^2 \delta_g^2$$

$$\overset{(d)}{\leq} \frac{\beta_{\max}}{P\tau} \delta_g^2, \tag{59}$$

where (a) holds because clients are selected without replacement, (b) follows from the definition $\widetilde{w}_i = \frac{n}{P} w_i$, where (c) comes from the Assumption 3.2 (ii), where (d) comes from the inequality $w_i \leq \frac{\beta_{\max}}{n}$.

For the second term $\mathbb{E}\left\|\sum_{i \in C^{(t)}} \widetilde{w}_i \widetilde{h}_{\theta,i}^{(t)}\right\|^2$ in Eq.(58), by Equation (24) in (Yang et al. (2023)), we have that

$$\mathbb{E}\left\|\sum_{i \in C^{(t)}} \widetilde{w}_i \widetilde{h}_{\theta,i}^{(t)}\right\|^2 = \frac{n}{P}\left(\frac{P-1}{n-1}\right)\mathbb{E}\left\|\sum_{i=1}^n w_i \widetilde{h}_{\theta,i}^{(t)}\right\|^2 + \frac{n}{P}\left(\frac{n-P}{n-1}\right)\sum_{i=1}^n w_i^2 \mathbb{E}\left\|\widetilde{h}_{\theta,i}^{(t)}\right\|^2. \tag{60}$$

For the term $\mathbb{E}\left\|\sum_{i=1}^n w_i \widetilde{h}_{\theta,i}^{(t)}\right\|^2$ in Eq.(60),

$$\mathbb{E}\left\|\sum_{i=1}^n w_i \widetilde{h}_{\theta,i}^{(t)}\right\|^2$$

$$=\mathbb{E}\left\|\sum_{i=1}^n w_i \frac{1}{\tau}\sum_{k=0}^{\tau-1}\widetilde{h}_{\theta,i}^{(t,k)}\right\|^2$$

$$\overset{(a)}{\leq} 2\sum_{i=1}^n w_i \frac{1}{\tau}\sum_{k=0}^{\tau-1}\left(\mathbb{E}\left\|\widetilde{h}_{\theta,i}^{(t,k)} - \nabla_y g_i(x^{(t)},\theta^{(t)}) + \frac{1}{\gamma}(\theta^{(t)} - y^{(t)})\right\|^2\right)$$

$$+2\left(\mathbb{E}\left\|\sum_{i=1}^n w_i \nabla_y g_i(x^{(t)},\theta^{(t)}) + \frac{1}{\gamma}(\theta^{(t)} - y^{(t)})\right.\right.$$

$$\left.\left.-\left(\nabla_y g_i(x^{(t)},\theta_\gamma^*(x^{(t)},y^{(t)})) + \frac{1}{\gamma}(\theta_\gamma^*(x^{(t)},y^{(t)}) - y^{(t)})\right)\right\|^2\right)$$

$$\overset{(b)}{\leq} 6\sum_{i=1}^n w_i \frac{1}{\tau}\sum_{k=0}^{\tau-1}\left(L_2^2\mathbb{E}\left\|x_i^{(t,k)} - x^{(t)}\right\|^2 + (L_2^2 + \frac{1}{\gamma^2})\mathbb{E}\left\|\theta_i^{(t,k)} - \theta^{(t)}\right\|^2 + \frac{1}{\gamma^2}\mathbb{E}\left\|y_i^{(t,k)} - y^{(t)}\right\|^2\right)$$

$$+4(L_2^2 + \frac{1}{\gamma^2})\mathbb{E}\left\|\theta^{(t)} - \theta_\gamma^*(x^{(t)},y^{(t)})\right\|^2$$

$$\leq 6L_2^2\Delta_x^{(t)} + 6\left(L_2^2 + \frac{1}{\gamma^2}\right)\Delta_\theta^{(t)} + 6\frac{1}{\gamma^2}\Delta_y^{(t)} + 4(L_2^2 + \frac{1}{\gamma^2})\mathbb{E}\left\|\theta^{(t)} - \theta_\gamma^*(x^{(t)},y^{(t)})\right\|^2, \tag{61}$$

where (a) comes from the definition of $\theta_\gamma^*(x^{(t)},y^{(t)})$ and (b) comes from L-smoothness of $f_i(x,y)$ and $g_i(x,y)$ in Assumption 3.1 (i) and Assumption 3.2 (i).

For the term $\sum_{i=1}^n w_i^2 \left\|\widetilde{h}_{\theta,i}^{(t)}\right\|^2$ in Eq.(60),

$$\sum_{i=1}^n w_i^2 \left\|\widetilde{h}_{\theta,i}^{(t)}\right\|^2$$

$$\leq \frac{\beta_{max}}{n}\left(3\sum_{i=1}^n w_i \frac{1}{\tau}\sum_{k=0}^{\tau-1}\left\|\widetilde{h}_{\theta,i}^{(t,k)} - \mathbb{E}\left[\nabla_y g_i(x^{(t)},\theta^{(t)}) + \frac{1}{\gamma}(\theta^{(t)} - y^{(t)})\right]\right\|^2\right.$$

$$+3\sum_{i=1}^n w_i \frac{1}{\tau}\sum_{k=0}^{\tau-1}\left\|\mathbb{E}\left[\nabla_y g_i(x^{(t)},\theta^{(t)}) + \frac{1}{\gamma}(\theta^{(t)} - y^{(t)})\right]\right.$$

$$\left.-\mathbb{E}\left[\sum_{i=1}^n w_i \nabla_y g_i(x^{(t)},\theta^{(t)}) + \frac{1}{\gamma}(\theta^{(t)} - y^{(t)})\right]\right\|^2$$

$$+3\sum_{i=1}^n w_i \left\|\mathbb{E}\left[\sum_{i=1}^n w_i \nabla_y g_i(x^{(t)},\theta^{(t)}) + \frac{1}{\gamma}(\theta^{(t)} - y^{(t)})\right]\right.$$

$$\left.-\mathbb{E}\left[\sum_{i=1}^n w_i \nabla_y g_i(x^{(t)},\theta_\gamma^*(x^{(t)},y^{(t)})) + \frac{1}{\gamma}(\theta_\gamma^*(x^{(t)},y^{(t)}) - y^{(t)})\right]\right\|^2\right)$$

$$\leq \frac{\beta_{max}}{n} \left( 9L_2^2 \sum_{i=1}^n w_i \frac{1}{\tau} \sum_{k=0}^{\tau-1} \mathbb{E} \left\| x_i^{(t,k)} - x^{(t)} \right\|^2 + 9\left(L_2^2 + \frac{1}{\gamma^2}\right) \sum_{i=1}^n w_i \frac{1}{\tau} \sum_{k=0}^{\tau-1} \mathbb{E} \left\| \theta_i^{(t,k)} - \theta^{(t)} \right\|^2 \right.$$

$$\left. + 9\frac{1}{\gamma^2} \sum_{i=1}^n w_i \frac{1}{\tau} \sum_{k=0}^{\tau-1} \mathbb{E} \left\| y_i^{(t,k)} - y^{(t)} \right\|^2 + 3\Delta^2 + 6\left(L_2^2 + \frac{1}{\gamma^2}\right) \mathbb{E} \left\| \theta^{(t)} - \theta_\gamma^*(x^{(t)}, y^{(t)}) \right\|^2 \right)$$

$$\leq \frac{\beta_{max}}{n} \left( 9L_2^2 \Delta_x^{(t)} + 9\left(L_2^2 + \frac{1}{\gamma^2}\right)\Delta_\theta^{(t)} + 9\frac{1}{\gamma^2}\Delta_y^{(t)} + 3\Delta^2 \right.$$

$$\left. + 6\left(L_2^2 + \frac{1}{\gamma^2}\right) \mathbb{E} \left\| \theta^{(t)} - \theta_\gamma^*(x^{(t)}, y^{(t)}) \right\|^2 \right). \tag{62}$$

Substitute Eqs.(59), (60), (61) and (62)into Eq.(58),

$$\mathbb{E} \left\| h_\theta^{(t)} \right\|^2$$

$$= \mathbb{E} \left\| \sum_{i \in C^{(t)}} \widetilde{w}_i \left( h_{\theta,i}^{(t)} - \widetilde{h}_{\theta,i}^{(t)} \right) \right\|^2 + \mathbb{E} \left\| \sum_{i \in C^{(t)}} \widetilde{w}_i \widetilde{h}_{\theta,i}^{(t)} \right\|^2$$

$$\leq \frac{\beta_{\max}}{P\tau} \delta_g^2 + \mathbb{E} \left\| \sum_{i \in C^{(t)}} \widetilde{w}_i \widetilde{h}_{\theta,i}^{(t)} \right\|^2$$

$$= \frac{\beta_{\max}}{P\tau} \delta_g^2 + \frac{n}{P}\left(\frac{P-1}{n-1}\right) \mathbb{E} \left\| \sum_{i=1}^n w_i \widetilde{h}_{\theta,i}^{(t)} \right\|^2 + \frac{n}{P}\left(\frac{n-P}{n-1}\right) \sum_{i=1}^n w_i^2 \mathbb{E} \left\| \widetilde{h}_{\theta,i}^{(t)} \right\|^2$$

$$\leq \frac{\beta_{\max}}{P\tau} \delta_g^2 + \frac{n}{P}\left(\frac{P-1}{n-1}\right) \left( 6L_2^2 \Delta_x^{(t)} + 6\left(L_2^2 + \frac{1}{\gamma^2}\right)\Delta_\theta^{(t)} + 6\frac{1}{\gamma^2}\Delta_y^{(t)} \right.$$

$$\left. + 4\left(L_2^2 + \frac{1}{\gamma^2}\right)\mathbb{E} \left\| \theta^{(t)} - \theta_\gamma^*(x^{(t)}, y^{(t)}) \right\|^2 \right)$$

$$+ \frac{\beta_{max}}{P}\left(\frac{n-P}{n-1}\right) \left( 9L_2^2 \Delta_x^{(t)} + 9\left(L_2^2 + \frac{1}{\gamma^2}\right)\Delta_\theta^{(t)} + 9\frac{1}{\gamma^2}\Delta_y^{(t)} + 3\Delta^2 \right.$$

$$\left. + 6\left(L_2^2 + \frac{1}{\gamma^2}\right) \left\| \theta^{(t)} - \theta_\gamma^*(x^{(t)}, y^{(t)}) \right\|^2 \right). \tag{63}$$

Combining Eqs.(56) , (57) with Eq.(63), we have

$$\mathbb{E} \left\| \theta^{(t+1)} - \theta_\gamma^*(x^{(t)}, y^{(t)}) \right\|^2$$

$$= \mathbb{E} \left\| \theta^{(t)} - \theta_\gamma^*(x^{(t)}, y^{(t)}) \right\|^2 + \lambda_\theta^{(t)2} \mathbb{E} \left\| h_\theta^{(t)} \right\|^2 - 2\lambda_\theta^{(t)} \mathbb{E} \left\langle \theta^{(t)} - \theta_\gamma^*(x^{(t)}, y^{(t)}), h_\theta^{(t)} \right\rangle$$

$$\leq \mathbb{E} \left\| \theta^{(t)} - \theta_\gamma^*(x^{(t)}, y^{(t)}) \right\|^2 + \lambda_\theta^{(t)2} \left\{ \frac{\beta_{\max}}{P\tau}\delta_g^2 + \frac{n}{P}\left(\frac{P-1}{n-1}\right)\left( \left(6L_2^2\Delta_x^{(t)} + 6\left(L_2^2 + \frac{1}{\gamma^2}\right)\Delta_\theta^{(t)}\right.\right.\right.$$

$$\left. + 6\frac{1}{\gamma^2}\Delta_y^{(t)} + 4\left(L_2^2 + \frac{1}{\gamma^2}\right)\mathbb{E} \left\| \theta^{(t)} - \theta_\gamma^*(x^{(t)}, y^{(t)}) \right\|^2 \right) + \frac{\beta_{max}}{P}\left(\frac{n-P}{n-1}\right)\left( 9L_2^2\Delta_x^{(t)} \right.$$

$$\left.\left. + 9\left(L_2^2 + \frac{1}{\gamma^2}\right)\Delta_\theta^{(t)} + 9\frac{1}{\gamma^2}\Delta_y^{(t)} + 3\Delta^2 + 6\left(L_2^2 + \frac{1}{\gamma^2}\right)\mathbb{E} \left\| \theta^{(t)} - \theta_\gamma^*(x^{(t)}, y^{(t)}) \right\|^2 \right) \right\}$$

$$+ 2\lambda_\theta^{(t)} \left( \frac{3}{\rho}L_2^2\Delta_x^{(t)} + \frac{3}{\rho}\left(L_2^2 + \frac{1}{\gamma^2}\right)\Delta_\theta^{(t)} + \frac{3}{\rho}\frac{1}{\gamma^2}\Delta_y^{(t)} - \frac{\rho}{2}\mathbb{E} \left\| \theta^{(t)} - \theta_\gamma^*(x^{(t)}, y^{(t)}) \right\|^2 \right). \tag{64}$$

Substituting equations Eqs.(56), (57) and (64) into Eq. (53), we can obtain

$$\mathbb{E} \left\| \theta^{(t+1)} - \theta_\gamma^*(x^{(t+1)}, y^{(t+1)}) \right\|^2$$

$$\leq 2L_\theta^2\big(1+\frac{1}{\delta_{t,1}}\big)\mathbb{E}\Big\|(x^{(t+1)},y^{(t+1)})-(x^{(t)},y^{(t)})\Big\|^2 + 3(1+\delta_{t,1})\frac{\beta_{max}}{P}\left(\frac{n-P}{n-1}\right)\Delta^2\lambda_\theta^{(t)2}$$

$$+ (1+\delta_{t,1})\frac{\beta_{\max}}{P\tau}\delta_g^2\lambda_\theta^{(t)2} + (1+\delta_{t,1})\left(\Big(1-\rho\lambda_\theta^{(t)}+4(L_2^2+\frac{1}{\gamma^2})\frac{n}{P}\left(\frac{P-1}{n-1}\right)\lambda_\theta^{(t)2}\right.$$

$$+ 6(L_2^2+\frac{1}{\gamma^2})\frac{\beta_{max}}{P}\left(\frac{n-P}{n-1}\right)\lambda_\theta^{(t)2}\Big)\mathbb{E}\Big\|\theta^{(t)}-\theta_\gamma^*(x^{(t)},y^{(t)})\Big\|^2$$

$$\Big(6L_2^2\frac{n}{P}\left(\frac{P-1}{n-1}\right)\lambda_\theta^{(t)2}+9L_2^2\frac{\beta_{max}}{P}\left(\frac{n-P}{n-1}\right)\lambda_\theta^{(t)2}+\frac{6}{\rho}L_2^2\lambda_\theta^{(t)}\Big)\Delta_x^{(t)}$$

$$\Big(6(L_2^2+\frac{1}{\gamma^2})\frac{n}{P}\left(\frac{P-1}{n-1}\right)\lambda_\theta^{(t)2}+9(L_2^2+\frac{1}{\gamma^2})\frac{\beta_{max}}{P}\left(\frac{n-P}{n-1}\right)\lambda_\theta^{(t)2}+\frac{6}{\rho}(L_2^2+\frac{1}{\gamma^2})\lambda_\theta^{(t)}\Big)\Delta_\theta^{(t)}$$

$$\Big(6\frac{1}{\gamma^2}\frac{n}{P}\left(\frac{P-1}{n-1}\right)\lambda_\theta^{(t)2}+9\frac{1}{\gamma^2}\frac{\beta_{max}}{P}\left(\frac{n-P}{n-1}\right)\lambda_\theta^{(t)2}+\frac{6}{\rho}\frac{1}{\gamma^2}\lambda_\theta^{(t)}\Big)\Delta_y^{(t)}\Big),$$

where $\delta_{t,1}>0, \rho:=\frac{1}{\gamma}-\rho_{\mathbf{G}_2}$. $\qquad\square$

**Lemma D.7.** *Fix the number of communication rounds $T$, local update rounds $\tau$, and the number of participating clients $P$ per communication round. Under Assumptions 3.1 and 3.2, the client drifts $\Delta_x^{(t)}$, $\Delta_y^{(t)}$ and $\Delta_\theta^{(t)}$ defined in (85) can be bounded as follows:*

$$\Delta_x^{(t)} \leq 3\eta_x^{(t)2}\tau\big(\frac{\delta_f^2}{c_t^2}+2\delta_g^2\big)+3\eta_x^{(t)2}\tau\big(\frac{L_f^2}{c_t^2}+2L_g^2\big),$$

$$\Delta_\theta^{(t)} \leq \frac{1}{\frac{\left(1-12\tau\eta_y^{(t)2}\frac{1}{\gamma^2}\right)}{6\tau\eta_\theta^{(t)2}\frac{1}{\gamma^2}}\big(1-6\tau\eta_\theta^{(t)2}(L_2^2+\frac{1}{\gamma^2})\big)-12\tau\eta_y^{(t)2}\frac{1}{\gamma^2}}\Bigg\{\frac{\left(1-12\tau\eta_y^{(t)2}\frac{1}{\gamma^2}\right)}{6\tau\eta_\theta^{(t)2}\frac{1}{\gamma^2}}\left(\eta_\theta^{(t)2}\tau\delta_g^2\right.$$

$$+ 6\tau\eta_\theta^{(t)2}L_2^2\Delta_x^{(t)}+4\eta_\theta^{(t)2}\tau\Delta^2+4\tau\eta_\theta^{(t)2}\big(\frac{1}{\gamma^2}+L_2^2\big)\mathbb{E}\Big\|\theta^{(t)}-\theta_\gamma^*(x^{(t)},y^{(t)})\Big\|^2\Big)$$

$$+ 2\eta_y^{(t)2}\tau\big(\delta_f^2+\delta_g^2\big)+4\tau\eta_y^{(t)2}\frac{L_f^2}{c_t^2}+10\tau\eta_y^{(t)2}L_g^2$$

$$+ 12\big(\frac{1}{\gamma^2}+L_2^2\big)\eta_y^{(t)2}\tau\Big\|\theta^{(t)}-\theta_\gamma^*(x^{(t)},y^{(t)})\Big\|^2\Bigg\},$$

$$\Delta_y^{(t)} \leq \frac{1}{\big(1-12\tau\eta_y^{(t)2}\frac{1}{\gamma^2}\big)-\frac{12\tau\eta_y^{(t)2}\frac{1}{\gamma^2}}{\left(1-6\tau\eta_\theta^{(t)2}(L_2^2+\frac{1}{\gamma^2})\right)}6\tau\eta_\theta^{(t)2}\frac{1}{\gamma^2}}\Bigg\{\frac{12\tau\eta_y^{(t)2}\frac{1}{\gamma^2}}{\left(1-6\tau\eta_\theta^{(t)2}(L_2^2+\frac{1}{\gamma^2})\right)}\left(\eta_\theta^{(t)2}\tau\delta_g^2\right.$$

$$+ 6\tau\eta_\theta^{(t)2}L_2^2\Delta_x^t+4\eta_\theta^{(t)2}\tau\Delta^2+4\tau\eta_\theta^{(t)2}\big(\frac{1}{\gamma^2}+L_2^2\big)\mathbb{E}\Big\|\theta^{(t)}-\theta_\gamma^*(x^{(t)},y^{(t)})\Big\|^2\Big)$$

$$+ 2\eta_y^{(t)2}\tau\big(\delta_f^2+\delta_g^2\big)+4\tau\eta_y^{(t)2}\frac{L_f^2}{c_t^2}+10\tau\eta_y^{(t)2}L_g^2$$

$$+ 12\big(\frac{1}{\gamma^2}+L_2^2\big)\eta_y^{(t)2}\tau\Big\|\theta^{(t)}-\theta_\gamma^*(x^{(t)},y^{(t)})\Big\|^2\Bigg\}. \tag{65}$$

*Proof.* For $\Delta_x^{(t)}$,

$$\Delta_x^{(t)} = \sum_{i=1}^n w_i\frac{1}{\tau}\sum_{k=0}^{\tau-1}\mathbb{E}\|x_i^{t,k}-x^{(t)}\|^2$$

$$= \sum_{i=1}^n w_i\eta_x^{(t)2}\sum_{k=0}^{\tau-1}\mathbb{E}\Big\|\sum_{j=0}^{k-1}\widetilde{h}_{i,x}^{(t,j)}-h_{i,x}^{(t,j)}\Big\|^2+\sum_{i=1}^n w_i\eta_x^{(t)2}\sum_{k=0}^{\tau-1}\mathbb{E}\Big\|\sum_{j=0}^{k-1}\widetilde{h}_{i,x}^{(t,j)}\Big\|^2$$

$$\overset{(a)}{\leq} \eta_x^{(t)2} \frac{1}{\tau} \sum_{k=0}^{\tau-1} 3k \big(\frac{\delta_f^2}{c_t^2} + 2\delta_g^2\big) + \eta_x^{(t)2} \frac{1}{\tau} \sum_{k=0}^{\tau-1} 3k \big(\frac{L_f^2}{c_t^2} + 2L_g^2\big)$$

$$\leq 3\eta_x^{(t)2} \tau \big(\frac{\delta_f^2}{c_t^2} + 2\delta_g^2\big) + 3\eta_x^{(t)2} \tau \big(\frac{L_f^2}{c_t^2} + 2L_g^2\big),$$

where the first term in (a) comes from L-smoothness of $f_i$ and $g_i$, the second term in (a) comes from the Lipschitz continuity of $f_i$ and $g_i$.

Next, we will analyze $\Delta_y^{(t)}$ and $\Delta_\theta^{(t)}$. For $\Delta_\theta^{(t)}$,

$$\Delta_\theta^{(t)} := \sum_{i=1}^n w_i \frac{1}{\tau} \sum_{k=0}^{\tau-1} \mathbb{E}\|\theta_i^{t,k} - \theta^{(t)}\|^2$$

$$= \sum_{i=1}^n w_i \frac{\eta_\theta^{(t)2}}{\tau} \sum_{k=0}^{\tau-1} \mathbb{E}\bigg\|\sum_{j=0}^{k-1} h_{i,\theta}^{(t,j)}\bigg\|^2$$

$$= \sum_{i=1}^n w_i \eta_\theta^{(t)2} \frac{1}{\tau} \sum_{k=0}^{\tau-1} \mathbb{E}\bigg\|\sum_{j=0}^{k-1} \widetilde{h}_{i,\theta}^{(t,j)} - h_{i,\theta}^{(t,j)}\bigg\|^2 + \sum_{i=1}^n w_i \eta_\theta^{(t)2} \frac{1}{\tau} \sum_{k=0}^{\tau-1} \mathbb{E}\bigg\|\sum_{j=0}^{k-1} \widetilde{h}_{i,\theta}^{(t,j)}\bigg\|^2$$

$$\overset{(a)}{\leq} \eta_\theta^{(t)2} \tau \delta_g^2 + \sum_{i=1}^n w_i \eta_\theta^{(t)2} \frac{1}{\tau} \sum_{k=0}^{\tau-1} \mathbb{E}\bigg\|\sum_{j=0}^{k-1} \widetilde{h}_{i,\theta}^{(t,j)}\bigg\|^2, \tag{66}$$

where the first term in (a) comes from L-smoothness of $f_i$ and $g_i$. For the second term in (66),

$$\sum_{i=1}^n w_i \eta_\theta^{(t)2} \frac{1}{\tau} \sum_{k=0}^{\tau-1} \mathbb{E}\bigg\|\sum_{j=0}^{k-1} \widetilde{h}_{i,\theta}^{(t,j)}\bigg\|^2$$

$$\leq \sum_{i=1}^n w_i \frac{\eta_\theta^{(t)2}}{\tau} \sum_{k=0}^{\tau-1} \mathbb{E}\bigg\|\sum_{j=0}^{k-1} \nabla_y g_i(x_i^{(t,j)}, \theta_i^{(t,j)}) + \frac{1}{\gamma}(\theta_i^{(t,j)} - y_i^{(t,j)})\bigg\|^2$$

$$\overset{(a)}{\leq} 2\sum_{i=1}^n w_i \frac{\eta_\theta^{(t)2}}{\tau} \sum_{k=0}^{\tau-1} \sum_{j=0}^{k-1} \bigg(\mathbb{E}\bigg\|\nabla_y g_i(x_i^{(t,j)}, \theta_i^{(t,j)}) + \frac{1}{\gamma}(\theta_i^{(t,j)} - y_i^{(t,j)})$$

$$- \nabla_y g_i(x^{(t)}, \theta^{(t)}) - \frac{1}{\gamma}(\theta^{(t)} - y^{(t)})\bigg\|^2\bigg)$$

$$+ 2\sum_{i=1}^n w_i \frac{\eta_\theta^{(t)2}}{\tau} \sum_{k=0}^{\tau-1} \sum_{j=0}^{k-1} \bigg(\mathbb{E}\bigg\|\nabla_y g_i(x^{(t)}, \theta^{(t)}) + \frac{1}{\gamma}(\theta^{(t)} - y^{(t)}) - \sum_{i=1}^n \nabla_y g_i(x^{(t)}, \theta^{(t)})$$

$$- \frac{1}{\gamma}(\theta^{(t)} - y^{(t)}) + \sum_{i=1}^n \nabla_y g_i(x^{(t)}, \theta^{(t)}) + \frac{1}{\gamma}(\theta^{(t)} - y^{(t)}) - \sum_{i=1}^n \nabla_y g_i(x^{(t)}, \theta_\gamma^*(x^{(t)}, y^{(t)}))$$

$$- \frac{1}{\gamma}(\theta_\gamma^*(x^{(t)}, y^{(t)}) - y^{(t)})\bigg\|^2\bigg)$$

$$\overset{(b)}{\leq} 6L_2^2 \eta_\theta^{(t)2} \sum_{i=1}^n w_i \frac{1}{\tau} \sum_{k=0}^{\tau-1} \sum_{j=0}^{k-1} \mathbb{E}\bigg\|x_i^{(t,j)} - x^{(t)}\bigg\|^2 + 6\eta_\theta^{(t)2} \frac{1}{\gamma^2} \sum_{i=1}^n w_i \frac{1}{\tau} \sum_{k=0}^{\tau-1} \sum_{j=0}^{k-1} \mathbb{E}\bigg\|y_i^{(t,j)} - y^{(t)}\bigg\|^2$$

$$+ 6\eta_\theta^{(t)2} (L_2^2 + \frac{1}{\gamma^2}) \sum_{i=1}^n w_i \frac{1}{\tau} \sum_{k=0}^{\tau-1} \sum_{j=0}^{k-1} \mathbb{E}\bigg\|\theta_i^{(t,j)} - \theta^{(t)}\bigg\|^2 + 4\eta_\theta^{(t)2} \tau \Delta^2$$

$$+ 4\tau \eta_\theta^{(t)2} (\frac{1}{\gamma^2} + L_2^2) \mathbb{E}\bigg\|\theta^{(t)} - \theta_\gamma^*(x^{(t)}, y^{(t)})\bigg\|^2$$

$$\leq 6\tau \eta_\theta^{(t)2} L_2^2 \Delta_x^t + 6\tau \eta_\theta^{(t)2} (L_2^2 + \frac{1}{\gamma^2}) \Delta_\theta^t + 6\tau \eta_\theta^{(t)2} \frac{1}{\gamma^2} \Delta_y^t + 4\eta_\theta^{(t)2} \tau \Delta^2$$

$$+ 4\tau \eta_\theta^{(t)2} \big(\frac{1}{\gamma^2} + L_2^2\big) \mathbb{E}\Big\|\theta^{(t)} - \theta_\gamma^*(x^{(t)}, y^{(t)})\Big\|^2,$$

where (a) comes from the definition of $\theta_\gamma^*(x^{(t)}, y^{(t)})$ and (b) comes from Assumption 3.2 (iii) and L-smoothness of $f_i(x, y)$ and $g_i(x, y)$ in Assumption 3.1 (i) and Assumption 3.2 (i). Then we have

$$\Delta_\theta^t \leq \eta_\theta^{(t)2} \tau \delta_g^2 + 6\tau \eta_\theta^{(t)2} L_2^2 \Delta_x^t + 6\tau \eta_\theta^{(t)2}\big(L_2^2 + \frac{1}{\gamma^2}\big)\Delta_\theta^t + 6\tau \eta_\theta^{(t)2}\frac{1}{\gamma^2}\Delta_y^t + 4\eta_\theta^{(t)2}\tau\Delta^2$$

$$+ 4\tau \eta_\theta^{(t)2}\big(\frac{1}{\gamma^2} + L_2^2\big)\mathbb{E}\Big\|\theta^{(t)} - \theta_\gamma^*(x^{(t)}, y^{(t)})\Big\|^2. \tag{67}$$

For $\Delta_y^t$,

$$\Delta_y^t = \sum_{i=1}^{n} w_i \frac{1}{\tau} \sum_{k=0}^{\tau-1} \mathbb{E}\|y_i^{t,k} - y^{(t)}\|^2$$

$$= \sum_{i=1}^{n} w_i \frac{\eta_y^{(t)2}}{\tau} \sum_{k=0}^{\tau-1} \mathbb{E}\Big\|\sum_{j=0}^{k-1} h_{i,y}^{(t,j)}\Big\|^2$$

$$= \sum_{i=1}^{n} w_i \eta_y^{(t)2} \frac{1}{\tau} \sum_{k=0}^{\tau-1} \mathbb{E}\Big\|\sum_{j=0}^{k-1} \widetilde{h}_{i,y}^{(t,j)} - h_{i,y}^{(t,j)}\Big\|^2 + \sum_{i=1}^{n} w_i \eta_y^{(t)2} \frac{1}{\tau} \sum_{k=0}^{\tau-1} \mathbb{E}\Big\|\sum_{j=0}^{k-1} \widetilde{h}_{i,y}^{(t,j)}\Big\|^2$$

$$\overset{(a)}{\leq} 2\eta_y^{(t)2} \tau \big(\frac{\delta_f^2}{c_t^2} + \delta_g^2\big) + \sum_{i=1}^{n} w_i \eta_y^{(t)2} \frac{1}{\tau} \sum_{k=0}^{\tau-1} \mathbb{E}\Big\|\sum_{j=0}^{k-1} \widetilde{h}_{i,y}^{(t,j)}\Big\|^2, \tag{68}$$

where the first term in (a) comes from L-smoothness of $f_i$ and $g_i$. For the second term in (68),

$$\sum_{i=1}^{n} w_i \eta_y^{(t)2} \frac{1}{\tau} \sum_{k=0}^{\tau-1} \mathbb{E}\Big\|\sum_{j=0}^{k-1} \widetilde{h}_{i,y}^{(t,j)}\Big\|^2$$

$$\leq \sum_{i=1}^{n} w_i \frac{\eta_y^{(t)2}}{\tau} \sum_{k=0}^{\tau-1} \mathbb{E}\Big\|\sum_{j=0}^{k-1} \frac{1}{c_t}\nabla_y f_i(x_i^{(t,j)}, y_i^{(t,j)}) + \nabla_y g_i(x_i^{(t,j)}, y_i^{(t,j)}) - \frac{1}{\gamma}(y_i^{(t,j)} - \theta_i^{(t,j)})\Big\|^2$$

$$\overset{(a)}{\leq} 2\sum_{i=1}^{n} w_i \frac{\eta_y^{(t)2}}{\tau} \sum_{k=0}^{\tau-1}\sum_{j=0}^{k-1}\Big(\mathbb{E}\Big\|\frac{1}{c_t}\nabla_y f_i(x_i^{(t,j)}, y_i^{(t,j)}) + \nabla_y g_i(x_i^{(t,j)}, y_i^{(t,j)})\Big\|^2\Big)$$

$$+ 2\sum_{i=1}^{n} w_i \frac{\eta_y^{(t)2}}{\tau} \sum_{k=0}^{\tau-1}\sum_{j=0}^{k-1}\Big(\mathbb{E}\Big\|\frac{1}{\gamma}(\theta_i^{(t,j)} - y_i^{(t,j)}) - \frac{1}{\gamma}(\theta^{(t)} - y^{(t)}) + \frac{1}{\gamma}(\theta^{(t)} - y^{(t)})$$

$$+ \sum_{i=1}^{n} w_i \nabla_y g_i(x^{(t)}, \theta^{(t)}) - \sum_{i=1}^{n} w_i \nabla_y g_i(x^{(t)}, \theta_\gamma^*(x^{(t)}, y^{(t)})) - \frac{1}{\gamma}(\theta_\gamma^*(x^{(t)}, y^{(t)}) - y^{(t)})$$

$$- \sum_{i=1}^{n} w_i \nabla_y g_i(x^{(t)}, \theta^{(t)})\Big\|^2\Big)$$

$$\leq 2\sum_{i=1}^{n} w_i \frac{\eta_y^{(t)2}}{\tau} \sum_{k=0}^{\tau-1}\sum_{j=0}^{k-1}\Big(\mathbb{E}\Big\|\frac{1}{c_t}\nabla_y f_i(x_i^{(t,j)}, y_i^{(t,j)}) + \nabla_y g_i(x_i^{(t,j)}, y_i^{(t,j)})\Big\|^2\Big)$$

$$+ 6\sum_{i=1}^{n} w_i \frac{\eta_y^{(t)2}}{\tau} \sum_{k=0}^{\tau-1}\sum_{j=0}^{k-1}\Big(\mathbb{E}\Big\|\frac{1}{\gamma}(\theta_i^{(t,j)} - y_i^{(t,j)}) - \frac{1}{\gamma}(\theta^{(t)} - y^{(t)})\Big\|^2$$

$$+ \mathbb{E}\Big\|\sum_{i=1}^{n} w_i \nabla_y g_i(x^{(t)}, \theta^{(t)})\Big\|^2 + \mathbb{E}\Big\|\frac{1}{\gamma}(\theta^{(t)} - y^{(t)}) + \sum_{i=1}^{n} w_i \nabla_y g_i(x^{(t)}, \theta^{(t)})$$

$$- \sum_{i=1}^{n} \nabla_y g_i(x^{(t)}, \theta_\gamma^*(x^{(t)}, y^{(t)})) - \frac{1}{\gamma}(\theta_\gamma^*(x^{(t)}, y^{(t)}) - y^{(t)})\Big\|^2\Big)$$

$$\overset{(b)}{\leq} 4\tau\eta_y^{(t)2}\big(\frac{L_f^2}{c_t{}^2} + L_g^2\big) + 6\tau\eta_y^{(t)2}L_g^2 + \frac{12}{\gamma^2}\eta_y^{(t)2}\sum_{i=1}^n w_i\frac{1}{\tau}\sum_{k=0}^{\tau-1}\sum_{j=0}^{k-1}\mathbb{E}\Big\|\theta_i^{(t,j)} - \theta^{(t)}\Big\|^2$$

$$+ \frac{12}{\gamma^2}\eta_y^{(t)2}\sum_{i=1}^n w_i\frac{1}{\tau}\sum_{k=0}^{\tau-1}\sum_{j=0}^{k-1}\mathbb{E}\Big\|y_i^{(t,j)} - y^{(t)}\Big\|^2 + 12\big(\frac{1}{\gamma^2} + L_2^2\big)\eta_y^{(t)2}\tau\Big\|\theta^{(t)} - \theta_\gamma^*(x^{(t)}, y^{(t)})\Big\|^2$$

$$\leq 4\tau\eta_y^{(t)2}\frac{L_f^2}{c_t{}^2} + 10\tau\eta_y^{(t)2}L_g^2 + \frac{12}{\gamma^2}\tau\eta_y^{(t)2}\Delta_\theta^t + \frac{12}{\gamma^2}\tau\eta_y^{(t)2}\Delta_y^t$$

$$+ 12\big(\frac{1}{\gamma^2} + L_2^2\big)\eta_y^{(t)2}\tau\Big\|\theta^{(t)} - \theta_\gamma^*(x^{(t)}, y^{(t)})\Big\|^2,$$

where (a) comes from the definition of $\theta_\gamma^*(x^{(t)}, y^{(t)})$ and (b) comes from the Lipschitz continuity and L-smoothness of $f_i(x, y)$ and $g_i(x, y)$ in Assumption 3.1 (i) and Assumption 3.2 (i). Then we have

$$\Delta_y^t \leq 2\eta_y^{(t)2}\tau\big(\frac{\delta_f^2}{c_t^2} + \delta_g^2\big) + 4\tau\eta_y^{(t)2}\frac{L_f^2}{c_t{}^2} + 10\tau\eta_y^{(t)2}L_g^2 + \frac{12}{\gamma^2}\tau\eta_y^{(t)2}\Delta_\theta^t + \frac{12}{\gamma^2}\tau\eta_y^{(t)2}\Delta_y^t$$

$$+ 12\big(\frac{1}{\gamma^2} + L_2^2\big)\eta_y^{(t)2}\tau\Big\|\theta^{(t)} - \theta_\gamma^*(x^{(t)}, y^{(t)})\Big\|^2. \tag{69}$$

Recall the inequality (67),

$$\Delta_\theta^t \leq \eta_\theta^{(t)2}\tau\delta_g^2 + 6\tau\eta_\theta^{(t)2}L_2^2\Delta_x^t + 6\tau\eta_\theta^{(t)2}\big(L_2^2 + \frac{1}{\gamma^2}\big)\Delta_\theta^t + 6\tau\eta_\theta^{(t)2}\frac{1}{\gamma^2}\Delta_y^t + 4\eta_\theta^{(t)2}\tau\Delta^2$$

$$+ 4\tau\eta_\theta^{(t)2}\big(\frac{1}{\gamma^2} + L_2^2\big)\mathbb{E}\Big\|\theta^{(t)} - \theta_\gamma^*(x^{(t)}, y^{(t)})\Big\|^2.$$

Next, we will prove that $\Delta_y^t$ and $\Delta_\theta^t$ can be bounded by $\mathbb{E}\Big\|\theta^{(t)} - \theta_\gamma^*(x^{(t)}, y^{(t)})\Big\|^2$ and some constants. Let's reformulate the inequalities (67) and (69),

$$\big(1 - 6\tau\eta_\theta^{(t)2}\big(L_2^2 + \frac{1}{\gamma^2}\big)\big)\Delta_\theta^t \leq \eta_\theta^{(t)2}\tau\delta_g^2 + 6\tau\eta_\theta^{(t)2}L_2^2\Delta_x^t + 6\tau\eta_\theta^{(t)2}\frac{1}{\gamma^2}\Delta_y^t + 4\eta_\theta^{(t)2}\tau\Delta^2$$

$$+ 4\tau\eta_\theta^{(t)2}\big(\frac{1}{\gamma^2} + L_2^2\big)\mathbb{E}\Big\|\theta^{(t)} - \theta_\gamma^*(x^{(t)}, y^{(t)})\Big\|^2, \tag{70}$$

and

$$\big(1 - 12\tau\eta_y^{(t)2}\frac{1}{\gamma^2}\big)\Delta_y^t \leq 2\eta_y^{(t)2}\tau\big(\frac{\delta_f^2}{c_t^2} + \delta_g^2\big) + 4\tau\eta_y^{(t)2}\frac{L_f^2}{c_t{}^2} + 10\tau\eta_y^{(t)2}L_g^2$$

$$+ \frac{12}{\gamma^2}\tau\eta_y^{(t)2}\Delta_\theta^t + 12\big(\frac{1}{\gamma^2} + L_2^2\big)\eta_y^{(t)2}\tau\Big\|\theta^{(t)} - \theta_\gamma^*(x^{(t)}, y^{(t)})\Big\|^2. \tag{71}$$

Multiply both sides of the inequality (70) by $\frac{\big(1 - 12\tau\eta_y^{(t)2}\frac{1}{\gamma^2}\big)}{6\tau\eta_\theta^{(t)2}\frac{1}{\gamma^2}}$,

$$\frac{\big(1 - 12\tau\eta_y^{(t)2}\frac{1}{\gamma^2}\big)}{6\tau\eta_\theta^{(t)2}\frac{1}{\gamma^2}}\big(1 - 6\tau\eta_\theta^{(t)2}\big(L_2^2 + \frac{1}{\gamma^2}\big)\big)\Delta_\theta^t$$

$$\leq \frac{\big(1 - 12\tau\eta_y^{(t)2}\frac{1}{\gamma^2}\big)}{6\tau\eta_\theta^{(t)2}\frac{1}{\gamma^2}}\Big(\eta_\theta^{(t)2}\tau\delta_g^2 + 6\tau\eta_\theta^{(t)2}L_2^2\Delta_x^t + 6\tau\eta_\theta^{(t)2}\frac{1}{\gamma^2}\Delta_y^t + 4\eta_\theta^{(t)2}\tau\Delta^2$$

$$+ 4\tau\eta_\theta^{(t)2}\big(\frac{1}{\gamma^2} + L_2^2\big)\mathbb{E}\Big\|\theta^{(t)} - \theta_\gamma^*(x^{(t)}, y^{(t)})\Big\|^2\Big). \tag{72}$$

Adding the inequality (72) to the inequality (71), we get:

$$\frac{\big(1 - 12\tau\eta_y^{(t)2}\frac{1}{\gamma^2}\big)}{6\tau\eta_\theta^{(t)2}\frac{1}{\gamma^2}}\big(1 - 6\tau\eta_\theta^{(t)2}\big(L_2^2 + \frac{1}{\gamma^2}\big)\big)\Delta_\theta^t$$

$$\leq \frac{\left(1 - 12\tau\eta_y^{(t)^2}\frac{1}{\gamma^2}\right)}{6\tau\eta_\theta^{(t)^2}\frac{1}{\gamma^2}} \left( \eta_\theta^{(t)^2}\tau\delta_g^2 + 6\tau\eta_\theta^{(t)^2}L_2^2\Delta_x^t + 4\eta_\theta^{(t)^2}\tau\Delta^2 \right.$$

$$+ 4\tau\eta_\theta^{(t)^2}\left(\frac{1}{\gamma^2} + L_2^2\right)\mathbb{E}\left\|\theta^{(t)} - \theta_\gamma^*(x^{(t)}, y^{(t)})\right\|^2 \right)$$

$$+ 2\eta_y^{(t)^2}\tau\left(\frac{\delta_f^2}{c_t^2} + \delta_g^2\right) + 4\tau\eta_y^{(t)^2}\frac{L_f^2}{c_t^2} + 10\tau\eta_y^{(t)^2}L_g^2 + \frac{12}{\gamma^2}\tau\eta_y^{(t)^2}\Delta_\theta^t$$

$$+ 12\left(\frac{1}{\gamma^2} + L_2^2\right)\eta_y^{(t)^2}\tau\left\|\theta^{(t)} - \theta_\gamma^*(x^{(t)}, y^{(t)})\right\|^2.$$

Then we can obtain

$$\Delta_\theta^t \leq \frac{1}{\frac{\left(1-12\tau\eta_y^{(t)^2}\frac{1}{\gamma^2}\right)}{6\tau\eta_\theta^{(t)^2}\frac{1}{\gamma^2}}\left(1 - 6\tau\eta_\theta^{(t)^2}(L_2^2 + \frac{1}{\gamma^2})\right) - 12\tau\eta_y^{(t)^2}\frac{1}{\gamma^2}} \left\{ \frac{\left(1 - 12\tau\eta_y^{(t)^2}\frac{1}{\gamma^2}\right)}{6\tau\eta_\theta^{(t)^2}\frac{1}{\gamma^2}} \left( \eta_\theta^{(t)^2}\tau\delta_g^2 \right.\right.$$

$$+ 6\tau\eta_\theta^{(t)^2}L_2^2\Delta_x^t + 4\eta_\theta^{(t)^2}\tau\Delta^2 + 4\tau\eta_\theta^{(t)^2}\left(\frac{1}{\gamma^2} + L_2^2\right)\mathbb{E}\left\|\theta^{(t)} - \theta_\gamma^*(x^{(t)}, y^{(t)})\right\|^2 \right)$$

$$+ 2\eta_y^{(t)^2}\tau\left(\frac{\delta_f^2}{c_t^2} + \delta_g^2\right) + 4\tau\eta_y^{(t)^2}\frac{L_f^2}{c_t^2} + 10\tau\eta_y^{(t)^2}L_g^2$$

$$+ 12\left(\frac{1}{\gamma^2} + L_2^2\right)\eta_y^{(t)^2}\tau\left\|\theta^{(t)} - \theta_\gamma^*(x^{(t)}, y^{(t)})\right\|^2 \right\}. \tag{73}$$

Then the $\Delta_\theta^t$ is bounded by $\mathbb{E}\left\|\theta^{(t)} - \theta_\gamma^*(x^{(t)}, y^{(t)})\right\|^2$ and some constants. Similarly, we can bound the $\Delta_y^{(t)}$,

$$\Delta_y^t \leq \frac{1}{\left(1 - 12\tau\eta_y^{(t)^2}\frac{1}{\gamma^2}\right) - \frac{12\tau\eta_y^{(t)^2}\frac{1}{\gamma^2}}{\left(1-6\tau\eta_\theta^{(t)^2}(L_2^2+\frac{1}{\gamma^2})\right)}6\tau\eta_\theta^{(t)^2}\frac{1}{\gamma^2}} \left\{ \frac{12\tau\eta_y^{(t)^2}\frac{1}{\gamma^2}}{\left(1 - 6\tau\eta_\theta^{(t)^2}(L_2^2 + \frac{1}{\gamma^2})\right)} \left( \eta_\theta^{(t)^2}\tau\delta_g^2 \right.\right.$$

$$+ 6\tau\eta_\theta^{(t)^2}L_2^2\Delta_x^t + 4\eta_\theta^{(t)^2}\tau\Delta^2 + 4\tau\eta_\theta^{(t)^2}\left(\frac{1}{\gamma^2} + L_2^2\right)\mathbb{E}\left\|\theta^{(t)} - \theta_\gamma^*(x^{(t)}, y^{(t)})\right\|^2 \right) \tag{74}$$

$$+ 2\eta_y^{(t)^2}\tau\left(\frac{\delta_f^2}{c_t^2} + \delta_g^2\right) + 4\tau\eta_y^{(t)^2}\frac{L_f^2}{c_t^2} + 10\tau\eta_y^{(t)^2}L_g^2$$

$$+ 12\left(\frac{1}{\gamma^2} + L_2^2\right)\eta_y^{(t)^2}\tau\left\|\theta^{(t)} - \theta_\gamma^*(x^{(t)}, y^{(t)})\right\|^2 \right\}.$$

$\square$

Note that we can choose proper $\eta_y^{(t)}$, $\eta_\theta^{(t)}$ to simplify inequalities (73) and (74). Now we choose $\eta_y^{(t)}$, $\eta_\theta^{(t)}$ such that $1 - 12\tau\eta_y^{(t)^2}\frac{1}{\gamma^2} \geq \frac{2}{3}, 1 - 6\tau\eta_\theta^{(t)^2}(L_2^2 + \frac{1}{\gamma^2}) \geq \frac{2}{3}$, namely, $12\tau\eta_y^{(t)^2}\frac{1}{\gamma^2} \leq \frac{1}{3}, 6\tau\eta_\theta^{(t)^2}(L_2^2 + \frac{1}{\gamma^2}) \leq \frac{1}{3}$.

For the term $\frac{1}{\frac{\left(1-12\tau\eta_y^{(t)^2}\frac{1}{\gamma^2}\right)}{6\tau\eta_\theta^{(t)^2}\frac{1}{\gamma^2}}\left(1-6\tau\eta_\theta^{(t)^2}(L_2^2+\frac{1}{\gamma^2})\right)-12\tau\eta_y^{(t)^2}\frac{1}{\gamma^2}}$ in inequality (73),

$$\frac{1}{\frac{\left(1-12\tau\eta_y^{(t)^2}\frac{1}{\gamma^2}\right)}{6\tau\eta_\theta^{(t)^2}\frac{1}{\gamma^2}}\left(1 - 6\tau\eta_\theta^{(t)^2}(L_2^2 + \frac{1}{\gamma^2})\right) - 12\tau\eta_y^{(t)^2}\frac{1}{\gamma^2}}$$

$$\leq \frac{6\tau\eta_\theta^{(t)^2}\frac{1}{\gamma^2} + 6\tau\eta_\theta^{(t)^2}\frac{1}{\gamma^2}}{\left(1 - 12\tau\eta_y^{(t)^2}\frac{1}{\gamma^2}\right)\left(1 - 6\tau\eta_\theta^{(t)^2}(L_2^2 + \frac{1}{\gamma^2})\right) - 12\tau\eta_y^{(t)^2}\frac{1}{\gamma^2}6\tau\eta_\theta^{(t)^2}\frac{1}{\gamma^2} + 6\tau\eta_\theta^{(t)^2}\frac{1}{\gamma^2}}$$

$$\leq \frac{12\tau\eta_\theta^{(t)^2}\frac{1}{\gamma^2}}{\left(1 - 12\tau\eta_y^{(t)^2}\frac{1}{\gamma^2}\right)\left(1 - 6\tau\eta_\theta^{(t)^2}(L_2^2)\right)}$$

$$\overset{(a)}{\leq} \frac{12\tau\eta_\theta^{(t)^2}(L_2^2 + \frac{1}{\gamma^2})}{\left(1 - 12\tau\eta_y^{(t)^2}\frac{1}{\gamma^2}\right)\left(1 - 6\tau\eta_\theta^{(t)^2}(L_2^2 + \frac{1}{\gamma^2})\right)} \leq 2,$$

where(a) comes from the inequalities $1 - 12\tau\eta_y^{(t)^2}\frac{1}{\gamma^2} \geq \frac{2}{3}, 1 - 6\tau\eta_\theta^{(t)^2}(L_2^2 + \frac{1}{\gamma^2}) \geq \frac{2}{3}$.

For the term $\frac{1}{\frac{\left(1 - 12\tau\eta_y^{(t)^2}\frac{1}{\gamma^2}\right)}{6\tau\eta_\theta^{(t)^2}\frac{1}{\gamma^2}}\left(1 - 6\tau\eta_\theta^{(t)^2}(L_2^2 + \frac{1}{\gamma^2})\right) - 12\tau\eta_y^{(t)^2}\frac{1}{\gamma^2}} \frac{\left(1 - 12\tau\eta_y^{(t)^2}\frac{1}{\gamma^2}\right)}{6\tau\eta_\theta^{(t)^2}\frac{1}{\gamma^2}}$ in inequality (73), we have

$$\frac{1}{\frac{\left(1 - 12\tau\eta_y^{(t)^2}\frac{1}{\gamma^2}\right)}{6\tau\eta_\theta^{(t)^2}\frac{1}{\gamma^2}}\left(1 - 6\tau\eta_\theta^{(t)^2}(L_2^2 + \frac{1}{\gamma^2})\right) - 12\tau\eta_y^{(t)^2}\frac{1}{\gamma^2}} \frac{\left(1 - 12\tau\eta_y^{(t)^2}\frac{1}{\gamma^2}\right)}{6\tau\eta_\theta^{(t)^2}\frac{1}{\gamma^2}}$$

$$\leq \frac{12\tau\eta_\theta^{(t)^2}\frac{1}{\gamma^2}}{\left(1 - 12\tau\eta_y^{(t)^2}\frac{1}{\gamma^2}\right)\left(1 - 6\tau\eta_\theta^{(t)^2}(L_2^2)\right)} \frac{\left(1 - 12\tau\eta_y^{(t)^2}\frac{1}{\gamma^2}\right)}{6\tau\eta_\theta^{(t)^2}\frac{1}{\gamma^2}}$$

$$\leq \frac{2}{1 - 6\tau\eta_\theta^{(t)^2}(L_2^2)} \leq \frac{2}{1 - 6\tau\eta_\theta^{(t)^2}(L_2^2 + \frac{1}{\gamma^2})} \leq 3.$$

Then the $\Delta_\theta^t$ can be simplified,

$$\Delta_\theta^t$$
$$\leq 2\left\{ 3\left(\eta_\theta^{(t)^2}\tau\delta_g^2 + 6\tau\eta_\theta^{(t)^2}L_2^2\Delta_x^t + 4\eta_\theta^{(t)^2}\tau\Delta^2 + 4\tau\eta_\theta^{(t)^2}\left(\frac{1}{\gamma^2} + L_2^2\right)\mathbb{E}\left\|\theta^{(t)} - \theta_\gamma^*(x^{(t)}, y^{(t)})\right\|^2\right) \right.$$

$$+ 2\eta_y^{(t)^2}\tau\left(\frac{\delta_f^2}{c_t^2} + \delta_g^2\right) + 4\tau\eta_y^{(t)^2}\frac{L_f^2}{c_t^2} + 10\tau\eta_y^{(t)^2}L_g^2$$

$$\left. + 12\left(\frac{1}{\gamma^2} + L_2^2\right)\eta_y^{(t)^2}\tau\mathbb{E}\left\|\theta^{(t)} - \theta_\gamma^*(x^{(t)}, y^{(t)})\right\|^2 \right\}. \tag{75}$$

$$\leq 6\eta_\theta^{(t)^2}\tau\delta_g^2 + 36\tau\eta_\theta^{(t)^2}L_2^2\Delta_x^t + 24\eta_\theta^{(t)^2}\tau\Delta^2 + 24\tau\eta_\theta^{(t)^2}\left(\frac{1}{\gamma^2} + L_2^2\right)\mathbb{E}\left\|\theta^{(t)} - \theta_\gamma^*(x^{(t)}, y^{(t)})\right\|^2$$

$$+ 8\eta_y^{(t)^2}\tau\left(\frac{\delta_f^2}{c_t^2} + \delta_g^2\right) + 8\tau\eta_y^{(t)^2}\frac{L_f^2}{c_t^2} + 20\tau\eta_y^{(t)^2}L_g^2 + 24\left(\frac{1}{\gamma^2} + L_2^2\right)\eta_y^{(t)^2}\tau\mathbb{E}\left\|\theta^{(t)} - \theta_\gamma^*(x^{(t)}, y^{(t)})\right\|^2$$

$$\leq 24\left(\frac{1}{\gamma^2} + L_2^2\right)\eta_y^{(t)^2}\tau\mathbb{E}\left\|\theta^{(t)} - \theta_\gamma^*(x^{(t)}, y^{(t)})\right\|^2 + 24\left(\frac{1}{\gamma^2} + L_2^2\right)\eta_\theta^{(t)^2}\tau\mathbb{E}\left\|\theta^{(t)} - \theta_\gamma^*(x^{(t)}, y^{(t)})\right\|^2$$

$$+ \tau\eta_\theta^{(t)^2}\left(6\delta_g^2 + 24\Delta^2\right) + \tau\eta_y^{(t)^2}\left(8\left(\frac{\delta_f^2}{c_t^2} + \delta_g^2\right) + 8\frac{L_f^2}{c_t^2} + 20L_g^2\right) + \tau\eta_\theta^{(t)^2}\tau\eta_x^{(t)^2}\mathcal{O}(1).$$

For the term $\frac{1}{\left(1 - 12\tau\eta_y^{(t)^2}\frac{1}{\gamma^2}\right) - \frac{12\tau\eta_y^{(t)^2}\frac{1}{\gamma^2}}{\left(1 - 6\tau\eta_\theta^{(t)^2}(L_2^2 + \frac{1}{\gamma^2})\right)}6\tau\eta_\theta^{(t)^2}\frac{1}{\gamma^2}}$ in inequality (74),

$$\frac{1}{\left(1 - 12\tau\eta_y^{(t)^2}\frac{1}{\gamma^2}\right) - \frac{12\tau\eta_y^{(t)^2}\frac{1}{\gamma^2}}{\left(1 - 6\tau\eta_\theta^{(t)^2}(L_2^2 + \frac{1}{\gamma^2})\right)}6\tau\eta_\theta^{(t)^2}\frac{1}{\gamma^2}}$$

$$\leq \frac{1 - 6\tau\eta_\theta^{(t)^2}(L_2^2 + \frac{1}{\gamma^2})}{\left(1 - 12\tau\eta_y^{(t)^2}\frac{1}{\gamma^2}\right)\left(1 - 6\tau\eta_\theta^{(t)^2}(L_2^2 + \frac{1}{\gamma^2})\right) - 12\tau\eta_y^{(t)^2}\frac{1}{\gamma^2}6\tau\eta_\theta^{(t)^2}\frac{1}{\gamma^2}}$$

$$\leq \frac{\left(1 - 12\tau\eta_y^{(t)^2}\frac{1}{\gamma^2}\right)\left(1 - 6\tau\eta_\theta^{(t)^2}(L_2^2 + \frac{1}{\gamma^2})\right) + 12\tau\eta_y^{(t)^2}\frac{1}{\gamma^2}\left(1 - 6\tau\eta_\theta^{(t)^2}(L_2^2 + \frac{1}{\gamma^2})\right)}{\left(1 - 12\tau\eta_y^{(t)^2}\frac{1}{\gamma^2}\right)\left(1 - 6\tau\eta_\theta^{(t)^2}(L_2^2 + \frac{1}{\gamma^2})\right) - 12\tau\eta_y^{(t)^2}\frac{1}{\gamma^2}6\tau\eta_\theta^{(t)^2}(\frac{1}{\gamma^2} + L_2^2)}$$

$$\leq 1 + \frac{12\tau\eta_y^{(t)^2}\frac{1}{\gamma^2} + 12\tau\eta_y^{(t)^2}\frac{1}{\gamma^2}}{\left(1 - 12\tau\eta_y^{(t)^2}\frac{1}{\gamma^2}\right)\left(1 - 6\tau\eta_\theta^{(t)^2}(L_2^2 + \frac{1}{\gamma^2})\right) + 12\tau\eta_y^{(t)^2}\frac{1}{\gamma^2}\left(1 - 6\tau\eta_\theta^{(t)^2}(\frac{1}{\gamma^2} + L_2^2)\right)}$$

$$= 1 + \frac{24\tau\eta_y^{(t)^2}\frac{1}{\gamma^2}}{1 - 6\tau\eta_\theta^{(t)^2}(L_2^2 + \frac{1}{\gamma^2})} \leq 2.$$

Then we have

$$\Delta_y^t \tag{76}$$

$$\leq 2\Bigg\{ \left(\eta_\theta^{(t)^2}\tau\delta_g^2 + 6\tau\eta_\theta^{(t)^2}L_2^2\Delta_x^t + 4\eta_\theta^{(t)^2}\tau\Delta^2 + 4\tau\eta_\theta^{(t)^2}\left(\frac{1}{\gamma^2} + L_2^2\right)\mathbb{E}\left\|\theta^{(t)} - \theta_\gamma^*(x^{(t)}, y^{(t)})\right\|^2\right)$$

$$+ 4\tau\eta_y^{(t)^2}\frac{L_f^2}{c_t^2} + 10\tau\eta_y^{(t)^2}L_g^2 + 12\left(\frac{1}{\gamma^2} + L_2^2\right)\eta_y^{(t)^2}\tau\mathbb{E}\left\|\theta^{(t)} - \theta_\gamma^*(x^{(t)}, y^{(t)})\right\|^2$$

$$+ 2\eta_y^{(t)^2}\tau\left(\frac{\delta_f^2}{c_t^2} + \delta_g^2\right) \Bigg\}$$

$$\leq 2\eta_\theta^{(t)^2}\tau\delta_g^2 + 12\tau\eta_\theta^{(t)^2}L_2^2\Delta_x^{(t)} + 8\eta_\theta^{(t)^2}\tau\Delta^2 + 8\tau\eta_\theta^{(t)^2}\left(\frac{1}{\gamma^2} + L_2^2\right)\mathbb{E}\left\|\theta^{(t)} - \theta_\gamma^*(x^{(t)}, y^{(t)})\right\|^2$$

$$+ 4\eta_y^{(t)^2}\tau\left(\frac{\delta_f^2}{c_t^2} + \delta_g^2\right) + 8\tau\eta_y^{(t)^2}\frac{L_f^2}{c_t^2} + 2\tau\eta_y^{(t)^2}L_g^2 + 12\tau\eta_y^{(t)^2}\left(\frac{1}{\gamma^2} + L_2^2\right)\mathbb{E}\left\|\theta^{(t)} - \theta_\gamma^*(x^{(t)}, y^{(t)})\right\|^2$$

$$\leq 8\tau\eta_\theta^{(t)^2}\left(\frac{1}{\gamma^2} + L_2^2\right)\mathbb{E}\left\|\theta^{(t)} - \theta_\gamma^*(x^{(t)}, y^{(t)})\right\|^2 + 12\tau\eta_y^{(t)^2}\left(\frac{1}{\gamma^2} + L_2^2\right)\left\|\theta^{(t)} - \theta_\gamma^*(x^{(t)}, y^{(t)})\right\|^2$$

$$+ \tau\eta_\theta^{(t)^2}\left(2\delta_g^2 + 8\Delta^2\right) + \tau\eta_y^{(t)^2}\left(4\left(\frac{\delta_f^2}{c_t^2} + \delta_g^2\right) + 8\frac{L_f^2}{c_t^2} + 2L_g^2\right) + \tau\eta_\theta^{(t)^2}\tau\eta_x^{(t)^2}\mathcal{O}(1).$$

**Lemma D.8** (Descent in Lyapunov function $\Psi_{c_t}(x^{(t)}, y^{(t)})$). *Fix the number of communication rounds $T$, local update rounds $\tau$, and the number of participating clients $P$ per communication round. We define the Lyapunov function as:*

$$\Psi_{c_t}(x^{(t)}, y^{(t)}) := \mathbb{E}\left[\Phi_{c_t}(x^{(t)}, y^{(t)})\right] + K|\theta^{(t)} - \theta_\gamma^*(x^{(t)}, y^{(t)})|^2, \tag{77}$$

*where $K := \frac{1}{2L_\theta}\sqrt{L_2^2 + \frac{3}{\gamma^2}}$, and $L_\theta$ is a positive constant provided in Lemma D.3. Under Assumptions 3.1 and 3.2, let the step sizes $\lambda_x^{(t)}, \lambda_y^{(t)}, \lambda_\theta^{(t)}, \eta_x^{(t)}, \eta_y^{(t)}, \eta_\theta^{(t)}$ satisfy the following conditions:*

$$\lambda_\theta^{(t)} \leq \frac{8KL_\theta^2}{\rho L_{\Phi_t}}, \quad \lambda_\theta^{(t)} \leq \frac{4}{5\rho}, \quad \lambda_x^{(t)} = \lambda_y^{(t)} = c_\lambda\lambda_\theta^{(t)}, \quad \eta_x^{(t)} = \eta_y^{(t)} = \eta_\theta^{(t)}, \tag{78}$$

$$\lambda_\theta^{(t)} \leq \frac{\rho}{2}\frac{1}{4(L_2^2 + \frac{1}{\gamma^2})\frac{n}{|C^{(t)}|}\left(\frac{|C^{(t)}|-1}{n-1}\right) + 6(L_2^2 + \frac{1}{\gamma^2})\frac{n}{|C^{(t)}|}\left(\frac{n-|C^{(t)}|}{n-1}\right)}, \tag{79}$$

$$\lambda_\theta^{(t)} \leq \frac{6}{\rho}L_2^2\left\{\frac{1}{9L_2^2\frac{\beta_{max}}{P}\left(\frac{n-P}{n-1}\right)}, \frac{1}{9\frac{1}{\gamma^2}\frac{\beta_{max}}{P}\left(\frac{n-P}{n-1}\right)}, \frac{1}{6L_2^2\frac{n}{P}\left(\frac{P-1}{n-1}\right)}, \frac{1}{6\frac{1}{\gamma^2}\frac{n}{P}\left(\frac{P-1}{n-1}\right)}\right\}, \tag{80}$$

$$\tau\eta_\theta^{(t)2}\lambda_\theta^{(t)} \le \frac{1}{128KL_\theta^2}$$

$$\cdot \frac{\left(L_2^2 + \frac{3}{\gamma^2}\right)\rho}{20\left(\frac{1}{\gamma^2} + L_2^2\right)\left(\left(\frac{3\rho}{32KL_\theta^2}\left(\frac{L_1^2}{c_t^2} + L_2^2\right) + \frac{1}{2}\frac{\rho}{32KL_\theta^2}\left(6\frac{L_1^2}{c_t^2} + 6L_2^2 + \frac{3}{\gamma^2}\right)\right) + 2K\frac{18}{\rho}\frac{1}{\gamma^2}\right)},$$

$$(81)$$

$$\tau\eta_\theta^{(t)2}\lambda_\theta^{(t)} \le \frac{1}{128KL_\theta^2}\frac{\left(L_2^2 + \frac{3}{\gamma^2}\right)\rho}{48\left(\frac{1}{\gamma^2} + L_2^2\right)\left(\frac{3\rho}{32KL_\theta^2}\left(L_2^2 + \frac{1}{\gamma^2}\right) + 2K\frac{18}{\rho}\left(L_2^2 + \frac{1}{\gamma^2}\right)\right)},$$

$$(82)$$

$$\tau\eta_\theta^{(t)2}\lambda_\theta^{(t)} \le \frac{1}{36}\gamma^2, \quad \tau\eta_\theta^{(t)2}\lambda_\theta^{(t)} \le \frac{1}{18}\frac{1}{L_2^2 + \frac{1}{\gamma^2}},$$

$$(83)$$

*with $c_\lambda := \frac{\rho}{32KL_\theta^2}$ and $\rho := \frac{1}{\gamma} - L_2$, then we have*

$$\Psi_{c_{t+1}}(x^{(t+1)}, y^{(t+1)}) - \Psi_{c_t}(x^{(t)}, y^{(t)})$$

$$\le -\frac{4KL_\theta^2}{\rho}\frac{1}{\lambda_\theta^{(t)}}\mathbb{E}\|x^{(t+1)} - x^{(t)}\|^2 - \frac{4KL_\theta^2}{\rho}\frac{1}{\lambda_\theta^{(t)}}\mathbb{E}\|y^{(t+1)} - y^{(t)}\|^2$$

$$- \frac{1}{64KL_\theta^2}\left(L_2^2 + \frac{3}{\gamma^2}\right)\lambda_\theta^{(t)}\mathbb{E}\left\|\theta^{(t)} - \theta_\gamma^*(x^{(t)}, y^{(t)})\right\|^2 + \epsilon_{sto}^{(t)} + \epsilon_{dh}^{(t)} + \epsilon_{cd}^{(t)},$$

$$(84)$$

*where the error $\epsilon_{sfo}^{(t)}, \epsilon_{dh}^{(t)}, \epsilon_{cd}^{(t)}$ respectively come from stochastic first-order oracle, data heterogeneity and client drifts:*

$$\epsilon_{sfo}^{(t)} := M_1'\frac{\lambda_\theta^{(t)2}}{P\tau}, \quad \epsilon_{dh}^{(t)} := M_2'\frac{1}{P}\frac{n - P}{n - 1}\lambda_\theta^{(t)2}, \quad \epsilon_{cd}^{(t)} := M_3'\tau\eta_\theta^{(t)2}\lambda_\theta^{(t)}.$$

$$(85)$$

*Here*

$$M_1' := 2K\beta_{\max}\delta_g^2, M_2' := 6K\beta_{max}\Delta^2,$$

$$(86)$$

$$M_3' := \left(6\delta_g^2 + 24\Delta^2 + 8\left(\frac{\delta_f^2}{c_t^2} + \delta_g^2\right) + 8\frac{L_f^2}{c_t^2} + 20L_g^2 + \tau\eta_x^{(t)2}\mathcal{O}(1)\right)$$

$$\cdot \left(\left(3L_2^2 + \frac{3}{\gamma^2}\right)\frac{\rho}{32KL_\theta^2} + (1 + \delta_{t,1})K\frac{18}{\rho}\left(L_2^2 + \frac{1}{\gamma^2}\right)\right)$$

$$+ \left(2\delta_g^2 + 8\Delta^2 + 4\left(\frac{\delta_f^2}{c_t^2} + \delta_g^2\right) + 8\frac{L_f^2}{c_t^2} + 2L_g^2 + \tau\eta_\theta^{(t)2}\tau\eta_x^{(t)2}\mathcal{O}(1)\right)$$

$$\cdot \left(3\left(\frac{L_1^2}{c_t^2} + L_2^2\right) + \frac{1}{2}\left(6\frac{L_1^2}{c_t^2} + 6L_2^2 + \frac{3}{\gamma^2}\right) + (1 + \delta_{t,1})K\frac{18}{\rho}\frac{1}{\gamma^2}\right),$$

$$(87)$$

*with $\delta_{t,1} := \frac{\lambda_\theta^{(t)}\rho}{4\left(1 - \frac{\lambda_\theta^{(t)}\rho}{2}\right)}$.*

*Proof.*

$$\Psi_{c_{t+1}}(x^{(t+1)}, y^{(t+1)}) - \Psi_{c_t}(x^{(t)}, y^{(t)})$$

$$\le \mathbb{E}\left[\Phi_{c_{t+1}}(x^{(t+1)}, y^{(t+1)})\right] + K\|\theta^{(t+1)} - \theta_\gamma^*(x^{(t+1)}, y^{(t+1)})\|^2$$

$$- \left(\mathbb{E}\left[\Phi_{c_t}(x^{(t)}, y^{(t)})\right] + K\|\theta^{(t)} - \theta_\gamma^*(x^{(t)}, y^{(t)})\|^2\right)$$

$$\overset{(a)}{\le} -\left(\frac{1}{2\lambda_x^{(t)}} - \frac{L_{\Phi_t}}{2}\right)\mathbb{E}\|x^{(t+1)} - x^{(t)}\|^2 - \left(\frac{1}{2\lambda_y^{(t)}} - \frac{L_{\Phi_t}}{2}\right)\mathbb{E}\|y^{(t+1)} - y^{(t)}\|^2$$

$$+ \left(\lambda_x^{(t)}L_2^2 + \lambda_y^{(t)}\frac{3}{\gamma^2}\right)\mathbb{E}\left\|\theta^{(t)} - \theta_\gamma^*(x^{(t)}, y^{(t)})\right\|^2 + \left(3\lambda_x^{(t)}\left(\frac{L_1^2}{c_t^2} + 2L_2^2\right) + 3\lambda_y^{(t)}\left(\frac{L_1^2}{c_t^2} + L_2^2\right)\right)\Delta_x^{(t)}$$

$$+ \left( 3\lambda_x^{(t)}\big(\frac{L_1^2}{c_t^2} + L_2^2\big) + \frac{\lambda_y^{(t)}}{2}\big(6\frac{L_1^2}{c_t^2} + 6L_2^2 + \frac{3}{\gamma^2}\big) \right) \Delta_y^{(t)} + \left( 3\lambda_x^{(t)}L_2^2 + \frac{3}{\gamma^2}\lambda_y^{(t)} \right) \Delta_\theta^{(t)}$$

$$+ K \Bigg\{ 2L_\theta^2\big(1 + \frac{1}{\delta_{t,1}}\big) \left\| (x^{(t+1)}, y^{(t+1)}) - (x^{(t)}, y^{(t)}) \right\|^2 + (1 + \delta_{t,1})\frac{\beta_{\max}}{P\tau}\delta_g^2\lambda_\theta^{(t)\,2}$$

$$+ 3(1 + \delta_{t,1})\frac{\beta_{max}}{P}\left(\frac{n - P}{n - 1}\right)\Delta^2\lambda_\theta^{(t)\,2} + (1 + \delta_{t,1})\Bigg( \Big(1 - \rho\lambda_\theta^{(t)}$$

$$+ 4(L_2^2 + \frac{1}{\gamma^2})\frac{n}{P}\left(\frac{P - 1}{n - 1}\right)\lambda_\theta^{(t)\,2} + 6(L_2^2 + \frac{1}{\gamma^2})\frac{\beta_{max}}{P}\left(\frac{n - P}{n - 1}\right)\lambda_\theta^{(t)\,2}\Big)\mathbb{E}\left\| \theta^{(t)} - \theta_\gamma^*(x^{(t)}, y^{(t)}) \right\|^2$$

$$+ \Big(6L_2^2\frac{n}{P}\left(\frac{P - 1}{n - 1}\right)\lambda_\theta^{(t)\,2} + 9L_2^2\frac{\beta_{max}}{P}\left(\frac{n - P}{n - 1}\right)\lambda_\theta^{(t)\,2}$$

$$+ \frac{6}{\rho}L_2^2\lambda_\theta^{(t)}\Big)\Delta_x^{(t)} + \Big(6(L_2^2 + \frac{1}{\gamma^2})\frac{n}{P}\left(\frac{P - 1}{n - 1}\right)\lambda_\theta^{(t)\,2} + 9(L_2^2 + \frac{1}{\gamma^2})\frac{\beta_{max}}{P}\left(\frac{n - P}{n - 1}\right)\lambda_\theta^{(t)\,2}$$

$$+ \frac{6}{\rho}(L_2^2 + \frac{1}{\gamma^2})\lambda_\theta^{(t)}\Big)\Delta_\theta^{(t)} + \Big(6\frac{1}{\gamma^2}\frac{n}{P}\left(\frac{P - 1}{n - 1}\right)\lambda_\theta^{(t)\,2} + 9\frac{1}{\gamma^2}\frac{\beta_{max}}{P}\left(\frac{n - P}{n - 1}\right)\lambda_\theta^{(t)\,2}$$

$$+ \frac{6}{\rho}\frac{1}{\gamma^2}\lambda_\theta^{(t)}\Big)\Delta_y^{(t)}\Bigg) - \mathbb{E}\left\| \theta^{(t)} - \theta_\gamma^*(x^{(t)}, y^{(t)}) \right\|^2 \Bigg\}.$$

$$\overset{(b)}{\leq} - \Big(\frac{1}{2\lambda_x^{(t)}} - \frac{L_{\Phi_t}}{2} - 2KL_\theta^2\big(1 + \frac{1}{\delta_{t,1}}\big)\Big)\mathbb{E}\|x^{(t+1)} - x^{(t)}\|^2$$

$$- \Big(\frac{1}{2\lambda_y^{(t)}} - \frac{L_{\Phi_t}}{2} - 2KL_\theta^2\big(1 + \frac{1}{\delta_{t,1}}\big)\Big)\mathbb{E}\|y^{(t+1)} - y^{(t)}\|^2$$

$$+ \Bigg\{ \big(\lambda_x^{(t)}L_2^2 + \lambda_y^{(t)}\frac{3}{\gamma^2}\big) - K\Bigg( (1 + \delta_{t,1})\Big(1 - \rho\lambda_\theta^{(t)} + 4(L_2^2 + \frac{1}{\gamma^2})\frac{n}{P}\left(\frac{P - 1}{n - 1}\right)\lambda_\theta^{(t)\,2}$$

$$+ 6(L_2^2 + \frac{1}{\gamma^2})\frac{\beta_{max}}{P}\left(\frac{n - P}{n - 1}\right)\lambda_\theta^{(t)\,2}\Big) - 1\Bigg) \Bigg\}\mathbb{E}\left\| \theta^{(t)} - \theta_\gamma^*(x^{(t)}, y^{(t)}) \right\|^2$$

$$+ K\Bigg( (1 + \delta_{t,1})\frac{\beta_{\max}}{P\tau}\delta_g^2\lambda_\theta^{(t)\,2} + 3(1 + \delta_{t,1})\frac{\beta_{max}}{P}\left(\frac{n - P}{n - 1}\right)\Delta^2\lambda_\theta^{(t)\,2}\Bigg)$$

$$+ \Bigg( \Big(3\lambda_x^{(t)}\big(\frac{L_1^2}{c_t^2} + 2L_2^2\big) + 3\lambda_y^{(t)}\big(\frac{L_1^2}{c_t^2} + L_2^2\big)\Big)$$

$$+ (1 + \delta_{t,1})K\Big(6L_2^2\frac{n}{P}\left(\frac{P - 1}{n - 1}\right)\lambda_\theta^{(t)\,2} + 9L_2^2\frac{\beta_{max}}{P}\left(\frac{n - P}{n - 1}\right)\lambda_\theta^{(t)\,2} + \frac{6}{\rho}L_2^2\lambda_\theta^{(t)}\Big)\Bigg)\Delta_x^{(t)}$$

$$+ \Bigg( 8\tau\eta_\theta^{(t)\,2}\big(\frac{1}{\gamma^2} + L_2^2\big)\mathbb{E}\left\| \theta^{(t)} - \theta_\gamma^*(x^{(t)}, y^{(t)}) \right\|^2 + 12\tau\eta_y^{(t)\,2}\big(\frac{1}{\gamma^2} + L_2^2\big)\mathbb{E}\left\| \theta^{(t)} - \theta_\gamma^*(x^{(t)}, y^{(t)}) \right\|^2$$

$$+ \tau\eta_\theta^{(t)\,2}\Big(2\delta_g^2 + 8\Delta^2 + 4\big(\frac{\delta_f^2}{c_t^2} + \delta_g^2\big) + 8\frac{L_f^2}{c_t^2} + 2L_g^2\Big) + \tau\eta_\theta^{(t)\,2}\tau\eta_x^{(t)\,2}\mathcal{O}(1) \Bigg)$$

$$\cdot \Bigg( \Big(3\lambda_x^{(t)}\big(\frac{L_1^2}{c_t^2} + L_2^2\big) + \frac{\lambda_y^{(t)}}{2}\big(6\frac{L_1^2}{c_t^2} + 6L_2^2 + \frac{3}{\gamma^2}\big)\Big)$$

$$+ (1 + \delta_{t,1})K\Big(6\frac{1}{\gamma^2}\frac{n}{P}\left(\frac{P - 1}{n - 1}\right)\lambda_\theta^{(t)\,2} + 9\frac{1}{\gamma^2}\frac{\beta_{max}}{P}\left(\frac{n - P}{n - 1}\right)\lambda_\theta^{(t)\,2} + \frac{6}{\rho}\frac{1}{\gamma^2}\lambda_\theta^{(t)}\Big)\Bigg)$$

$$+ \Bigg( 24\tau\eta_\theta^{(t)\,2}\big(\frac{1}{\gamma^2} + L_2^2\big)\mathbb{E}\left\| \theta^{(t)} - \theta_\gamma^*(x^{(t)}, y^{(t)}) \right\|^2 + 24\tau\eta_y^{(t)\,2}\big(\frac{1}{\gamma^2} + L_2^2\big)\left\| \theta^{(t)} - \theta_\gamma^*(x^{(t)}, y^{(t)}) \right\|^2$$

$$+ \tau\eta_\theta^{(t)\,2}\Big(6\delta_g^2 + 24\Delta^2 + 8\big(\frac{\delta_f^2}{c_t^2} + \delta_g^2\big) + 8\frac{L_f^2}{c_t^2} + 20L_g^2\Big) + \tau\eta_\theta^{(t)\,2}\tau\eta_x^{(t)\,2}\mathcal{O}(1) \Bigg)$$

$$
\cdot \left( \left( 3\lambda_x^{(t)} L_2^2 + \frac{3}{\gamma^2} \lambda_y^{(t)} \right) + (1 + \delta_{t,1}) K \left( 6 \left( L_2^2 + \frac{1}{\gamma^2} \right) \frac{n}{P} \left( \frac{P-1}{n-1} \right) \lambda_\theta^{(t)\,2} \right. \right.
$$
$$
\left. \left. + 9 \left( L_2^2 + \frac{1}{\gamma^2} \right) \frac{\beta_{max}}{P} \left( \frac{n-P}{n-1} \right) \lambda_\theta^{(t)\,2} + \frac{6}{\rho} \left( L_2^2 + \frac{1}{\gamma^2} \right) \lambda_\theta^{(t)} \right) \right), \tag{88}
$$

where (a) comes from Lemmas D.5 and D.6, (b) comes from the Lemma D.7 and the inequalities (75) and (76).

For the term $\frac{1}{2\lambda_x^{(t)}} - \frac{L_{\Phi_t}}{2} - 2KL_\theta^2 \left( 1 + \frac{1}{\delta_{t,1}} \right)$ in the first line of Eq.(88), we take $\delta_{t,1} = \frac{\lambda_\theta^{(t)} \rho}{4 \left( 1 - \frac{\lambda_\theta^{(t)} \rho}{2} \right)}$, then we have

$$
- \left( \frac{1}{2\lambda_x^{(t)}} - \frac{L_{\Phi_t}}{2} - 2KL_\theta^2 \left( 1 + \frac{1}{\delta_{t,1}} \right) \right)
$$
$$
= - \left( \frac{1}{2\lambda_x^{(t)}} - \frac{L_{\Phi_t}}{2} - 2KL_\theta^2 \left( \frac{4}{\rho \lambda_\theta^{(t)}} - 2 \right) \right)
$$
$$
\leq - \left( \frac{1}{2\lambda_x^{(t)}} - \frac{L_{\Phi_t}}{2} - \frac{8KL_\theta^2}{\rho \lambda_\theta^{(t)}} \right)
$$
$$
\overset{(a)}{\leq} - \left( \frac{4KL_\theta^2}{\rho \lambda_\theta^{(t)}} \right), \tag{89}
$$

where (a) comes from the condition (78). Similarly, for the term $\frac{1}{2\lambda_y^{(t)}} - \frac{L_{\Phi_t}}{2} - 2KL_\theta^2 \left( 1 + \frac{1}{\delta_{t,1}} \right)$ in the first line of Eq.(88),

$$
- \left( \frac{1}{2\lambda_y^{(t)}} - \frac{L_{\Phi_t}}{2} - 2KL_\theta^2 \left( 1 + \frac{1}{\delta_{t,1}} \right) \right) \leq - \left( \frac{4KL_\theta^2}{\rho \lambda_\theta^{(t)}} \right). \tag{90}
$$

For the coefficients of $\mathbb{E} \left\| \theta^{(t)} - \theta_\gamma^*(x^{(t)}, y^{(t)}) \right\|^2$ in the second and third lines in the Eq.(88), we have

$$
\left( \lambda_x^{(t)} L_2^2 + \lambda_y^{(t)} \frac{3}{\gamma^2} \right) - K \left( (1 + \delta_{t,1}) \left( 1 - \rho \lambda_\theta^{(t)} + 4 \left( L_2^2 + \frac{1}{\gamma^2} \right) \frac{n}{P} \left( \frac{P-1}{n-1} \right) \lambda_\theta^{(t)\,2} \right. \right.
$$
$$
\left. \left. + 6 \left( L_2^2 + \frac{1}{\gamma^2} \right) \frac{\beta_{max}}{P} \left( \frac{n-P}{n-1} \right) \lambda_\theta^{(t)\,2} \right) - 1 \right)
$$
$$
\overset{(a)}{\leq} \lambda_x^{(t)} L_2^2 + \lambda_y^{(t)} \frac{3}{\gamma^2} - K \frac{\lambda_\theta^{(t)} \rho}{4}
$$
$$
\overset{(b)}{=} \rho \lambda_\theta^{(t)} \left( -\frac{K}{4} + \frac{L_2^2}{32KL_\theta^2} + \frac{3}{\gamma^2} \frac{1}{32KL_\theta^2} \right) \overset{(c)}{=} - \frac{1}{32KL_\theta^2} \left( L_2^2 + \frac{3}{\gamma^2} \right) \lambda_\theta^{(t)} \rho, \tag{91}
$$

where (a) comes from the condition (80), (b), (c) comes from the condition (78). Similarly, we take the conditions (78), (81), (82), (79) and (80), we have

$$
\left( 8\tau \eta_\theta^{(t)\,2} \left( \frac{1}{\gamma^2} + L_2^2 \right) \mathbb{E} \left\| \theta^{(t)} - \theta_\gamma^*(x^{(t)}, y^{(t)}) \right\|^2 + 12\tau \eta_y^{(t)\,2} \left( \frac{1}{\gamma^2} + L_2^2 \right) \mathbb{E} \left\| \theta^{(t)} - \theta_\gamma^*(x^{(t)}, y^{(t)}) \right\|^2 \right)
$$
$$
\cdot \left( \left( 3\lambda_x^{(t)} \left( \frac{L_1^2}{c_t^2} + L_2^2 \right) + \frac{\lambda_y^{(t)}}{2} \left( 6\frac{L_1^2}{c_t^2} + 6L_2^2 + \frac{3}{\gamma^2} \right) \right) \right.
$$
$$
\left. + (1 + \delta_{t,1}) K \left( 6 \frac{1}{\gamma^2} \frac{n}{P} \left( \frac{P-1}{n-1} \right) \lambda_\theta^{(t)\,2} + 9 \frac{1}{\gamma^2} \frac{\beta_{max}}{P} \left( \frac{n-P}{n-1} \right) \lambda_\theta^{(t)\,2} + \frac{6}{\rho} \frac{1}{\gamma^2} \lambda_\theta^{(t)} \right) \right)
$$
$$
\leq \frac{1}{128KL_\theta^2} \left( L_2^2 + \frac{3}{\gamma^2} \right) \lambda_\theta^{(t)} \rho, \tag{92}
$$

and

$$
\left(24\tau\eta_\theta^{(t)\,2}(\frac{1}{\gamma^2}+L_2^2)\mathbb{E}\left\|\theta^{(t)}-\theta_\gamma^*(x^{(t)},y^{(t)})\right\|^2 + 24\tau\eta_y^{(t)\,2}(\frac{1}{\gamma^2}+L_2^2)\left\|\theta^{(t)}-\theta_\gamma^*(x^{(t)},y^{(t)})\right\|^2\right)
$$

$$
\cdot\left(\left(3\lambda_x^{(t)}L_2^2+\frac{3}{\gamma^2}\lambda_y^{(t)}\right)+(1+\delta_{t,1})K\left(6\big(L_2^2+\frac{1}{\gamma^2}\big)\frac{n}{P}\left(\frac{P-1}{n-1}\right)\lambda_\theta^{(t)\,2}\right.\right.
$$

$$
\left.\left. +9\big(L_2^2+\frac{1}{\gamma^2}\big)\frac{\beta_{max}}{P}\left(\frac{n-P}{n-1}\right)\lambda_\theta^{(t)\,2}+\frac{6}{\rho}\big(L_2^2+\frac{1}{\gamma^2}\big)\lambda_\theta^{(t)}\right)\right)
$$

$$
\leq\frac{1}{128KL_\theta^2}\big(L_2^2+\frac{3}{\gamma^2}\big)\lambda_\theta^{(t)}\rho\mathbb{E}\left\|\theta^{(t)}-\theta_\gamma^*(x^{(t)},y^{(t)})\right\|^2. \tag{93}
$$

Combining with the inequalities (88), (91), (92), (93), we have

$$
\Psi_{c_{t+1}}(x^{(t+1)},y^{(t+1)})-\Psi_{c_t}(x^{(t)},y^{(t)})
$$

$$
\leq -\frac{4KL_\theta^2}{\rho}\frac{1}{\lambda_\theta^{(t)}}\|x^{(t+1)}-x^{(t)}\|^2 - \frac{4KL_\theta^2}{\rho}\frac{1}{\lambda_\theta^{(t)}}\|y^{(t+1)}-y^{(t)}\|^2
$$

$$
-\frac{1}{64KL_\theta^2}\big(L_2^2+\frac{3}{\gamma^2}\big)\lambda_\theta^{(t)}\mathbb{E}\left\|\theta^{(t)}-\theta_\gamma^*(x^{(t)},y^{(t)})\right\|^2+\epsilon_{\text{sto}}^{(t)}+\epsilon_{\text{dh}}^{(t)}+\epsilon_{\text{cd}}^{(t)}, \tag{94}
$$

where

$$
\epsilon_{\text{sfo}}^{(t)}:=2K\beta_{\max}\delta_g^2\lambda_\theta^{(t)\,2}\frac{\lambda_\theta^{(t)\,2}}{P\tau}, \quad \epsilon_{\text{dh}}^{(t)}:=6K\beta_{\max}\Delta^2\frac{1}{P}\frac{n-P}{n-1}\lambda_\theta^{(t)\,2} \tag{95}
$$

$$
\epsilon_{\text{cd}}^{(t)}:=\left(\tau\eta_\theta^{(t)\,2}\left(6\delta_g^2+24\Delta^2+8\big(\frac{\delta_f^2}{c_t^2}+\delta_g^2\big)+8\frac{L_f^2}{c_t^2}+20L_g^2\right)+\tau\eta_\theta^{(t)\,2}\tau\eta_x^{(t)\,2}\mathcal{O}\left(1\right)\right)
$$

$$
\cdot\left(\left(3\lambda_x^{(t)}L_2^2+\frac{3}{\gamma^2}\lambda_y^{(t)}\right)+(1+\delta_{t,1})K\frac{18}{\rho}\big(L_2^2+\frac{1}{\gamma^2}\big)\lambda_\theta^{(t)}\right)
$$

$$
+\left(\tau\eta_\theta^{(t)\,2}\left(2\delta_g^2+8\Delta^2+4\big(\frac{\delta_f^2}{c_t^2}+\delta_g^2\big)+8\frac{L_f^2}{c_t^2}+2L_g^2\right)+\tau\eta_\theta^{(t)\,2}\tau\eta_x^{(t)\,2}\mathcal{O}\left(1\right)\right)
$$

$$
\cdot\left(\left(3\lambda_x^{(t)}\big(\frac{L_1^2}{c_t^2}+L_2^2\big)+\frac{\lambda_y^{(t)}}{2}\big(6\frac{L_1^2}{c_t^2}+6L_2^2+\frac{3}{\gamma^2}\big)\right)+(1+\delta_{t,1})K\frac{18}{\rho}\frac{1}{\gamma^2}\lambda_\theta^{(t)}\right)
$$

$$
=\tau\eta_\theta^{(t)\,2}\lambda_\theta^{(t)}\left(\left(6\delta_g^2+24\Delta^2+8\big(\frac{\delta_f^2}{c_t^2}+\delta_g^2\big)+8\frac{L_f^2}{c_t^2}+20L_g^2+\tau\eta_x^{(t)\,2}\mathcal{O}\left(1\right)\right)\right.
$$

$$
\cdot\left(\big(3L_2^2+\frac{3}{\gamma^2}\big)\frac{\rho}{32KL_\theta^2}+(1+\delta_{t,1})K\frac{18}{\rho}\big(L_2^2+\frac{1}{\gamma^2}\big)\right)
$$

$$
+\left(2\delta_g^2+8\Delta^2+4\big(\frac{\delta_f^2}{c_t^2}+\delta_g^2\big)+8\frac{L_f^2}{c_t^2}+2L_g^2+\tau\eta_\theta^{(t)\,2}\tau\eta_x^{(t)\,2}\mathcal{O}\left(1\right)\right)
$$

$$
\left.\cdot\left(3\big(\frac{L_1^2}{c_t^2}+L_2^2\big)+\frac{1}{2}\big(6\frac{L_1^2}{c_t^2}+6L_2^2+\frac{3}{\gamma^2}\big)+(1+\delta_{t,1})K\frac{18}{\rho}\frac{1}{\gamma^2}\right)\right). \tag{96}
$$

$$\square$$

The following remarks provide crucial insights into the step size condition and the boundedness of certain terms in our analysis:

*Remark* D.9. It is important to note that the condition for the step size in Lemma D.8 is indeed satisfiable. We maintain the same inequality direction with the step size on the left-hand side and a constant on the right-hand side. Furthermore, we ensure that the right-hand side is always positive, thereby guaranteeing the existence of a valid step size.

Building upon the step size condition, we now turn our attention to the boundedness of a key term in our analysis:

*Remark* D.10. It is noteworthy that $M_3'$ defined in (86) can be regarded as a constant when the step sizes are bounded. Next, we briefly explain this: to prove that $M_3'$ can be regarded as a constant, we only need to prove that $M_3'$ is bounded. In $M_3'$, we only need to demonstrate that $\delta_{t,1}$ is bounded (since the step size are bounded and $\frac{1}{c_t}$ is decaying). Recall that $\delta_{t,1} := \frac{\lambda_\theta^{(t)} \rho}{4(1 - \frac{\lambda_\theta^{(t)} \rho}{2})}$. Noting that $\delta_{t,1}$ is increasing with respect to $\lambda_\theta^{(t)}$, and considering that $\lambda_\theta^{(t)} \leq \frac{4}{5\rho}$, we have $\delta_{t,1} \leq \frac{1}{3}$. Therefore, $M_3'$ is bounded.

It is noteworthy that, based on Remark D.10, we can conclude that a certain term $M_1$ in the following theorem can be regarded as a constant.

**Theorem D.11.** *Fix the number of communication rounds $T$, local update rounds $\tau$, and the number of participating clients $P$ per communication round. We define the Lyapunov function as:*

$$\Psi_{c_t}(x^{(t)}, y^{(t)}) := \mathbb{E}\Big[\Phi_{c_t}(x^{(t)}, y^{(t)})\Big] + K|\theta^{(t)} - \theta_\gamma^*(x^{(t)}, y^{(t)})|^2, \tag{97}$$

*where $K := \frac{1}{2L_\theta}\sqrt{L_2^2 + \frac{3}{\gamma^2}}$, and $L_\theta$ is a positive constant provided in Lemma D.3. Under Assumptions 3.1 and 3.2, let the step sizes $\lambda_x^{(t)}, \lambda_y^{(t)}, \lambda_\theta^{(t)}, \eta_x^{(t)}, \eta_y^{(t)}, \eta_\theta^{(t)}$ satisfy the following conditions: Under Assumptions 3.1 and 3.2, let $c_k = \underline{c}(k+1)^p$ with $\underline{c} > 0$ and $\gamma \in (0, \frac{1}{2L_2})$.*

*(i) For decreasing step sizes, we choose*

$$\lambda_\theta^{(t)} = c_\lambda \tau^{\frac{1}{4}}(t+1)^{-q}, \quad \frac{1}{2} < q < 1, \quad 0 < p < \frac{1-q}{2}, \quad \eta_\theta^{(t)} = c_\eta \frac{1}{\tau^{7/8}\sqrt{P}}(t+1)^{-1/4}, \tag{98}$$

*where $c_\lambda$ and $c_\eta$ are some positive constants. If the step sizes $\lambda_\theta^{(t)}, \lambda_x^{(t)}, \lambda_y^{(t)}, \eta_\theta^{(t)}, \eta_x^{(t)}, \eta_y^{(t)}$ satisfy the conditions in Lemma D.8, then the sequence of $(x^{(t)}, y^{(t)}, \theta^{(t)})$ generated by Algorithm 1 satisfies*

$$\min_{0 \leq t \leq T-1} \mathbb{E}\Big[R_t(x^{(t+1)}, y^{(t+1)})\Big] = \mathcal{O}\left(\frac{1}{P}\left(\frac{1}{T^{1-q}\tau^{1/2}} + \frac{\tau^{1/2}(n-P)}{T^{1-q}n} + \frac{1}{T^{3/2-q}\tau^{1/4}}\right)\right) \tag{99}$$

*(ii) For fixed step sizes, we choose $p \in (0, 1/4)$ and*

$$\lambda_\theta^{(t)} = c_\lambda \tau^{\frac{1}{4}}T^{-1/2}, \quad \eta_\theta^{(t)} = c_\eta \frac{1}{\tau^{7/8}T^{1/4}\sqrt{P}}, \tag{100}$$

*where the $c_\lambda$ and $c_\eta$ are some positive constants. If the step sizes $\lambda_\theta^{(t)}, \lambda_x^{(t)}, \lambda_y^{(t)}, \eta_\theta^{(t)}, \eta_x^{(t)}, \eta_y^{(t)}$ satisfy the conditions in Lemma D.8, then the sequence of $(x^{(t)}, y^{(t)}, \theta^{(t)})$ generated by Algorithm 1 satisfies*

$$\min_{0 \leq t \leq T-1} \mathbb{E}\Big[R_t(x^{(t+1)}, y^{(t+1)})\Big] = \mathcal{O}\left(\frac{1}{P}\left(\frac{1}{T^{1/2}\tau^{1/2}} + \frac{\tau^{1/2}(n-P)}{T^{1/2}n} + \frac{1}{T\tau^{1/4}}\right)\right). \tag{101}$$

*Proof.* Based on (84), we have

$$\frac{4KL_\theta^2}{\rho\lambda_\theta^{(t)}}\mathbb{E}\|x^{(t+1)} - x^{(t)}\|^2 + \frac{4KL_\theta^2}{\rho\lambda_\theta^{(t)}}\mathbb{E}\|y^{(t+1)} - y^{(t)}\|^2$$

$$+ \frac{1}{64KL_\theta^2}\left(L_2^2 + \frac{1}{\gamma^2}\right)\lambda_\theta^{(t)}\rho\mathbb{E}\left\|\theta^{(t)} - \theta_\gamma^*(x^{(t)}, y^{(t)})\right\|^2 \tag{102}$$

$$\leq \mathbb{E}\Big(\Psi_{c_t}(x^{(t)}, y^{(t)}) - \Psi_{c_{t+1}}(x^{(t+1)}, y^{(t+1)})\Big) + \mathcal{O}\left(M_1'\frac{\lambda_\theta^{(t)2}}{P\tau}\right) + \mathcal{O}\left(M_2'\frac{1}{P}\frac{n-P}{n-1}\lambda_\theta^{(t)2}\right)$$

$$+ \mathcal{O}\left(M_3'\tau\eta_\theta^{(t)2}\lambda_\theta^{(t)}\right).$$

Upon telescoping (102) over the range $k = 0, 1, ..., K-1$, we derive

$$\sum_{t=0}^{T-1}\frac{4KL_\theta^2}{\rho\lambda_\theta^{(t)}}\mathbb{E}\|x^{(t+1)} - x^{(t)}\|^2 + \frac{4KL_\theta^2}{\rho\lambda_\theta^{(t)}}\mathbb{E}\|y^{(t+1)} - y^{(t)}\|^2$$

$$+ \frac{1}{64KL_\theta^2}\Big(L_2^2 + \frac{1}{\gamma^2}\Big)\lambda_\theta^{(t)}\rho\mathbb{E}\Big\|\theta^{(t)} - \theta_\gamma^*(x^{(t)}, y^{(t)})\Big\|^2$$

$$\leq \mathbb{E}\Psi_{c_0}(x^{(0)}, y^{(0)}) + \mathcal{O}\left(M_1'\frac{\sum_{t=0}^{T-1}\lambda_\theta^{(t)^2}}{P\tau}\right) + \mathcal{O}\left(M_2'\frac{1}{P}\frac{n-P}{n-1}\sum_{t=0}^{T-1}\lambda_\theta^{(t)^2}\right) \tag{103}$$

$$+ \mathcal{O}\left(M_3'\tau\sum_{t=0}^{T-1}\eta_\theta^{(t)^2}\lambda_\theta^{(t)}\right)$$

$$\leq \mathcal{O}\left(M_1'\frac{1}{P\sqrt{\tau}}\right) + \mathcal{O}\left(M_2'\frac{1}{P}\frac{n-P}{n-1}\sqrt{\tau}\right) + \mathcal{O}\left(\sqrt{\tau}\frac{1}{P}\Big(M_3' + P\Psi_{c_0}(x^{(0)}, y^{(0)})\Big).\right)$$

As a consequence of the weak convexity of $g$ and its continuous differentiability with respect to $y$, as stipulated in Assumption 3.2 (i) and supported by Gao et al. (2023), we deduce that

$$(e_x^{(t)}, e_y^{(t)}) \in \frac{1}{c_t}\nabla\mathbf{F}(x^{(t+1)}, y^{(t+1)}) + \nabla\mathbf{G}(x^{(t+1)}, y^{(t+1)}) - \nabla\mathbf{v}_\gamma(x^{(t+1)}, y^{(t+1)})$$
$$+ \mathcal{N}_{X\times Y}(x^{(t+1)}, y^{(t+1)}),$$

with

$$e_x^{(t)} := \nabla_x\Phi_{c_t}(x^{(t+1)}, y^{(t+1)}) - \widetilde{h}_x^{(t)} - \frac{1}{\lambda_x^{(t)}}\left(x^{(t+1)} - x^{(t)}\right),$$

$$e_y^{(t)} := \nabla_y\Phi_{c_t}(x^{(t+1)}, y^{(t+1)}) - \widetilde{h}_y^{(t)} - \frac{1}{\lambda_y^{(t)}}\left(y^{(t+1)} - y^{(t)}\right).$$

Next, we estimate $\|e_x^{(t)}\|$,

$$\|e_x^{(t)}\| \leq \|\nabla_x\Psi_{c_t}(x^{(t+1)}, y^{(t+1)}) - \nabla_x\Psi_{c_t}(x^{(t)}, y^{(t)})\|$$
$$+ \|\nabla_x\Psi_{c_t}(x^{(t)}, y^{(t)}) - \widetilde{h}_x^{(t)}\| + \frac{1}{\lambda_x^{(t)}}\mathbb{E}\|x^{(t+1)} - x^{(t)}\|. \tag{104}$$

Considering the first term on the right hand side of the preceding inequality, there exists $L_{\psi_1} > 0$,

$$\|\nabla_x\Psi_{c_t}(x^{(t+1)}, y^{(t+1)}) - \nabla_x\Psi_{c_t}(x^{(t)}, y^{(t)})\| \leq L_{\psi_1}\|(x^{(t+1)}, y^{(t+1)}) - (x^{(t)}, y^{(t)})\|. \tag{105}$$

For the term $\|\nabla_x\Psi_{c_t}(x^{(t)}, y^{(t)}) - h_x^{(t)}\|$, similar to the analysis of (52), we have

$$\|\nabla_x\Psi_{c_t}(x^{(t)}, y^{(t)}) - h_x^{(t)}\| \leq \left\|\nabla_x\phi_{c_t}(x^{(t)}, y^{(t)}) - h_x^{(t)}\right\|$$
$$\leq L_2\|\theta^{(t)} - \theta_\gamma^*(x^{(t)}, y^{(t)})\| + \mathcal{O}\left(\sqrt{\tau\eta_\theta^{(t)^2}\lambda_\theta^{(t)}}\right). \tag{106}$$

Hence, we have

$$\|e_x^{(t)}\| \leq L_{\psi_1}\|(x^{(t+1)}, y^{(t+1)}) - (x^{(t)}, y^{(t)})\| + \frac{1}{\lambda_x^{(t)}}\|x^{(t+1)} - x^{(t)}\|$$
$$+ \|\theta^{(t)} - \theta_\gamma^*(x^{(t)}, y^{(t)})\| + \mathcal{O}\left(\sqrt{\tau\eta_\theta^{(t)^2}\lambda_\theta^{(t)}}\right). \tag{107}$$

Similarly,

$$\|e_y^{(t)}\| \leq L_{\psi_2}\|(x^{(t+1)}, y^{(t+1)}) - (x^{(t)}, y^{(t)})\| + \frac{1}{\lambda_y^{(t)}}\|y^{(t+1)} - y^{(t)}\|$$
$$+ \frac{1}{\gamma}\|\theta^{(t)} - \theta_\gamma^*(x^{(t)}, y^{(t)})\| + \mathcal{O}\left(\sqrt{\tau\eta_\theta^{(t)^2}\lambda_\theta^{(t)}}\right). \tag{108}$$

By the definition of $D_t(x^{(t+1)}, y^{(t+1)})$ and the inequalities (107) and (108), we have

$$D_t(x^{(t+1)}, y^{(t+1)})$$

$$\leq (L_{\psi_1} + L_{\psi_1}) \|(x^{(t+1)}, y^{(t+1)}) - (x^{(t)}, y^{(t)})\| + (L_2 + \frac{c_t}{\gamma}) \|\theta^{(t)} - \theta_\gamma^*(x^{(t)}, y^{(t)})\|$$

$$+ \frac{1}{\lambda_x^{(t)}} \|x^{(t+1)} - x^{(t)}\| + \frac{1}{\lambda_y^{(t)}} \|y^{(t+1)} - y^{(t)}\| + \mathcal{O}\left(\sqrt{\tau \eta_\theta^{(t)^2} \lambda_\theta^{(t)}}\right).$$

By utilizing the inequality mentioned above and performing left and right multiplication by $\lambda_\theta^{(t)}$, we establish the existence of $C_R > 0$ such that

$$\lambda_\theta^{(t)} R_t(x^{(t+1)}, y^{(t+1)})$$

$$\leq C_R \left( \frac{4KL_\theta^2}{\rho \lambda_\theta^{(t)}} \|x^{(t+1)} - x^{(t)}\|^2 + \frac{4KL_\theta^2}{\rho \lambda_\theta^{(t)}} \|y^{(t+1)} - y^{(t)}\|^2 \right.$$

$$\left. + \frac{1}{64 K L_\theta^2} \left(L_2^2 + \frac{1}{\gamma^2}\right) \lambda_\theta^{(t)} \rho \left\|\theta^{(t)} - \theta_\gamma^*(x^{(t)}, y^{(t)})\right\|^2 \right) + \mathcal{O}\left(\lambda_\theta^{(t)^2} \tau \eta_\theta^{(t)^2}\right). \tag{109}$$

Combining this with (103) implies that

$$\sum_{t=0}^{T-1} \lambda_\theta^{(t)} R_t(x^{(t+1)}, y^{(t+1)}) \leq \mathcal{O}\left( M_1 \frac{\sum_{t=0}^{T-1} \lambda_\theta^{(t)^2}}{P\tau} \right) + \mathcal{O}\left( M_2 \frac{1}{P} \frac{n-P}{n-1} \sum_{t=0}^{T-1} \lambda_\theta^{(t)^2} \right)$$

$$+ \mathcal{O}\left( \sum_{t=0}^{T-1} \lambda_\theta^{(t)^2} \tau \eta_\theta^{(t)^2} \right), \tag{110}$$

where $M_1 := \mathbb{E}\Psi_{c_0}(x^{(0)}, y^{(0)}) + M_1' + M_3'$ and $M_2 := M_2'$.

For decreasing step size, we choose the step size as in (98), then $\sum_{t=0}^{T-1} \lambda_\theta^{(t)^2} = \mathcal{O}(1)$ and $\sum_{t=0}^{T-1} \eta_\theta^{(t)^2} \lambda_\theta^{(t)} = \mathcal{O}(1)$. Because $1/2 < q < 1$, it holds that

$$\mathcal{O}\left( \sum_{t=0}^{T-1} \lambda_\theta^{(t)} \right) = \mathcal{O}\left( \sum_{t=0}^{T-1} \left(\frac{1}{t+1}\right)^q \right) \geq \mathcal{O}\left( \frac{(T+1)^{1-q}}{(1-q)} \right). \tag{111}$$

Then we have

$$\min_{0 \leq t \leq T-1} \mathbb{E}\left[ R_t(x^{(t+1)}, y^{(t+1)}) \right] = \mathcal{O}\left( \frac{1}{P} \left( \frac{1}{T^{1-q}\tau^{1/2}} + \frac{\tau^{1/2}(n-P)}{T^{1-q}n} + \frac{1}{T^{3/2-q}\tau^{1/4}} \right) \right) \tag{112}$$

From the definition of $R_t$, we have

$$\min_{0 \leq t \leq T-1} \mathbb{E}\left[ \nabla \frac{1}{c_t^2} \mathbf{F}(x^{(t+1)}, y^{(t+1)}) \right] = \mathcal{O}\left( \frac{1}{P} \left( \frac{1}{T^{1-q}\tau^{1/2}} + \frac{\tau^{1/2}(n-P)}{T^{1-q}n} + \frac{1}{T^{3/2-q}\tau^{1/4}} \right) \right).$$

If $0 < p < \frac{1-q}{2}$, then $\nabla \mathbf{F}(x^{(t+1)}, y^{(t+1)})$ satisfies

$$\min_{0 \leq t \leq T-1} \mathbb{E}\left[ \nabla \mathbf{F}(x^{(t+1)}, y^{(t+1)}) \right]$$

$$= \mathcal{O}\left( \frac{1}{P} \left( \frac{1}{T^{1-q-2p}\tau^{1/2}} + \frac{\tau^{1/2}(n-P)}{T^{1-q-2p}n} + \frac{1}{T^{3/2-q-2p}\tau^{1/4}} \right) \right), \tag{113}$$

which ensures the convergence of $\nabla \mathbf{F}(x^{(t+1)}, y^{(t+1)})$. For fixed step size, we choose the step size as in (100), it holds that

$$\min_{0 \leq t \leq T-1} \mathbb{E}\left[ R_t(x^{(t+1)}, y^{(t+1)}) \right] = \mathcal{O}\left( \frac{1}{P} \left( \frac{1}{T^{1/2}\tau^{1/2}} + \frac{\tau^{1/2}(n-P)}{T^{1/2}n} + \frac{1}{T\tau^{1/4}} \right) \right).$$

$\square$

### D.5    COMPLEXITY ANALYSIS

**Corollary D.12.** *Under the setting of Theorem 3.4, we have the following results:*

*(i) In the case of full client participation, setting $\tau = \mathcal{O}(T)$, the per-client sample complexity is $\tau T = \mathcal{O}(\epsilon^{-2})$ and the communication complexity $T = \mathcal{O}(\epsilon^{-1})$.*

*(ii) For partial client participation, setting $\tau = \mathcal{O}(1)$, the per-client sample complexity is $\tau T = \mathcal{O}(P^{-2}\epsilon^{-2})$ and the communication complexity $T = \mathcal{O}(P^{-2}\epsilon^{-2})$.*

*Proof.* From Theorem D.11, for nearly full client participation, which means that $\frac{n-P}{n} \approx 0$, then we have

$$\min_{0 \leq t \leq T-1} \mathbb{E}\left[R_t(x^{(t+1)}, y^{(t+1)})\right] = \mathcal{O}\left(\frac{1}{P}\left(\frac{1}{T^{1/2}\tau^{1/2}} + \frac{1}{T\tau^{1/4}}\right)\right) \leq \epsilon. \quad (114)$$

Then we can obtain the per-client sample complexity $\tau T = \mathcal{O}\left(\epsilon^{-2}\right)$, then local update rounds can saving communication rounds, we take $\tau = \mathcal{O}(T)$, then we have $T = \mathcal{O}\left(\epsilon^{-1}\right)$.

For partial client participation, we can obtain that the number of local update cannot affect the whole convergence rate. From Theorem D.11, we have

$$\min_{0 \leq t \leq T-1} \mathbb{E}\left[R_t(x^{(t+1)}, y^{(t+1)})\right] = \mathcal{O}\left(\frac{1}{P}\left(\frac{1}{T^{1/2}\tau^{1/2}} + \frac{\tau^{1/2}(n-P)}{T^{1/2}n} + \frac{1}{T\tau^{1/4}}\right)\right). \quad (115)$$

When we take $\tau = \mathcal{O}(1)$ will lead to the best performance. Then we have

$$\min_{0 \leq t \leq T-1} \mathbb{E}\left[R_t(x^{(t+1)}, y^{(t+1)})\right] = \mathcal{O}\left(\frac{1}{P}\left(\frac{1}{T^{1/2}}\right)\right) \leq \epsilon,$$

Since $T \gg P$, the per-client sample complexity is $\tau T = \mathcal{O}(P^{-2}\epsilon^{-2})$ and communication rounds $T = \mathcal{O}\left(P^{-2}\epsilon^{-2}\right)$ $\qquad\square$

## E    RELATED WORK

### E.1    BILEVEL OPTIMIZATION WITHOUT LLSC

Since the seminal work by Bracken & McGill (1973), numerous studies have proposed various methods for addressing bilevel optimization problems. Extensive overviews of these approaches can be found in surveys Sinha et al. (2017); Liu et al. (2021a); Zhang et al. (2023). In this section, we provide a brief overview of relevant work on bilevel optimization (BLO) without the lower-level (LL) strong convexity assumption. Beyond the LL strong convexity assumption, Liu et al. (2021b) developed a method with initialization auxiliary and pessimistic trajectory truncation. Huang (2023a) proposed a momentum-based implicit gradient BLO algorithm and established a convergence analysis framework under a nondegenerate condition on the LL Hessian. Arbel & Mairal (2022) extended implicit differentiation to a class of non-convex LL functions with possibly degenerate critical points and developed unrolled optimization algorithms. Xiao et al. (2023) developed a generalized alternating method for BLO with a non-convex LL objective. However, these works require second-order gradient information. In contrast, the value function reformulation of BLO was first utilized in Liu et al. (2023) to develop BLO algorithms in machine learning using an interior-point method. Subsequently, Liu et al. (2022) introduced a fully first-order value function-based BLO algorithm and established non-asymptotic convergence results. Recently, Shen & Chen (2023) proposed a penalty-based fully first-order BLO algorithm, relaxing the relatively restrictive assumption on the boundedness of both the upper-level (UL) and LL objectives present in Liu et al. (2022). Notably, these works involve a double-loop structure, which makes them challenging to employ in Federated Bilevel Optimization. To mitigate the requirement of single-loop structure, Liu et al. (2024); Yao et al. (2023) developed single-loop and Hessian-free gradient-based methods utilizing a Moreau envelope-based reformulation of bilevel optimization.

## E.2 FEDERATED (BILEVEL) LEARNING

Federated Learning (FL) was initially proposed by Google to coordinate the collaborative training of a common task across thousands of clients while addressing data isolation and privacy concerns(McMahan et al., 2017). As the pioneering algorithm, FedAvg(McMahan et al., 2017) has effectively addressed the aforementioned issues; however, it has also given rise to a series of new challenges such as fairness, communication overhead, malicious participants and privacy (Mohri et al., 2019; Stich, 2019; Yu et al., 2019; Wang & Joshi, 2021; Bagdasaryan et al., 2020; Bhagoji et al., 2019; Kim & Hong, 2019; Aïvodji et al., 2019). To address these challenges, some research efforts have introduced a nested optimization structure, known as Federated Bilevel learning, such as (Xing et al., 2022; Li et al., 2022; Zeng et al., 2021; Li et al., 2021b; Hu et al., 2024a; Huang et al., 2022; Tolpegin et al., 2020; Sun et al., 2021; Zhang et al., 2021; Cheng et al., 2024). In response to the demand for solving problems with such model requirements, there have been some effective attempts. As one of the earliest methods of federated bilevel optimization(Tarzanagh et al., 2022), FedNest is a federated alternating stochastic gradient method based on AID-based hypergradient estimation to address general federated nested problems, which needs the federated hypergradient estimation. Additionally, there are other FBO algorithms based on AID-based hypergradient estimation(Huang et al., 2023). Xiao & Ji (2023) introduced a federated Bilevel Optimization algorithm with hypergradient estimation based on ITD-based hypergradient estimation. Recent FBO methods, such as SimFBO (Yang et al., 2023) and FedBiOAcc (Li et al., 2023), draw inspiration from SOBA (Dagréou et al., 2022). These approaches transform a linear system problem into a quadratic one, improving computational efficiency within a single-loop algorithmic framework. Notably, FedBiOAcc incorporates a momentum-based technique. While these algorithms have made significant progress, they continue to rely on Hessian matrix computations and are constrained by the requirement for lower-level strong convexity (LLSC).

In addition to the aforementioned works, there is a growing body of research on bilevel optimization in asynchronous settings, such as those by Jiao et al. (2022) and Li et al. (2024). Furthermore, FBO has demonstrated promising practical applications, particularly in the fine-tuning of large language models (LLMs) within federated settings. For instance, Wu et al. (2024) investigates the use of FBO for local fine-tuning of LLMs. As noted in Table 2 of Wu et al. (2024), these models can be optimized using either single-level or bilevel approaches. Notably, the bilevel optimization method, FedBiOT, proposed by Wu et al. (2024), exhibits significant advantages over single-level optimization, especially in scenarios involving hierarchical problem structures.

## F  ADDITIONAL EXPERIMENT RESULTS

Extending beyond Section 4.3, we conducted comprehensive experiments with both expanded datasets and more sophisticated neural architectures. We evaluated MeFBO (Algorithm 1), SimFBO (Yang et al., 2023), FedNest (Tarzanagh et al., 2022), and LFedNest (Tarzanagh et al., 2022) on additional data hyper-cleaning tasks. The experiments employed either a 2-layer MLP (as in Section 4.3) or a 7-layer CNN (LeCun et al., 1998) on Fashion MNIST (Xiao et al., 2017) and CIFAR-10 (Krizhevsky et al., 2009) datasets. Following (Xiao & Ji, 2023), we also implemented a 2-layer MLP architecture for CIFAR-10.

Fashion-MNIST represents a moderate increase in complexity compared to MNIST, featuring fashion items rather than handwritten digits. Both datasets contain 70,000 grayscale images (28×28). CIFAR-10, comprising 60,000 color images (32×32, RGB), presents greater complexity due to its real-world object representations and multi-channel color information.
Figure 14 summarizes our experimental findings, yielding several key insights:

**Results on Larger Datasets and Complex Neural Architectures.** Figure 14 presents comparative performance analyses across different network architectures and datasets under i.i.d. settings with a corruption rate (cr) of 0.7:

- **Fashion MNIST with 2-layer MLP**: As shown in Figure 14(a), while maintaining consistency with Section 4.3's architecture but scaling to a larger dataset, MeFBO demonstrated superior convergence characteristics, achieving the highest test accuracy (approximately 85

- **CIFAR-10 with 7-layer CNN**: As illustrated in Figure 14(b), when tested on the more complex CIFAR-10 dataset with a 7-layer CNN architecture, MeFBO maintained its superior performance, achieving the highest accuracy among all methods. Notably, SimFBO encountered memory constraints that prevented its execution on this larger-scale task.

- **Architecture Comparison on CIFAR-10**: Following (Xiao & Ji, 2023), we compared the performance of 2-layer MLP and 7-layer CNN architectures using MeFBO on CIFAR-10, as shown in Figure 14(c). The MLP exhibited faster initial convergence but plateaued at a lower accuracy, while the CNN achieved higher ultimate accuracy despite slower convergence. However, these performance levels suggest room for improvement. We hypothesize that architectural limitations may be constraining performance on this specific task, warranting further investigation in future research.

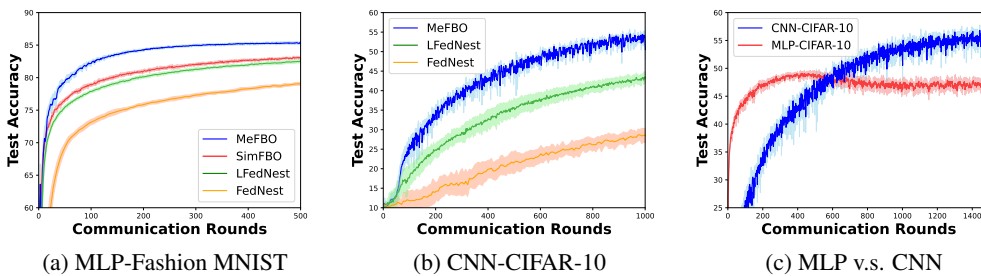

| (a) MLP-Fashion MNIST | (b) CNN-CIFAR-10 | (c) MLP v.s. CNN |

Figure 14: Comparison of different algorithms in federated data hyper-cleaning under a label corruption rate of $cr = 0.7$.

**Hyperparameter.** For all methods, 2 clients are randomly selected from a pool of 20 clients to participate in each communication round. All algorithms are implemented with a batch size of 256. For our method, MeFBO, and SimFBO, we set the number of local updates $\tau = 1$, and $w_i = 0.1$. For MeFBO, the parameter $c_k = 3(t+1)^{0.001}$ and $\gamma = 0.015$. The local step sizes $[\eta_x^{(t)}, \eta_y^{(t)}, \eta_\theta^{(t)}]$ and $[\lambda_x^{(t)}, \lambda_y^{(t)}, \lambda_\theta^{(t)}]$ are set to $[0.2, 0.15, 0.1]$ for MLP-Fashion MNIST, and $[0.3, 0.3, 0.05]$ for CNN-CIFAR-10. For SimFBO, the local step sizes $[\eta_x, \eta_y, \eta_v]$ and $[\gamma_x, \gamma_y, \eta_v]$ are both set to $[0.1, 0.05, 0.015]$. For FedNest and LFedNest, we set the inner and outer step sizes as follows: $\alpha = 0.02$, $\beta = 0.03$ for CNN-CIFAR-10, and $\alpha = 0.01$, $\beta = 0.02$ for MLP-Fashion MNIST.

# G   COMPARISON OF $R_t$ IN (13) AND HYPERGRADIENT

In the case where the lower-level problem is non-convex, the hypergradient cannot be well-defined, making a direct comparison with $R_t$ in (13) infeasible. Consequently, the analysis is confined to scenarios where the lower-level problem is strongly convex. Specifically, let $g_i(x, y)$ be a strongly convex function, with $X = \mathbb{R}^{d_x}$ and $Y = \mathbb{R}^{d_y}$. The hypergradient $\nabla \Phi(x)$ is associated with the hyper-objective $\Phi(x) := F(x, y^*(x))$, where $y^*(x)$ is the unique lower-level (LL) optimal solution. Using the expression (6) for $\nabla v_\gamma(x, y)$ and the optimality condition of $\theta_\gamma^* := \theta_\gamma^*(x, y)$, given as

$$\nabla_y \mathbf{G}(x, \theta_\gamma^*) + \frac{\theta_\gamma^* - y}{\gamma} = 0,$$

the residual function $R_t(x, y)$ can be expressed as:

$$R_t(x, y) = \frac{1}{c_t^2} \left\| \begin{pmatrix} \nabla_x \mathbf{F}(x, y) + c_t \left[ \nabla_x \mathbf{G}(x, y) - \nabla_x \mathbf{G}(x, \theta_\gamma^*(x, y)) \right] \\ \nabla_y \mathbf{F}(x, y) + c_t \left[ \nabla_y \mathbf{G}(x, y) - \nabla_y \mathbf{G}(x, \theta_\gamma^*(x, y)) \right] \end{pmatrix} \right\|^2 := \frac{1}{c_t^2} \left\| \begin{pmatrix} R_t^{(1)}(x, y) \\ R_t^{(2)}(x, y) \end{pmatrix} \right\|^2.$$

Compared to the stationarity measure $\tilde{R}_t(x, y)$ proposed in Equation (15) of Liu et al. (2024), our work establishes the following relationship:

$$\sqrt{R_t(x, y)} = \frac{1}{c_t} \tilde{R}_t(x, y)$$

Thus, we can directly apply Lemma A.14 in Liu et al. (2024):

**Lemma G.1.** *(Liu et al., 2024) Under Assumptions 3.1 and 3.2, suppose that $X = \mathbb{R}^{d_x}$, $Y = \mathbb{R}^{d_y}$, and the lower-level objective $g_i(x, y)$ is a $\mu$-strongly convex. Let $\gamma > 1/\mu$, then*

$$\|y - y^*(x)\| \leq \frac{2L_f + 4\|R_t^{(2)}(x, y)\|}{c_t \mu}. \tag{116}$$

*Additionally, suppose $\|R_t^{(2)}(x, y)\| \leq L_f/c_t$, then $c_t\|y - y^*(x)\| \leq 6L_f/\mu$. If further $\nabla_{xy}^2 G(x, \cdot), \nabla_{yy}^2 G(x, \cdot)$ are $L_{G,2}$-Lipschitz continuous, then*

$$\|\nabla \Phi(x) - R_t^{(1)}(x, y)\| \leq \frac{L_\mu}{c_t} + \frac{L_2}{\mu} \|R_t^{(2)}(x, y)\|,$$

*where $L_\mu := \frac{6L_f}{\mu} \left(1 + \frac{L_2}{\mu}\right) \left(L_1 + \frac{6L_{G,2}L_f}{\mu}\right) + \frac{6L_\Phi L_f}{\mu^2 \gamma}$ with $L_\Phi$ is a positive constant defined in Lemma 2.2 of Ghadimi & Wang (2018).*

