# OpenReview forum: "MeFBO: A Moreau Envelope Based First-Order Stochastic Gradient Method for Nonconvex Federated Bilevel Optimization"
_ICLR.cc/2025/Conference — Submitted to ICLR 2025_

### Official Review · Reviewer_u495 · 2024-10-26

**Soundness:** 1
**Presentation:** 2
**Contribution:** 1
**Rating:** 3
**Confidence:** 4

**Summary:**

This work aims to solve the federated bilevel optimization problem using the first-order based method. The theoretical analyses for the approximation problem of the federated bilevel optimization problem are provided in this work. Experiments on hyper-parameter optimization tasks are conducted to evaluate the proposed approach.

**Strengths:**

1. A method based on Moreau envelope is proposed for federated bilevel optimization problem.

2. Some theoretical analyses are provided for the proposed method.

**Weaknesses:**

I find it difficult to appreciate this work due to the following key reasons.

(W.1.) In the proposed method, the exterior penalty approach is employed to approximate the original constrained optimization problem (2) into the unconstrained optimization problem (4). However, how can you ensure that the optimal solution obtained from the approximation problem (4) is also optimal for the original problem (2)? Furthermore, I found that the convergence rate is related to reaching a stationary point for the approximation problem (4). The stationary point in (2) may deviate from the one in (4).

(W.2.) The main problem studied in this work is not interesting (at least to me). The first-order gradient method for nonconvex distributed centralized (federated) problems was already studied two years ago [1]. Additionally, I did not find any comparisons related to this work in the manuscript.

(W.3.) I have some concerns about the complexity of the proposed algorithm. In each communication round, the algorithm requires solving an additional projection problem as described in Eq. (12). In bilevel deep learning, the variables are often high-dimensional, which increases the complexity of solving the optimization problem.

(W.4.) The experimental results are not sufficiently convincing due to the following reasons:

(W.4.1) Only the MNIST dataset is used to evaluate the proposed method. Could you conduct experiments on larger-scale datasets?

(W.4.2) The evaluation is limited to hyper-parameter optimization tasks. It is unclear if the proposed method is applicable to other bilevel optimization tasks, such as meta-learning in [2] or neural architecture search (NAS) in [3]. Could you provide additional experimental results on these tasks?

(W.4.3) The experiments are conducted on 2- or 3-layer multi-layer perceptrons, raising concerns about the method’s efficiency when applied to models with more parameters. This point is also related to W.3, as larger-scale models may significantly impact the efficiency of the proposed algorithm.

Besides the key concerns, I also have lots of concerns as follows.

(W.5.) Additional comparisons of the convergence rate between the proposed method and existing distributed centralized (federated) bilevel methods should be included to demonstrate the superiority of the proposed approach.

(W.6.) The readability of this work is poor, as it contains numerous notations, some of which lack sufficient explanations. For instance, the exact meaning of $\nabla_1$ and $\nabla_2$ in Eq. (9) is not clearly defined.

(W.7.) In the experiment, the author claims that MeFBO outperforms other methods in both i.i.d. and non-i.i.d. settings. I would appreciate it if the authors could provide more detailed explanations and discussions on these results.

[1] Asynchronous distributed bilevel optimization

[2] Model-agnostic meta-learning for fast adaptation of deep networks

[3] Darts: Differentiable architecture search

**Questions:**

I have lots of questions about this work, please refer to the Weaknesses.

---

> ### Author Response · Authors · 2024-11-22
> **Response to theoretical concerns (Part 1/2)**
>
> > (W.1.) In the proposed method, the exterior penalty approach is employed to approximate the original constrained optimization problem (2) into the unconstrained optimization problem (4). However, how can you ensure that the optimal solution obtained from the approximation problem (4) is also optimal for the original problem (2)? Furthermore, I found that the convergence rate is related to reaching a stationary point for the approximation problem (4). The stationary point in (2) may deviate from the one in (4).
>
> $\textbf{Response:}$ First, solving problem (2) directly is challenging due to the absence of a constraint regularity (or qualification) condition. To the best of our knowledge, no universally accepted stationarity condition has been established for problem (2).
>
> Second, the penalty method provides a simple and effective approach for addressing constrained optimization problems. Approximate stationary solutions of the penalized problem (4) are widely regarded as suitable surrogates for solving problem (2). For further details, see Shen and Chen (ICLM 2023), Kwon et al. (ICLR 2024), and the references therein.
>
> Third, our algorithm, MeFBO, is theoretically proven to find approximate stationary solutions to problem (4). Empirically, it demonstrates comparable or superior performance while also offering improved robustness over existing federated bilevel optimization methods.
>
>
> > (W.2.) The main problem studied in this work is not interesting (at least to me). The first-order gradient method for nonconvex distributed centralized (federated) problems was already studied two years ago [1]. Additionally, I did not find any comparisons related to this work in the manuscript.
>
> $\textbf{Response:}$ (1)Thank you for informing us about [1]. It is an interesting study that focuses on the asynchronous setting. Our work, along with the closely related studies listed in Table 1 of our manuscript, focuses on the synchronous setting. Given the importance of the asynchronous setting in certain application scenarios, we will include a discussion of asynchronous algorithms, including [1], in our revised manuscript.
>
> (2)In contrast to [1], our work addresses the synchronous setting and introduces a first-order stochastic gradient method, named MeFBO, for federated bilevel optimization. MeFBO accommodates $\textbf{local stochastic gradient estimation, local multi-step training, and data heterogeneity}$, all of which pose unique technical challenges. Specifically, we leverage step size and penalty parameters to control client drift bounds. These parameters are carefully tuned to align with the variance from local stochastic gradient estimation and errors arising from client sampling, ensuring convergence.
>
>
> (3)Compared to other federated learning methods discussed in Table 1 of our manuscript, MeFBO is not developed or analyzed from the perspective of hypergradient estimation. Instead, $\textbf{it relaxes the convexity assumption on the lower-level problem and employs a Hessian-free approach}$, enhancing scalability and computational efficiency.
>
>
> >(W.3.) I have some concerns about the complexity of the proposed algorithm. In each communication round, the algorithm requires solving an additional projection problem as described in Eq. (12). In bilevel deep learning, the variables are often high-dimensional, which increases the complexity of solving the optimization problem.
>
> $\textbf{Response:}$ (1)Compared to other federated bilevel optimization methods in Table 1 that focus on the unconstrained setting, this work addresses more general constrained federated bilevel optimization problems. Consequently, projection operators are incorporated into the algorithm. Thank you for raising concerns regarding projection operations. If the lower-level problem is constrained, we assume that projections onto the sets $X$ and $Y$ are computationally simple.
>
> (2)In many machine learning applications, the lower-level problem is either unconstrained or involves simple box constraints. In such cases, projection operators are either unnecessary or have closed-form solutions, ensuring computational efficiency. Moreover, in our algorithm, MeFBO, the projection operators are implemented exclusively on the server side, further enhancing its practicality and efficiency.

---

> > ### Author Response · Authors · 2024-11-26
> >
> > For W2, we have included a discussion of asynchronous algorithms, including [1], in the Related Work section of our revised manuscript.

---

> ### Author Response · Authors · 2024-11-22
> **Response to theoretical concerns (Part 2/2)**
>
> >(W.5.) Additional comparisons of the convergence rate between the proposed method and existing distributed centralized (federated) bilevel methods should be included to demonstrate the superiority of the proposed approach.
>
> $\textbf{Response:}$ (1)This work focuses on non-convex federated bilevel optimization, whereas the closely related works listed in Table 1 of our manuscript typically assume the strong convexity of the lower-level problem and use the hypergradient norm as their stationarity measure. In contrast, our method does not rely on strong convexity in the lower-level problem and adopts a Hessian-free approach. Specifically, we use a stationarity measure,  $R_t$, tailored for non-convex lower-level objectives, which offers a broader range of applications.
>
> (2)As stated in lines 309–311 of the first submitted manuscript, for strongly convex lower-level problems, our stationarity measure  $R_t$  can be compared to the hypergradient norm, as shown in Lemma A.13 of Liu et al. (2024). Thank you for your suggestion. We will include a comparison of  $R_t$  and the hypergradient norm in Appendix G, focusing on the case where the lower-level objective function is strongly convex.
>
>
> >(W.6.) The readability of this work is poor, as it contains numerous notations, some of which lack sufficient explanations. For instance, the exact meaning of $\nabla_1$ and $\nabla_2$ in Eq. (9) is not clearly defined.
>
> $\textbf{Response:}$ Thank you for the reminder.  $\nabla_1$  and  $\nabla_2$  denote the gradients of a function with respect to its first and second variables, respectively.

---

> ### Author Response · Authors · 2024-11-22
> **Response to experimental concerns**
>
> > (W.4.1) Only the MNIST dataset is used to evaluate the proposed method. Could you conduct experiments on larger-scale datasets?
>
> $\textbf{Response:}$ Thank you for the suggestion. We have expanded our experiments to include larger-scale datasets, such as Fashion MNIST and CIFAR-10, within the context of data hyper-cleaning. The results have been added to Section F of the appendix in the updated manuscript. These latest results demonstrate that our algorithm maintains consistent performance and continues to achieve state-of-the-art results.
>
>
> >(W.4.2) The evaluation is limited to hyper-parameter optimization tasks. It is unclear if the proposed method is applicable to other bilevel optimization tasks, such as meta-learning in [2] or neural architecture search (NAS) in [3]. Could you provide additional experimental results on these tasks?
>
> $\textbf{Response:}$ First, our algorithm is designed to address general bilevel optimization problems, making it applicable to a variety of tasks beyond hyperparameter optimization. Notably, it can handle simple constrained problems and accommodates non-convexity in the lower-level problem.
>
> Second, our experiments include hyper-representation learning, which is categorized as a meta-learning task. For a detailed discussion, please refer to Section 4 of Franceschi et al. (2018).
>
> Thank you for your suggestion. We have performed additional experiments using larger-scale datasets and models within the context of data hyper-cleaning. The results demonstrate that the proposed method consistently achieves state-of-the-art performance. Detailed results are provided in Section F of the appendix in the updated manuscript.
>
> >(W.4.3) The experiments are conducted on 2- or 3-layer multi-layer perceptrons, raising concerns about the method’s efficiency when applied to models with more parameters. This point is also related to W.3, as larger-scale models may significantly impact the efficiency of the proposed algorithm.
>
> $\textbf{Response:}$ Thank you for the suggestion. We have expanded our experiments to include a 7-layer convolutional neural network (CNN) within the context of data hyper-cleaning. The results have been added to Section F of the appendix in the updated manuscript. These findings confirm that our algorithm maintains consistent performance and continues to achieve state-of-the-art results.
>
>
>
>
> >(W.7.) In the experiment, the author claims that MeFBO outperforms other methods in both i.i.d. and non-i.i.d. settings. I would appreciate it if the authors could provide more detailed explanations and discussions on these results.
>
> $\textbf{Response:}$
> First, we would like to clarify that $\textbf{we did not claim MeFBO outperforms all other methods in every experiment}$ under both i.i.d. and non-i.i.d. settings. As noted in line 402 of the first submitted manuscript, in repeated experiments with 300 communication rounds, SimFBO (Yang et al., 2023) achieves the best accuracy when the  $rc$  value is not equal to 0.07. For a more intuitive comparison, please refer to Figure 5.
>
> Second, we have provided detailed explanations in the “Results” sections of the respective experiments, showcasing our observations and insights from the numerical experiments. Thank you for your suggestion. We have conducted additional experiments using larger-scale datasets and models in the context of data hyper-cleaning. We will include more detailed explanations and discussions on these results, which demonstrate that the proposed method consistently achieves state-of-the-art performance. The detailed results are presented in Section F of the appendix in the updated manuscript.

---

### Official Review · Reviewer_L3De · 2024-11-01

**Soundness:** 3
**Presentation:** 3
**Contribution:** 2
**Rating:** 6
**Confidence:** 3

**Summary:**

The author(s) developed FedBO, an algorithm that solves the federated bilevel optimization problems without assuming the lower-level objective to be strongly convex. On the theoretical side, the convergence properties of FedBO is studied; on the practical side, FedBO shows superior performance against baselines on some toy learning tasks based on the MNIST datasets.

**Strengths:**

- The paper is well-written, related work are well-addressed to my knowledge;
- The author(s) studied MeFBO both theoretically and empirically.

**Weaknesses:**

- The motivation of MeFBO is ok but not very convincing to me:
    - The paper relaxs the LL-convexity assumption, which feels like the author(s) developed a new algorithm in a slightly different setting using existing analytical tools;
    - The hessian-vector product computation is not very expensive using existing auto-differentiation framework such as PyTorch, so the motivation for avoiding Hessian-vector product is only partially convincing.
- Experiments are based on the tiny MNIST dataset. I understand that the paper is not really motivated from applications in practice, but the experiments can still be improved. For example, the author(s) could use larger dataset and maybe include some other more interesting learning tasks.

**Questions:**

- Is it really technically challenging to relax the LL-convexity assumption? Or is the analysis based on simple combination of existing proof templates?
- Is bilevel optimization really being used in pratical federated learning applications? Or is it just a interesting topic from the theoretical side?

---

> ### Author Response · Authors · 2024-11-22
> **Response to Weaknesses.**
>
> >(W. 1.) The motivation of MeFBO is ok but not very convincing to me: (1) The paper relaxs the LL-convexity assumption, which feels like the author(s) developed a new algorithm in a slightly different setting using existing analytical tools; (2) The hessian-vector product computation is not very expensive using existing auto-differentiation framework such as PyTorch, so the motivation for avoiding Hessian-vector product is only partially convincing.
>
> $\textbf{Response:}$ Thank you for your thoughtful comment, which provides us with an opportunity to explain more clearly the motivation behind MeFBO.
>
> $\textbf{Challenges with Strong Convexity Assumption.}$
> In many federated bilevel learning scenarios, the lower-level loss function does not naturally satisfy the strong convexity assumption, particularly with MLP or CNN architectures. To address this, some models introduce regularization terms to enforce strong convexity. However, this approach raises two significant challenges:
> (i)Regularization Parameter Selection: Choosing an appropriate regularization parameter can be difficult due to the complexity of neural architectures.
>
> (ii)Robustness Issues: Algorithms relying on Hessian-based computations may become sensitive to the choice of the regularization parameter. This sensitivity is highlighted in our analysis of robustness to $rc$ in Subsection 4.1 (see Figure 1). Additional experiments in Section A of the appendix in the first submitted manuscript (Figures 4, 8–9, and 12) further demonstrate that regularization techniques used to enforce strong convexity can degrade performance. These findings emphasize the urgent need to design federated bilevel optimization algorithms tailored for non-convex scenarios.
>
> $\textbf{Memory Consumption.}$
> The Hessian-Vector Product (HVP) computations in closely related works (see Table 1) arise from designing and analyzing federated bilevel optimization algorithms through the lens of hypergradient estimation. This approach incurs significant memory consumption, often 2–3 times higher than gradient-based methods. For example, in our supplementary experiments on CNN-CIFAR for data hyper-cleaning (Section F in the appendix), SimFBO was unable to complete the task on our devices due to insufficient memory caused by HVP computations.
>
>
> >(W. 2.)Experiments are based on the tiny MNIST dataset. I understand that the paper is not really motivated from applications in practice, but the experiments can still be improved. For example, the author(s) could use larger dataset and maybe include some other more interesting learning tasks.
>
> $\textbf{Response:}$  Thank you for the suggestion. We have expanded our experiments to include larger-scale datasets, such as Fashion MNIST and CIFAR-10, within the context of data hyper-cleaning. The results have been added to Section F of the appendix in the updated manuscript. These latest results demonstrate that our algorithm maintains consistent performance and continues to achieve state-of-the-art results.

---

> ### Author Response · Authors · 2024-11-22
> **Response to Questions**
>
> >(Q. 1.) Is it really technically challenging to relax the LL-convexity assumption? Or is the analysis based on simple combination of existing proof templates?
>
> $\textbf{Response:}$
> First, we would like to emphasize that, to the best of our knowledge, prior to the submission of this work, no studies had addressed federated bilevel optimization with non-convex lower-level problems.
>
> Second, this work extends MEHA by Liu et al. (2024) to the federated bilevel optimization setting. However, this extension is neither straightforward nor simple. Unlike MEHA, which is designed for deterministic settings, our algorithm operates in a stochastic environment, requiring a comprehensive sample complexity analysis. Additionally, our method is tailored for distributed systems, necessitating a communication complexity analysis, whereas MEHA is limited to single-machine settings.
>
> Third, the technical novelty of our approach lies in addressing both sample complexity and communication complexity under stochastic gradient estimation in the federated setting. Our method also accounts for partial client participation and data heterogeneity, which introduce unique challenges. Specifically, we carefully tune step size and penalty parameters to control client drift bounds, aligning them with the variance from local stochastic gradient estimation and errors from client sampling, thereby ensuring convergence.
>
>
> >(Q. 2.)Is bilevel optimization really being used in pratical federated learning applications? Or is it just a interesting topic from the theoretical side?
>
> $\textbf{Response:}$
> Good question! Recent developments have demonstrated the practical applications of federated bilevel optimization, particularly in the fine-tuning of large language models (LLMs) in federated settings. For example, the work [a] explores the use of federated bilevel optimization for local fine-tuning of LLMs. As highlighted in Table 2 of [a], such models can be optimized using either single-level or bilevel approaches. Notably, the bilevel optimization method (FedBiOT by [a]) demonstrates significant advantages over single-level optimization, particularly in scenarios requiring hierarchical problem structures.
>
>
> Additionally, the complexity of large models, such as LLMs, underscores the critical need for Hessian-free approaches, which provide greater efficiency and scalability in practical applications.
>
>
> [a] FedBiOT: LLM Local Fine-tuning in Federated Learning without Full Model

---

> ### Comment · Reviewer_L3De · 2024-12-02
> **Thanks for the author(s) response**
>
> I would like to thank the author(s) for the detailed response. The technical novelty still seems limited to me and I am maintaining my current score.

---

### Official Review · Reviewer_dTxG · 2024-11-01

**Soundness:** 2
**Presentation:** 2
**Contribution:** 2
**Rating:** 5
**Confidence:** 4

**Summary:**

This paper studies the Federated Bilevel Optimization (FBO) problem and puts forward a first-order method without the requirement of lower-level function strong convexity. Besides, it also provides theoretical analysis and sufficient experimental results to show the advantages of their method.

**Strengths:**

1. This paper has some interesting experimental observations on the previous FBO methods, such as the sensitivity.
2. There are solid experiments.

**Weaknesses:**

1. The presentation of this paper is confusing in some places. The important question that this paper asks on the first page involves both privacy and communication issues. Yet, there is no focus on privacy, which makes the question confusing to readers. Besides, there is a mismatch in the description of the work by Xiao & Ji (2023). In line 40, it is described as an AID-based method, and an ITD-based method in line 132.
2. I suggest that the authors mention the comparison between the two convergence measures in this paper (line 310) for the following reasons. First, it could allow readers to have a more smooth understanding. More importantly, it helps compare the sample complexity and communication costs.
3. In the convergence analysis, in eq (109), there is an expectation of the hypergradient on the left-hand side, which should be wrong. I guess it should be the expectation of the hypergradient norm. Meanwhile, it raises another question. Do authors compare the communication cost with previous works? If they do, previous works establish communication cost comparisons when the hypergradient norm square is small. So could authors please provide a more detailed comparison on it?
4. Lastly, the novelty of this paper is limited. In terms of communication, the result (current version) in Corollary does not have a big improvement over SimFBO. Admittedly, this paper is hessian-free and relaxes the assumption of strong convexity. However, it directly benefits from the Moreau envelope proposed by Liu et al. (2024). Please correct me if I missed anything.

**Questions:**

Please check the weaknesses.

---

> ### Author Response · Authors · 2024-11-22
> **Response to Weaknesses (part 1/2)**
>
> >(W. 1.) The presentation of this paper is confusing in some places. The important question that this paper asks on the first page involves both privacy and communication issues. Yet, there is no focus on privacy, which makes the question confusing to readers. Besides, there is a mismatch in the description of the work by Xiao & Ji (2023). In line 40, it is described as an AID-based method, and an ITD-based method in line 132.
>
> $\textbf{Response:}$
> Thank you for raising these concerns.
>
> Since the federated setting in the problem we consider already reflects the goal of preserving client privacy, we will revise the question on the first page to avoid confusing readers:
> “Can we develop a stochastic gradient method for FBO that does not require strong convexity in the lower-level problem, while reducing computation and communication costs?”
>
> Regarding the mismatch in the description of Xiao & Ji (2023), you are correct that there is a typo in line 132. The term “ITD-based” should be corrected to “AID-based.”
>
> >(W. 2.) I suggest that the authors mention the comparison between the two convergence measures in this paper (line 310) for the following reasons. First, it could allow readers to have a more smooth understanding. More importantly, it helps compare the sample complexity and communication costs.
>
> $\textbf{Response:}$ Thank you for your constructive suggestion. We will include a comparison of $R_t$ and the hypergradient norm in Appendix G. It is worth noting that the hypergradient norm typically assume the strong convexity of the lower-level problem. In contrast, our method does not rely on strong convexity in the lower-level problem and adopts a Hessian-free approach, utilizing a stationarity measure, $R_t$, specifically tailored for non-convex lower-level objectives, offering a broader range of applications.
>
>
> >(W. 3.) In the convergence analysis, in eq (109), there is an expectation of the hypergradient on the left-hand side, which should be wrong. I guess it should be the expectation of the hypergradient norm. Meanwhile, it raises another question. Do authors compare the communication cost with previous works? If they do, previous works establish communication cost comparisons when the hypergradient norm square is small. So could authors please provide a more detailed comparison on it?
>
> $\textbf{Response:}$
> Thank you for raising these concerns regarding Eq. (109). Upon further review, we confirm that Lines 2704–2745 should indeed be removed, as we have finished establishing the convergence results with respect to $R_t$ in Line 2703 of the first submitted manuscript.
>
> Additionally, we clarify that $\nabla F(x, y)$ in this context refers to the gradient of the upper-level objective function $F(x, y)$ and not the hypergradient. Consequently, this work does not use the hypergradient norm as the stationarity measure, as it may not be well-defined for cases where the lower-level problem is non-convex or constrained.
>
> This work focuses on non-convex federated bilevel optimization, whereas the closely related works listed in Table 1 of our manuscript generally assume strong convexity of the lower-level problem and rely on the hypergradient norm as their stationarity measure. Due to these fundamental differences, it is not applicable to theoretically compare the communication costs directly. However, experimental results provide a direct comparison of communication costs.
>
> This is why we present extensive comparison results for test accuracy v.s. communication rounds in the Numerical Experiments section. These results demonstrate that MeFBO achieves comparable or superior and more robust performance than existing FBO approaches. It is also worth noting that, as mentioned in line 402 of the first submitted manuscript, in repeated experiments with 300 communication rounds, SimFBO (Yang et al., 2023) achieves the best accuracy when the $rc$ value is not equal to 0.07. For a more intuitive comparison, please refer to Figure 5.

---

> > ### Comment · Reviewer_dTxG · 2024-11-25
> >
> > W1. If I understand correctly, Xiao & Ji (2023) is ITD-based method instead of AID-based method.
> > W2-W3. I understand that different convergence measures are used in different papers and it is OK to compare the communication cost with experiments.
> > W4. I did not receive any response of this part.

---

> > > ### Author Response · Authors · 2024-11-26
> > >
> > > For W1, thank you for pointing this out. This has been corrected in the revised version of the manuscript.

---

> > > > ### Comment · Reviewer_dTxG · 2024-12-01
> > > >
> > > > I have read the response to W4. However, I want to further ask the specific novelty of the analysis. I am aware that the authors applied the deterministic and single-task method under stochastic and federated settings. Then what are the challenges and how to solve them specially in your convergence analysis?

---

> > > > > ### Author Response · Authors · 2024-12-02
> > > > >
> > > > > $\textbf{Response:}$ Thank you for taking the time to review our work and for your valuable suggestions. We have addressed your questions as follows:
> > > > >
> > > > > In the convergence and complexity analysis of the proposed algorithm, the penalty and proximal parameters can be set relatively simply, as demonstrated in Theorem 3.4. The main challenge lies in $\textbf{carefully selecting step sizes to effectively control}$ $\textbf{estimation errors arising from variancein stochastic }$ $\textbf{estimation, data heterogeneity, and client drift}$, ensuring that the residual function is appropriately bounded using a standard Lyapunov function analysis with a suitably chosen Lyapunov function. The proof sketch is provided in Section D.2 and consists of two main steps (Steps 1 and 2). The key result is the estimation (33) in Step 2, whose proof is divided into Steps a–e, also detailed in Section D.2.
> > > > >
> > > > > $\textbf{Steps a and e are standard, but Steps b–d introduce significant differences between deterministic, non-federated settings}$ $\textbf{and stochastic, federated settings.}$ Specifically, compared to the deterministic, non-federated case, $\textbf{additional terms arise}$ in controlling the error of the distance between $\theta^{(t)}$ and $\theta_{\gamma}^*(x^{(t)}, y^{(t)})$. These extra terms include $\textbf{the server’s stochastic gradient estimation}$ $\mathbb{E}\\| h_{\theta}^{(t)} \\|^2$ and  $\textbf{the client drifts}$ $\Delta_x^{(t)}$, $\Delta_y^{(t)}$, and $\Delta_{\theta}^{(t)}$; see Step b for more details.
> > > > >
> > > > > Hence, we must effectively control these terms to harmonize them with others, ensuring that the difference between two iterations of the Lyapunov function is appropriately bounded (i.e., (33) holds). Specifically, the server’s stochastic gradient estimation $\mathbb{E}\\| h_{\theta}^{(t)} \\|^2$ is analyzed in Step c, where the interplay among partial client participation, multiple local iterations, and data heterogeneity is reflected in (35). The client drifts are bounded in Step d. Notably, the client drifts are influenced by their own values, with coefficients controlled by the number of local update rounds $\tau$ and client learning rates. They are also affected by the variance of stochastic gradient estimators, the measure of data heterogeneity, and the distance between $\theta^{(t)}$ and $\theta_{\gamma}^*(x^{(t)}, y^{(t)})$. By combining Steps a, b, c, and d, we ensure that the conditions in Lemma D.8 are satisfied through the appropriate adjustment of step sizes; see Lemma D.8 for details.

---

> ### Author Response · Authors · 2024-11-22
> **Response to Weaknesses (part 2/2)**
>
> >(W. 4.) Lastly, the novelty of this paper is limited. In terms of communication, the result (current version) in Corollary does not have a big improvement over SimFBO. Admittedly, this paper is hessian-free and relaxes the assumption of strong convexity. However, it directly benefits from the Moreau envelope proposed by Liu et al. (2024). Please correct me if I missed anything.
>
>
> $\textbf{Response:}$
> First, since different convergence measures are used in SimFBO (Yang et al., 2023) and this work, a direct comparison is not appropriate. Specifically, SimFBO employs the hypergradient norm as the stationarity measure, which assumes strong convexity in the lower-level problem. In contrast, our method is designed for general non-convex settings and utilizes a tailored stationarity measure,  $R_t$, defined via the penalty method, which is more broadly applicable.
>
> To evaluate communication efficiency, we present extensive comparative results of test accuracy v.s. communication rounds in the numerical experiments section. These results show that our method MeFBO achieves comparable or superior and more robust performance than existing FBO approaches.  It is also worth noting that, as mentioned in line 402 of the initial manuscript, in repeated experiments with 300 communication rounds, SimFBO (Yang et al., 2023) achieves the best accuracy when the $rc$ value is not equal to 0.07. For a more intuitive comparison, please refer to Figure 5.
>
>
> Although this work adopts the Moreau envelope reformulation proposed by Liu et al. (2024), extending their approach to the federated setting is neither straightforward nor trivial. Unlike MEHA, which is designed for deterministic settings, our algorithm operates in a stochastic environment, requiring comprehensive sample complexity analysis. Furthermore, our method is tailored for distributed systems, necessitating a communication complexity analysis, whereas MEHA is restricted to single-machine settings.
>
> $\textbf{Theoretical contributions.}$ The technical novelty of our approach lies in addressing both sample complexity and communication complexity under stochastic gradient estimation in federated settings. Our method also accounts for partial client participation and data heterogeneity, which pose unique challenges. Specifically, we carefully tune the step size and penalty parameters to control client drift bounds, aligning them with the variance from local stochastic gradient estimation and errors from client sampling, thereby ensuring convergence.
>
>
> $\textbf{Experimental contributions.}$ Existing federated bilevel optimization methods primarily focus on hypergradient estimation and rely on the strong convexity assumption in the lower-level problem. To ensure strong convexity, it is common to introduce regularization terms in the lower-level problem. This work challenges this approach, demonstrating that such regularization techniques can degrade performance (see Figure 1 in Subsection 4.1 and additional experiments in Figures 4, 8–9, and 12 in Section A of the appendix). These findings highlight the urgent need to design federated bilevel optimization algorithms tailored for non-convex scenarios.

---

> ### Author Response · Authors · 2024-11-25
> **Response to Visibility Issue for W4**
>
> Apologies for the oversight regarding W4. The visibility settings for our response were not updated previously. This has now been corrected, and you should be able to access the response.

---

### Official Review · Reviewer_XXEP · 2024-11-01

**Soundness:** 3
**Presentation:** 3
**Contribution:** 2
**Rating:** 5
**Confidence:** 3

**Summary:**

The paper concerns Federated Bilevel Optimization (FBO) under weaker conditions than previous work provided. It proposes a Moreau envelope-based min-max reformulation and develops a first-order stochastic gradient method for the general FBO problem. The paper then provides a convergence and complexity analysis, establishing a theoretical guarantee of convergence and communication complexity.  Numerical experiments are performed under various conditions, including federated loss function tuning on unbalanced datasets and federated hyper-representation, which demonstrate faster convergence and superior performance.

**Strengths:**

1.	Compared to previous learning algorithms for solving bilevel optimization, the proposed algorithm is developed under weaker assumptions of upper- and lower-level objectives, and hessian-free, giving an advantage in communication and computational efficiency in Federated Learning.
2.	The theoretical work contains two major novelties: (1). The convergence and complexity analysis is developed, releasing restrictive assumptions required in previous work; (2). The results obtained clarify the relationship between step size, number of participants, number of local iterations and convergence.
3.	Extensive experimental results on challenging settings demonstrate the efficiency of the proposed algorithm.

**Weaknesses:**

1.	There is a lack of technical novelty, as the analysis is based on previous work that leverages the Moreau envelope and hessian-free approach. For the theoretical part, it is unclear whether the complexity caused by the boundedness of the data heterogeneity introduces additional novelty beyond previous work. As a result, the proposed method seems like a combination of existing works.
2.	It lacks non-asymptotic analysis / numerical evaluations to demonstrate how the level of data heterogeneity affects convergence and robustness.
3.	The number of tuning parameters is relatively large in my opinion (local update rounds, server learning rates, client learning rates, penalty parameter, and proximal parameter). It could not be easy to choose those parameters in real applications, which affects the efficiency of the algorithm.
4.	It is recommended to list some examples of bilevel optimization that fit the assumptions of the upper and lower objectives, which will strengthen the motivation of this work.

**Questions:**

1.	The author states that the theoretical analysis is more complex compared to Liu et al. (2024) due to the bounds of client drifts. Could authors give intuitions on the proof, especially on the complexity caused by the client drifts?
2.	Could authors comment on the impact of the level of data heterogeneity on the convergence and robustness?
3.	Could authors provide guidance on the choice of the tuning parameter in algorithm 1?

---

> ### Author Response · Authors · 2024-11-22
> **Comment 1 Title: Response to Weaknesses (part 1/2).**
>
> >(W.1.) There is a lack of technical novelty, as the analysis is based on previous work that leverages the Moreau envelope and hessian-free approach. For the theoretical part, it is unclear whether the complexity caused by the boundedness of the data heterogeneity introduces additional novelty beyond previous work. As a result, the proposed method seems like a combination of existing works.
>
>
> $\textbf{Response:}$
> Thank you for highlighting the impact of the boundedness of data heterogeneity.
>
> The boundedness of data heterogeneity $\Delta^{2}$ appears explicitly in Theorem 3.4, specifically in the second and third terms on the right-hand side of Eq. (14). To further highlight the impact of $\Delta^2$, we can express Eq. (14) as:
>
> $ \min_{0\leq t\leq T-1}\mathbb{E}R_t(x^{(t+1)},y^{(t+1)})=\mathcal{O}\left(
>   \frac{1}{P}
>   \left(
>   \frac{1}{T^{1/2} \tau^{1/2}}
>   +
>   \frac{\tau^{1/2} (n-P)}{T^{1/2}n} \Delta^{2}
> 		+
>   \frac{1}{T\tau^{1/4} }M_3^{'}
>   \right)
>   \right), $
>
> where $M_3^{'} := M_{3}^{'}(\Delta^2)$ is a positive constant dependent on $\Delta^2$. Notably, $\Delta^2 = 0$ does not imply that $M_3^{'}(\Delta^2) = 0$, and the explicit form of $M_3^{'}$ is given in Eq.(87) in updated version and in Eq.(82) original version.
>
>
> As detailed in the submission (following Table 1), we provide a comprehensive comparison with related works to highlight the $\textbf{ technical novelty}$ of our approach.
>
> As detailed in the submission (after Table 1), we provide a comprehensive comparison with related works.
>
> First, compared to MEHA by Liu et al. (2024), our algorithm operates in a stochastic setting, necessitating a sample complexity analysis, whereas MEHA is designed for deterministic settings. Additionally, our method is tailored for distributed systems, requiring communication complexity analysis, while MEHA is a single-machine method. The technical novelty of our approach lies in addressing both sample complexity and communication complexity under $\textbf{stochastic gradient estimation}$, $\textbf{multi-round local updates}$, $\textbf{partial client participation}$,  $\textbf{client drifts}$ and $\textbf{data heterogeneity}$.
>
> >(W. 2.) It lacks non-asymptotic analysis / numerical evaluations to demonstrate how the level of data heterogeneity affects convergence and robustness.
>
> $\textbf{Response:}$ Thank you for the suggestion.
> First, we have explicitly incorporated the level of data heterogeneity, $\Delta^2$, into our analysis. To further emphasize its impact, we can express Eq. (14) as:
> $ \min_{0\leq t\leq T-1}\mathbb{E}R_t(x^{(t+1)},y^{(t+1)})=\mathcal{O}\left(
>   \frac{1}{P}
>   \left(
>   \frac{1}{T^{1/2} \tau^{1/2}}
>   +
>   \frac{\tau^{1/2} (n-P)}{T^{1/2}n} \Delta^{2}
> 		+
>   \frac{1}{T\tau^{1/4} }M_3^{'}
>   \right)
>   \right).$
>
> where $M_3^{'} := M_{3}^{'}(\Delta^2)$ is a positive constant dependent on $\Delta^2$. Notably, $\Delta^2 = 0$ does not imply that $M_3^{'}(\Delta^2) = 0$, and the explicit form of $M_3^{'}$ is given in Eq.(87) in updated version and in Eq.(82) original version..
>
> Here, $\Delta^2$ arises from the error control in the distance between $\theta^{(t)}$ and $\theta_{\gamma}^*(x^{(t)}, y^{(t)})$ in Lemma D.5, as well as client drifts, also analyzed in Lemma D.6. By retaining the contributions of $\Delta^2$ in these lemmas, we derive the convergence results shown above.
>
> Second, in our $\textbf{experiments}$, we use both i.i.d. and non-i.i.d. datasets to reflect different levels of data heterogeneity, following the approach used by Tarzanagh et al. (2022). To facilitate comparisons, we evaluate the results for i.i.d. and non-i.i.d. datasets (e.g., Figures 7 and 10). As shown in these figures, datasets with lower heterogeneity (e.g., i.i.d. datasets) demonstrate $\textbf{ faster convergence and greater robustness }$ compared to datasets with higher heterogeneity (e.g., non-i.i.d. datasets).

---

> ### Author Response · Authors · 2024-11-22
> **Response to Weaknesses (Part 2/2)**
>
> >(W. 3.) The number of tuning parameters is relatively large in my opinion (local update rounds, server learning rates, client learning rates, penalty parameter, and proximal parameter). It could not be easy to choose those parameters in real applications, which affects the efficiency of the algorithm.
>
> $\textbf{Response:}$
> Based on our experience, the step sizes on the client side and server side have a greater impact on convergence compared to other parameters. To simplify tuning, we set these two step sizes to be the same and perform a dense grid search. The penalty parameter $c_t = \underline{c}(t+1)^p$ requires the exponent $p$ to be sufficiently small.
>
> Below, we provide specific parameter ranges and examples used in federated loss function tuning on imbalanced datasets:
>
> •	Penalty Parameter: 1) $\underline{c} \in \\{2, 4, 6, 8, 10\\}$; 2) $p \in \\{0.001, 0.01, 0.1\\}$
>
> •	Proximal Parameter: The proximal parameter $1/\gamma$ is tuned within:
> $1/\gamma \in \\{0.005, 0.015, 0.03, 0.05, 0.1\\}$
>
> •	Step Sizes: 1) $\eta_x^{(t)}/\lambda_x^{(t)} \in [0.06, 0.2]$ with a step of 0.02;
> 2)	$\eta_y^{(t)}/\lambda_y^{(t)} \in [0.03, 0.1]$ with a step of 0.01;
> 3)	 $\eta_\theta^{(t)}/\lambda_\theta^{(t)} \in [0.006, 0.016]$ with a step of 0.002.
>
> As shown in Theorem 3.4, our algorithm places relatively simple requirements on the penalty and proximal parameters. In most cases, these parameters can be selected without significant difficulty, ensuring ease of implementation.
>
> >(W. 4.) It is recommended to list some examples of bilevel optimization that fit the assumptions of the upper and lower objectives, which will strengthen the motivation of this work.
>
> $\textbf{Response:}$ Thank you for your suggestion.
>
> First, in lines 284–296, we provide a detailed explanation of the assumptions on the upper- and lower-level objectives, as well as a comparison with related works. Our assumptions are relatively standard, primarily requiring Lipschitz continuity of $f(x, y)$ and $g(\cdot, y)$.
>
> Second, one example that satisfies these assumptions is the federated data hyper-cleaning task. In this case, the smoothness of the network's activation functions ensures that the Lipschitz continuity conditions are met. This is explicitly demonstrated in Equation (20) of the paper.

---

> ### Author Response · Authors · 2024-11-22
> **Response to Questions**
>
> >(Q. 1.) The author states that the theoretical analysis is more complex compared to Liu et al. (2024) due to the bounds of client drifts. Could authors give intuitions on the proof, especially on the complexity caused by the client drifts?
>
> $\textbf{Response:}$  First, we would like to clarify that, as detailed in the submission (after Table 1), compared to MEHA by Liu et al. (2024), the increased complexity in our theoretical analysis stems not only from the $\textbf{bounds of client drifts}$ but also from the following factors: $\textbf{stochastic gradient estimation}$, $\textbf{multi-round local updates}$ and $\textbf{partial client participation}$ and $\textbf{data heterogeneity}$.
>
> The client drifts directly affect the descent property of the objective function $\Psi_{c_t}(x^{(t)},y^{(t)})$ as defined in Equation (34). Specifically, From Lemma D.4, we observe how client drifts influence the descent property of $\Psi_{c_t}$.
> From Lemma D.5, we see that client drifts affect the error control in the distance between $\theta^{(t)}$ and $\theta_{\gamma}^*(x^{(t)}, y^{(t)})$ as well as the bounds of the server-side stochastic gradient estimation.
>
> To offer better intuition about the complexity of the theoretical analysis, we will provide a new proof sketch in subsection D.2 of the appendix  of the updated version. In particular, from Step 2 of the proof sketch, we will highlight how variance in stochastic estimation, data heterogeneity, and client drifts contribute to the overall complexity of the analysis.
>
> >(Q. 2.) Could authors comment on the impact of the level of data heterogeneity on the convergence and robustness?
>
> $\textbf{Response:}$
> First, we have explicitly incorporated the level of data heterogeneity, $\Delta^2$, into our analysis. To further emphasize its impact, we can express Eq. (14) as:
>
> $ \min_{0\leq t\leq T-1}\mathbb{E}R_t(x^{(t+1)},y^{(t+1)})=\mathcal{O}\left(
>   \frac{1}{P}
>   \left(
>   \frac{1}{T^{1/2} \tau^{1/2}}
>   +
>   \frac{\tau^{1/2} (n-P)}{T^{1/2}n} \Delta^{2}
> 		+
>   \frac{1}{T\tau^{1/4} }M_3^{'}
>   \right)
>   \right).$
>
> where $M_3^{'} := M_{3}^{'}(\Delta^2)$ is a positive constant dependent on $\Delta^2$. Notably, $\Delta^2 = 0$ does not imply that $M_3^{'}(\Delta^2) = 0$, and the explicit form of $M_3^{'}$ is given in Eq.(87).
>
> Here, $\Delta^2$ arises from the error control in the distance between $\theta^{(t)}$ and $\theta_{\gamma}^*(x^{(t)}, y^{(t)})$ in Lemma D.5, as well as client drifts, also analyzed in Lemma D.5. By retaining the contributions of $\Delta^2$ in these lemmas, we derive the convergence results shown above.
>
> Second, in our experiments, we use both i.i.d. and non-i.i.d. datasets to reflect different levels of data heterogeneity, following the approach used by Tarzanagh et al. (2022). To facilitate comparisons, we evaluate the results for i.i.d. and non-i.i.d. datasets (e.g., Figures 7 and 10). As shown in these figures, datasets with lower heterogeneity (e.g., i.i.d. datasets) demonstrate faster convergence and greater robustness compared to datasets with higher heterogeneity (e.g., non-i.i.d. datasets).
>
> >(Q. 3.) Could authors provide guidance on the choice of the tuning parameter in algorithm 1?
>
> $\textbf{Response:}$ Based on our experience, the step sizes on the client side and server side have a greater impact on convergence compared to other parameters. To simplify tuning, we set these two step sizes to be the same and perform a dense grid search. The penalty parameter $c_t = \underline{c}(t+1)^p$ requires the exponent $p$ to be sufficiently small.
>
> Below, we provide specific parameter ranges and examples used in federated loss function tuning on imbalanced datasets:
>
> •	Penalty Parameter: 1) $\underline{c} \in \\{2, 4, 6, 8, 10\\}$; 2) $p \in \\{0.001, 0.01, 0.1\\}$
>
> •	Proximal Parameter: The proximal parameter $1/\gamma$ is tuned within:
> $1/\gamma \in \\{0.005, 0.015, 0.03, 0.05, 0.1\\}$
>
> •	Step Sizes: 1) $\eta_x^{(t)}/\lambda_x^{(t)} \in [0.06, 0.2]$ with a step of 0.02;
> 2)	$\eta_y^{(t)}/\lambda_y^{(t)} \in [0.03, 0.1]$ with a step of 0.01;
> 3)	 $\eta_\theta^{(t)}/\lambda_\theta^{(t)} \in [0.006, 0.016]$ with a step of 0.002.

---

> > ### Comment · Reviewer_XXEP · 2024-11-23
> > **Response to rebuttal**
> >
> > Thank you for your response. I appreciate it, but I will maintain my current rating.

---

### Meta-Review · Area_Chair_suKk · 2024-12-14

**Metareview:**

This paper studies federated bilevel optimization (FBO). By leveraging a Moreau envelope-based minimax reformulation, the proposed method allows the lower-level problem to be nonconvex, which is a weaker assumption comparing with existing works. Though the paper presents some interesting ideas, the reviewers found that the novelty is limited, as the results do not have significant improvement over some existing works. The overall presentation is not clear, and it takes a large amount of work for a satisfying revision. The numerical experiments also need to be strengthened in order to provide more informative conclusions.

**Additional Comments On Reviewer Discussion:**

Further discussed the technical novelty.

---

### Decision · Program_Chairs · 2025-01-22

Reject